# RANDOM SPARSE LIFTS: CONSTRUCTION, ANALYSIS AND CONVERGENCE OF FINITE SPARSE NETWORKS

**David A. R. Robin**
INRIA - ENS Paris
PSL Research University

**Kevin Scaman**
INRIA - ENS Paris
PSL Research University

**Marc Lelarge**
INRIA - ENS Paris
PSL Research University

## ABSTRACT

We present a framework to define a large class of neural networks for which, by construction, training by gradient flow provably reaches arbitrarily low loss when the number of parameters grows. Distinct from the fixed-space global optimality of non-convex optimization, this new form of convergence, and the techniques introduced to prove such convergence, pave the way for a usable deep learning convergence theory in the near future, without overparameterization assumptions relating the number of parameters and training samples. We define these architectures from a simple computation graph and a mechanism to lift it, thus increasing the number of parameters, generalizing the idea of increasing the widths of multi-layer perceptrons. We show that architectures similar to most common deep learning models are present in this class, obtained by sparsifying the weight tensors of usual architectures at initialization. Leveraging tools of algebraic topology and random graph theory, we use the computation graph's geometry to propagate properties guaranteeing convergence to any precision for these large sparse models.

## 1 INTRODUCTION

We consider the supervised learning task, of infering a function $f^\star : \mathcal{X} \to \mathcal{Y}$ from pairs of samples $(x, f^\star(x)) \in \mathcal{X} \times \mathcal{Y}$, with $x$ following a distribution $\mathcal{D}$. We restrict our attention to the case $\mathcal{X} = \mathbb{R}^d$ and $\mathcal{Y} = \mathbb{R}^k$, with a parametric model $F : \mathbb{R}^m \times \mathbb{R}^d \to \mathbb{R}^k$, approximating the target function as $F(\theta^\star, \cdot) : \mathbb{R}^d \to \mathbb{R}^k$ for a fixed parameter $\theta^\star \in \mathbb{R}^m$. The loss $\mathcal{L} : \theta \mapsto \mathbb{E}_{(x,y)} \left[ \| F(\theta, x) - y \|_2^2 \right]$ is used to "learn" the parameter $\theta : \mathbb{R}_+ \to \mathbb{R}^m$ by a gradient flow $\partial_t \theta = -\nabla \mathcal{L}(\theta)$ from $\theta_0 \in \mathbb{R}^m$.

In the context of deep learning, contrary to more generic non-convex optimization, we are not *intrinsically* interested in global minima of the fixed-dimension parametric loss $\mathcal{L}$, or in this model $F$ specifically. This optimization is only a means to the end of finding, among a family of models $F_m : \mathbb{R}^m \times \mathbb{R}^d \to \mathbb{R}^k$ of varying sizes $m \in \mathbb{N}$, one model-parameter pair $(F_m, \theta^\star \in \mathbb{R}^m)$ such that $F_m(\theta^\star, \cdot) \approx f^\star$. The ability of families of models, given as neural networks, to "scale up" to consistently avoid optimization issues is what the present theory aims to explain and predict.

**Challenges.** The class of models empirically known to reach near-zero loss when scaled up is large, and it remains unclear whether it is possible to construct a formalism both sufficiently flexible to cover all such neural networks and sufficiently rigid to allow non-trivial claims. For two-layer networks of width $h$ with parameters $\theta \in \mathbb{R}^{d \times h} \times \mathbb{R}^{h \times k}$, Cybenko (1989) and Barron (1993) have shown for various non-linearities $\sigma$ that for any continuous $f^\star : \mathcal{X} \to \mathbb{R}^k$ with compact domain $\mathcal{X} \subseteq \mathbb{R}^d$, and any threshold $\varepsilon \in \mathbb{R}_+^\star$, as $h \to +\infty$, there exists $\theta$ such that $\sup_x \| f_\theta(x) - f^\star(x) \|_2 \leq \varepsilon$. We call the minimal such width $h$ the *approximation number* $h_0(f^\star, \varepsilon) \in \mathbb{N}$. This existence result does not imply that a gradient flow will converge to such a parameter. Nonetheless, the question of whether a given architecture enjoys similar approximation properties has sparked much interest over the following decades, with extensions to more layers Leshno et al. (1993), more non-linearities (Pinkus, 1999), convolutional architectures (Cohen & Shashua, 2016), graph-processing (Xu et al., 2019; Brüel Gabrielsson, 2020) and outside Euclidean geometry (Kratsios & Bilokopytov, 2020), and with investigations of the link between width and depth (Poggio et al., 2017; Johnson, 2019), and handling of symmetries (Lawrence, 2022). For the harder question of convergence, with two layers, Robin et al. (2022, Proposition 4.6) shows that for any $\varepsilon \in \mathbb{R}_+^\star$ and $\delta \in ]0, 1[$, there exists a *learning number* $h_1(f^\star, \varepsilon, \delta) \in \mathbb{N}$ such that if $h \geq h_1$, then under standard initialization schemes, a gradient flow will

converge to a loss value below $\varepsilon$ with probability greater than $(1 - \delta)$. Note that earlier infinite-width results hinted at the existence of this number (without assumptions on the sample count), e.g. Chizat & Bach (2018). The problem of focusing on only approximation is that the corresponding learning number could be orders of magnitude larger than the approximation number, since it corresponds to a much stronger property: the target function is not only *representable* with a model, it is *learnable* with this model. Universal approximation is insufficient to predict which models will perform better in practice, because it fails to quantify the learning number $h_1$, which could be intractably larger than the approximation number $h_0$; in which case at all tractable sizes $h \in \mathbb{N}$ with $h_0 \leq h < h_1$, we would observe models not reaching low loss despite the existence of parameters in dimension $h$ with lower loss. We seek to upper-bound learning numbers when approximation has been established. This size-varying gradient flow point of view is meant to model practical deployment of neural networks, in contrast with works that aim to find a global minimum of fixed size by e.g. convex reformulations of two and three layers ReLU networks (Ergen & Pilanci, 2021; Wang et al., 2021).

To curb the difficulty of constructing bounds for learning numbers with more than two layers, many attempts have focused on the simpler setting of a fixed dataset $X \in \mathcal{X}^n$ of $n \in \mathbb{N}$ training samples and growing model size, and provided (typically asymptotic) bounds for this $X$, independently of $f^\star$, by leveraging positive-definiteness of the neural tangent kernel (NTK) matrix under sufficient overparameterization (Jacot et al., 2018; Du et al., 2018; Ji & Telgarsky, 2020; Chen et al., 2021). This allows viewing functions $\mathcal{X} \to \mathbb{R}$ as finite-dimensional vectors in $\mathbb{R}^n$, by restriction to $X$. In contrast, our objective is to derive bounds for fixed $f^\star$ independently of $n$ (thus infinite dimension), to avoid overparameterization assumptions relating number of parameters and samples, so our bounds depend on a (model-specific) "complexity" of $f^\star$, but do not diverge with infinitely many samples.

**Related formalizing works.** We obtain convergence guarantees for broad classes of "neural networks" when "lifted", by formalizing a precise meaning for these terms, usable in rigorous proofs. The class of neural networks that can be described with this theory is more expressive than the *Tensor Programs* of Yang & Littwin (2021), in particular because it allows non-linear weight manipulations, and distinct "scaling dimensions" (e.g. embedding dimension versus hidden layer width in transformers, which need not be related). In that sense, the core idea is much closer to the description with *Named Tensors* of Chiang et al. (2023). Thus the focus is on large but finite-size networks, contrary to infinite-width limits. However, the convergence shown applies only to sparse networks (in the sense of Erdös-Renyi, different from the *Lottery Ticket* (Frankle & Carbin, 2019) use of the term "sparse"). Several ideas used in our proofs are inspired by Pham & Nguyen (2021), which — in the context of non-quantitative mean-field convergence — introduce tools to show that throughout training, the distribution of weights, albeit finitely supported, has positive density in a neighbourhood of any point.

**Contributions.** We discuss the tangent approximation property in Section 2, and show how it implies probably approximately correct convergence when the model size grows. We introduce in Section 3 a formalism to build architectures having by construction a way to lift, and recover usual architectures in this form. We show in Section 4 that when such systems are sparsified at initialization in a specific way, then tangent approximation is satisfied, thus PAC-convergence of gradient flows when lifting.

## 2 CONVERGENCE BY TANGENT APPROXIMATION

Let $\mathcal{X} \subseteq \mathbb{R}^d$ be a compact set, and let $\mathcal{Y} = \mathbb{R}^k$. Let $\mathcal{F}$ be the set of continuous functions from $\mathcal{X}$ to $\mathcal{Y}$. Let $\mathcal{D}$ be a distribution on $\mathcal{X}$. Let $f^\star : \mathcal{X} \to \mathcal{Y}$ be a continuous function. Let $\mathcal{L} : \mathcal{F} \to \mathbb{R}_+$ be the function $\mathcal{L} : f \mapsto \mathbb{E}_{x \sim \mathcal{D}} \left[ \|f(x) - f^\star(x)\|_2^2 \right]$. We write for shortness $\|\cdot\|_{\mathcal{D}}^2 : f \mapsto \mathbb{E}_{x \sim \mathcal{D}} \left[ \|f(x)\|_2^2 \right]$.

Let $(\mathcal{S}, \preceq)$ be an infinite partially ordered set. For $s \in \mathcal{S}$, let $G_s$ be a finite set representing "size-$s$ architectures", and let $\Theta_s$ be a finite-dimensional vector space with an inner product, of corresponding "weights". For every $s \in \mathcal{S}$, let $\mathcal{N}_s$ be a distribution on $G_s \times \Theta_s$. For every $s \in \mathcal{S}$ and $g \in G_s$, let $F_{(s,g)} : \Theta_s \to \mathcal{F}$ be a differentiable function. Let $\mathcal{S}_0 \subseteq \mathcal{S}$ be an infinite set, called "admissible set".

We will illustrate constructions with two examples, of (possibly sparse) two-and three-layer networks without biases to learn a function $\mathbb{R}^d \to \mathbb{R}$. For two layers, $\mathcal{S} = \mathbb{N}$ is the number of hidden nodes and $G_s = \{0, 1\}^{d \times s} \times \{0, 1\}^{s \times 1}$ the sparsity pattern of the layers. For three layers, $\mathcal{S} = \mathbb{N}^2$ is the number of hidden nodes of each layer, and $G_{s_0, s_1} = \{0, 1\}^{d \times s_0} \times \{0, 1\}^{s_0 \times s_1} \times \{0, 1\}^{s_1 \times 1}$ the corresponding pattern. A choice of admissible subset — e.g. $\mathcal{S}_0 = \{(s_0, s_1) \in \mathbb{N}^2 \mid s_0 \leq s_1\}$ of networks whose width increases with depth — rules out potentially ill-behaved sizes, to formulate statements like "all networks (not ill-behaved) above a given size (w.r.t. $\preceq$ on $\mathcal{S}$) have property $P$".

For $s \in \mathcal{S}$, $g \in G_s$, and any $(\kappa, \varepsilon) \in \mathbb{R}_+ \times \mathbb{R}_+^*$, let $\mathcal{A}_{s,g}(\kappa, \varepsilon) \subseteq \Theta_s$ be the set defined as

$$\theta \in \mathcal{A}_{s,g}(\kappa, \varepsilon) \Leftrightarrow \inf_{u \in \Theta_s, \|u\| \leq \kappa} \left\| F_{(s,g)}(\theta) + \mathrm{d}F_{(s,g)}(\theta) \cdot u - f^\star \right\|_{\mathcal{D}}^2 < \varepsilon$$

The function $\mathcal{A}_{s,g}(\,\cdot\,, \varepsilon)$ is easily seen to be increasing. Let us flesh out some intuition about it. If $\mathcal{A}_{s,g}(0, \varepsilon) \neq \varnothing$, then there exists a choice of weights $\theta$ at size $s$ such that $F_{(s,g)}(\theta)$ approximates $f^\star$ to error $\varepsilon$. This does not guarantee however that such weights are "learnable". The construction $\mathcal{A}_{s,g}(\kappa, \varepsilon)$ with a tiny non-zero $\kappa$ allows for some flexibility: the target is approximable by a linearization of $F_{(s,g)}$ with a tiny ($\|u\| \leq \kappa$) variation of weights. Hence, $\mathcal{A}_{s,g}(\kappa, \varepsilon)$ is much larger than $\mathcal{A}_{s,g}(0, \varepsilon)$, but under some regularity conditions on $F_{s,g}$ there should always be a small ball around a point of $\mathcal{A}_{s,g}(0, \varepsilon)$ that is included in $\mathcal{A}_{s,g}(\kappa, \varepsilon)$ (see Figure 1 for instance). This regularity is irrelevant for universal approximation theorems, but crucial for learning theorems. The following property is sufficient to get convergence to low loss with high probability. Intuitively, we need tangent approximation with tangent radius $\kappa$, not for a point $\theta_0$, but an entire ball around $\theta_0$, with a tangent radius allowed to grow affinely with the ball radius. Write $B_s(\theta_0, R) = \{\theta \in \Theta_s \mid \|\theta - \theta_0\| \leq R\}$ the $R$-ball in $\Theta_s$ around $\theta_0$.

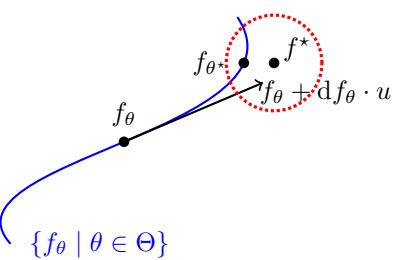

Figure 1: Tangent approximation to $\varepsilon$ error ($\varepsilon$-ball depicted by a dotted line) for fixed $s \in \mathcal{S}$, $g \in G_s$ with $f = F_{(s,g)}$. If $\|f_{\theta^\star} - f^\star\|_{\mathcal{D}}^2 < \varepsilon$, then $\theta^\star \in \mathcal{A}(0, \varepsilon)$. In contrast, $\theta \in \mathcal{A}(\|u\|, \varepsilon) \setminus \mathcal{A}(0, \varepsilon)$.

**Condition C1** (Tangent approximation with high probability). *Let $(\varepsilon, \delta) \in \mathbb{R}_+^* \times \,]0, 1]$. There exists $c \in \mathbb{R}_+$ (a constant intercept) such that for any $R \in \mathbb{R}_+$ (a maximal radius), there exists $s_1 \in \mathcal{S}$ (a threshold size) such that for all $s \in \{s \in \mathcal{S}_0 \mid s \succeq s_1\}$ (admissible sizes above threshold),*

$$\mathbb{P}_{(g,\theta) \sim \mathcal{N}_s}\Big( \forall r \leq R, \; B_s(\theta, r) \subseteq \mathcal{A}_{s,g}\left(c + r, \varepsilon\right) \Big) \geq (1 - \delta)$$

In words, with a three-layer network, this is satisfied if for an arbitrarily large radius $R$, there exists a threshold size $s_1 = (H_0, H_1) \in \mathbb{N}^2$ such that if $s_0 = (h_0, h_1)$ is any network size satisfying $h_0 \leq h_1$ (i.e. $s_0 \in \mathcal{S}_0$) and $h_0 \geq H_0$ and $h_1 \geq H_1$ (i.e. *both* hidden layers are above the threshold size), then with high probability over a random weight $\theta$, an entire ball $B_s(\theta, R)$ around this initialization has tangent approximation. We expect in this setting that we can't say anything for networks whose width is not increasing with depth (they could be ill-behaved), but other than that, any configuration above the threshold size has (with high probability) the desirable property of tangent approximation *at initialization* and *on a large ball* around the randomly drawn initialization. Therefore if this holds, we can expect that a gradient flow will have some desirable properties (granted by tangent approximation) for a long time after initialization, until it eventually exits the ball. The desirable property in question is a "sufficient decrease" property of the loss, materialized by a local separable Kurdyka-Łojasiewicz inequality. Indeed, by a parallelogram identity, $-2\langle \mathrm{d}F(\theta) \cdot u, F(\theta) - f^\star \rangle = \|\mathrm{d}F(\theta) \cdot u\|^2 + \|F(\theta) - f^\star\|^2 - \|F(\theta) + \mathrm{d}F(\theta) \cdot u - f^\star\|^2$. Moreover, for a gradient flow $\partial_t \theta = -\nabla \ell(\theta)$, we have $\partial_t(\ell \circ \theta) = -\|\nabla \ell(\theta_t)\|_2^2$. Leveraging the variational form of the $\ell_2$-norm, we get $-\partial_t(\mathcal{L} \circ F) = -\sup_u 4\langle \mathrm{d}F(\theta) \cdot u, F(\theta) - f^\star \rangle^2 / \|u\|_2^2 \geq -(\mathcal{L} \circ F - \varepsilon)_+^2 / (c + \|\theta - \theta_0\|)^2$ for as long as $\theta \in B_s(\theta_0, R)$. The separable structure of this bound with respect to $\mathcal{L} \circ F$ and $\|\theta - \theta_0\|$, together with the ability to choose $R$ arbitrarily large, is sufficient to conclude.

To get a feeling of why this assumption might be satisfied, picture a large sparse network with a "diverse enough" sparsity pattern (the clarification of what this means is the subject of this paper). If there exists a network, sparse or not, of small size that approximates the target function, then at initialization, there should be a myriad of small subnetworks identical to that approximator inside the large network. If the last layer is linear, then "selecting" a convex combination of these outputs is sufficient to reach low loss. Although the output of the large network may differ from the target, there is a modification of its last-layer weights that approximates the target function. Lastly, if there are many such subnetworks, then a small modification of parameters cannot substantially modify them all if the large network is large enough. The following theorem ensures that tangent approximation is indeed sufficient, the rest of the paper gives a precise meaning to the intuitions of this first section.

**Convergence Criterion 1.** *Let $\Theta$ be a vector space. The pair $(\mathcal{L} : \Theta \to \mathbb{R}_+, \theta_0 \in \Theta)$ satisfies this criterion with limit error $\varepsilon \in \mathbb{R}_+$ and constant $\kappa \in \mathbb{R}_+^*$ if for all $\theta : \mathbb{R}_+ \to \Theta$ such that $\theta(0) = \theta_0$ satisfying $\partial_t \theta = -\nabla \mathcal{L}(\theta)$, it holds $\forall t \in \mathbb{R}_+^*$, $\mathcal{L}(\theta_t) \leq \varepsilon + 1/\sqrt[3]{\kappa t + 1/c}$ where $c = \mathcal{L}(\theta_0)^3 \in \mathbb{R}_+$.*

**Theorem 2.1** (Probably approximately correct convergence in loss). *Let $(\varepsilon_0, \delta_0) \in \mathbb{R}_+^* \times ]0,1]$. If for $(g, \theta_0) \sim \mathcal{N}_s$ the variable $\mathcal{L} \circ F_{(s,g)}(\theta_0) \in \mathbb{R}_+$ is bounded uniformly in $s$ with high probability, and if Condition C1 is satisfied for parameters $(\varepsilon_0, \delta_0)$, then for any $\varepsilon > \varepsilon_0$ and $\delta > \delta_0$, there exists $\kappa \in \mathbb{R}_+^*$ and $s_1 \in \mathcal{S}$ such that for all $s \in \mathcal{S}_0$ satisfying $s \succeq s_1$ and $(g, \theta_0) \sim \mathcal{N}_s$, with probability at least $(1 - \delta)$, the pair $(\mathcal{L} \circ F_{(s,g)}, \theta_0)$ satisfies Convergence Criterion 1 with error $\varepsilon$ and constant $\kappa$.*

This statement is a slight extension of Robin et al. (2022, Proposition 4.6) with nearly identical proof. The main contribution of this paper is the construction of the sparsification in the distribution $\mathcal{N}_s$ satisfying by construction the tangent approximation condition when lifting. This includes architectures such as multi-layer perceptrons, convolutional networks, and attention-like mechanisms.

## 3 PERCEPTRON MODULES

For consistency and shorter notations, we use the *co-product* symbol $\coprod$ for disjoint unions. Formally, for a family of sets $(A_c)_c$ indexed by $c \in C$, define $\coprod_{c \in C} A_c = \bigcup_{c \in C} \{(c,a) \mid a \in A_c\}$. A surjective function $p : Y \to V$ from a set $Y$ to a finite set $V$ induces a partition of $Y$ into $Y_v = p^{-1}(v)$ indexed by $v \in V$, i.e. there is a bijection (a.k.a an isomorphism of sets) $Y \approx \coprod_{v \in V} Y_v$. It is equivalent to specify $p$ or the partition of $Y$. Given $Y$, a choice of one element in each part (called a *section*[1]) is a function $s : V \to Y$ such that $\forall v, s(v) \in Y_v$ (i.e. $p \circ s = \mathrm{Id}_V$). This is also written $s \in \prod_{v \in V} Y_v$.

**Definition 3.1** (Euclidean bundle over a finite set). *Let $V$ be a finite set. A Euclidean bundle over $V$ is a set $Y$ and a partition $\coprod_{v \in V} Y_v$, such that for any $v \in V$, the set $Y_v$ has the structure of a finite-dimensional $\mathbb{R}$-vector-space equipped with an inner product (i.e. $Y_v$ is a Euclidean space).*

The simplest example of a Euclidean bundle is $Y = V \times \mathbb{R}$, which "attaches" to each $v \in V$ the Euclidean space $Y_v = \mathbb{R}$. The interesting property of these objects is that all attached vector spaces need not have the same dimension, e.g. $\mathbb{R}^n \coprod \mathbb{R}^{k \times m}$ with $V = \{0, 1\}$ is a Euclidean bundle, whose sections are pairs $(u, w) \in \mathbb{R}^n \times \mathbb{R}^{k \times m}$. We will thus use these objects to describe parameter spaces.

**Definition 3.2** (Pullbacks of bundles and sections). *Let $Y = \coprod_{v \in V} Y_v$ be a Euclidean bundle over $V$, and let $U$ be a finite set. Any function $\pi : U \to V$ defines a Euclidean bundle $\pi^* Y = \coprod_{u \in U} Y_{\pi(u)}$, and a function $\pi^* : \prod_{v \in V} Y_v \to \prod_{u \in U} Y_{\pi(u)}$, called a pullback of sections, $\pi^* : s \mapsto \left(s_{\pi(u)}\right)_{u \in U}$*

We will use pullbacks to describe how to "make copies" of a space with a function $\pi : U \to V$. This is illustrated in Fig. 2, where from a bundle $Y$ we construct a new bundle having a varying number of copies of each space. $\pi^*$ also transports a choice $s \in \prod_{v \in V} Y_v$ to $\prod_{u \in U} Y_{\pi(u)}$ by "copying over".

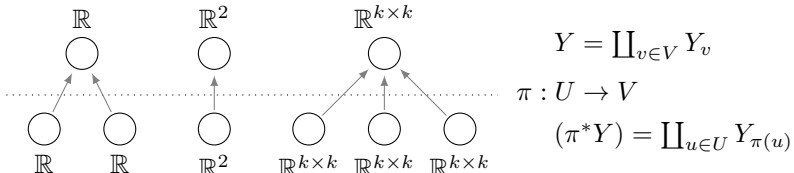

$$Y = \coprod_{v \in V} Y_v$$
$$\pi : U \to V$$
$$(\pi^* Y) = \coprod_{u \in U} Y_{\pi(u)}$$

Figure 2: Illustration of the construction of a pullback bundle, with $\pi : U \to V$ indicated by arrows. The top row is a set $V$ with 3 elements pictured as circles, and bottom row is $U$ with 6 elements. The vector spaces attached to each element are depicted next to each corresponding circle.

### 3.1 GENERALIZING MULTI-LAYER PERCEPTRONS TO ARBITRARY COMPUTATION GRAPHS

We use the definition of a (finite directed) graph $G = (V, E)$ as a finite set $V$ and a set $E \subseteq V \times V$. We write $G(-, v)$ the parents of $v \in V$ in $G$, and $G(v, -)$ its children, see Appendix A for a summary of notations and Appendix C for details on graphs. With the idea of describing neural networks, we take a computation graph whose vertices indicate variables in a program, and specify a set of "initial" vertices $I$ corresponding to inputs, and "terminal" vertices $T$ corresponding to outputs. Then, attach to each vertex $v$ a vector space $\mathcal{Y}_v$ of "activations at $v$", and to each edge $(u, v)$ a vector space $\mathcal{W}_{(u,v)}$ of "weights" of the connection from $u$ to $v$. For more generality, we also use a vector space $\mathcal{Z}_{(u,v)}$ of pre-activations to represent the intermediary variables before the non-linearity $\sigma_v$ associated to $v$.

---

[1]The nomenclature is that of algebraic topology, where these objects are common, but no prior knowledge is assumed. The names *bundle* and *section* serve only to distinguish products / co-products with words.

**Definition 3.3** (Perceptron module). *Let $B = (V, E)$ be a finite directed acyclic graph. A perceptron module over $B$ is a tuple $\mathbb{B} = ((I, T), (\mathcal{Y}, \mathcal{Z}, \mathcal{W}), (M, \sigma))$ composed of*

- *A subset $I \subseteq V$ such that $v \in I \Rightarrow B(-, v) = \varnothing$, and a subset $T \subseteq V$.*
- *Three Euclidean bundles, $\mathcal{Y} = \coprod_{v \in V} \mathcal{Y}_v$, and $\mathcal{W} = \coprod_{e \in E} \mathcal{W}_e$, and $\mathcal{Z} = \coprod_{e \in E} \mathcal{Z}_e$*
- *A family of functions $(M_e)_{e \in E}$ such that $M_{(u,v)} : \mathcal{W}_{(u,v)} \times \mathcal{Y}_u \to \mathcal{Z}_{(u,v)}$*
- *A family of functions $(\sigma_v)_{v \in V \setminus I}$ such that $\sigma_v : \prod_{u \in B(-,v)} \mathcal{Z}_{(u,v)} \to \mathcal{Y}_v$*

*We write $\mathrm{Param}(\mathbb{B}) = \prod_{e \in E} \mathcal{W}_e$, and $\mathrm{PMod}(B)$ the set of perceptron modules over $B$.*

The definition of perceptron modules (and later their lifts in Def. 3.5) is not restricted to multi-layer perceptrons, and is chosen very general precisely to tackle advanced computation graphs (e.g. Fig. 6). In most simple instances, the $M$ map would be a regular multiplication $(a, b) \in \mathbb{R} \times \mathbb{R} \mapsto ab \in \mathbb{R}$, and the $\sigma$ map an addition and non-linearity, e.g. $\sigma : z \in \mathbb{R}^k \mapsto \tanh\left(\sum_i z_i\right)$, summing over parents.

**Definition 3.4** (Activation map, or "forward" function, of a perceptron module). *Let $B = (V, E)$. A perceptron module $\mathbb{B} = ((I, T), (\mathcal{Y}, \mathcal{Z}, \mathcal{W}), (M, \sigma)) \in \mathrm{PMod}(B)$ comes equipped with a map*

$$\mathrm{F}[\mathbb{B}] : \prod_{e \in E} \mathcal{W}_e \times \prod_{v \in I} \mathcal{Y}_v \to \prod_{v \in V} \mathcal{Y}_v$$

*with $(\mathrm{F}[\mathbb{B}] : (w, \Phi) \mapsto f)$ defined inductively over $v \in V$ by $f_v = \Phi_v$ if $v \in I$, and for $v \notin I$ as*

$$f_v = \sigma_v \left( \left( M_{(u,v)} \left( w_{(u,v)}, f_u \right) \right)_{u \in B(-v)} \right)$$

In general, to be usable, it is also mandatory to describe how the inputs are connected. For instance in Fig. 4b with a function $c : I \to \{r, g, b\}$ to ensure $x \mapsto \mathrm{F}[\mathbb{B}](w, c^* x)$ has inputs in $\mathbb{R}^{\{r,g,b\}}$, not $\mathbb{R}^I$.

With the Euclidean bundles $\mathcal{W} = E \times \mathbb{R}$ and $\mathcal{Y} = V \times \mathbb{R}$, the restriction to $T$ of this function $(w, \Phi) \mapsto (\mathrm{F}[\mathbb{B}](w, \Phi))|_T$ has type $\mathbb{R}^m \times \mathbb{R}^d \to \mathbb{R}^k$, where $m = \#E \in \mathbb{N}$ is the number of parameters, $d = \#I \in \mathbb{N}$ the number of inputs, and $k = \#T \in \mathbb{N}$ the number of outputs. The notation with products will become more interesting when the dimension of weights are more unusual.

**Blueprint notation.** Instead of depicting the acyclic graph $B$, with lots of annotations, we use the following graphical convention. We draw vertices as circles, edges as squares, with horizontal arrows to indicate vertex-edge incidence, and vertical arrows from small circles to indicate weights. We annotate each arrow with a vector space ($\mathcal{W}$ from a small circle to a square, $\mathcal{Y}$ from a circle to a square, $\mathcal{Z}$ from a square to a circle), and each shape (circle or square) — except initial vertices — with a function (respectively $\sigma$ and $M$) whose domain and codomain matches the spaces defined by the incoming and outgoing arrows, cf. Figure 3. We depict terminal (resp. initial) vertices with a continuous (resp. dashed) double circle.

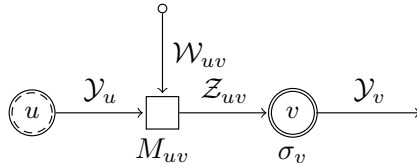

Figure 3: Generic blueprint notation

### 3.2 Lifting perceptron modules through homomorphisms

This theory describes complicated functions by constructing a "base" perceptron module $\mathbb{B}$, and then applying a "lift" operation (Def. 3.5) to build a more powerful module $G$, impossible to draw due to its large size. The intuition for how to do so is depicted in Fig. 4 with the example of the multi-layer perceptron (MLP), "replicating" vertices vertically to increase the "width". Fig. 4a depicts a perceptron module corresponding to an MLP architecture with two hidden layers, with activation bundle $\mathcal{Y} = V \times \mathbb{R}$ and weights $\mathcal{W} = E \times \mathbb{R}$, parent summation and non-linearity $\sigma : \mathbb{R} \to \mathbb{R}$, and maps $M_{(u,v)} : \mathbb{R} \times \mathbb{R} \to \mathbb{R}$ the multiplication $(a, b) \mapsto ab$. No matrices appear on this blueprint. Specifying this module, a "width" for each layer, and the connection of inputs, uniquely defines the MLP. Thus we annotate each vertex with an integer in Fig. 4a, to specify the width of the layer in Fig. 4b. The parameter space after lifting becomes $\mathbb{R}^{3 \times 5} \times \mathbb{R}^{5 \times 6} \times \mathbb{R}^{6 \times 4}$. Note how the weight matrices $W_{(0,1)} \in \mathbb{R}^{3 \times 5}$, $W_{(1,2)} \in \mathbb{R}^{5 \times 6}$ and $W_{(2,3)} \in \mathbb{R}^{6 \times 4}$ arise from the lift. Similarly, for a convolution operation for which we would use this lift operation to construct more channels, we expect a weight space $\mathcal{W}_{(u,v)} = V \approx \mathbb{R}^k$ to give rise to tensors $W_{(u,v)} \in V^{n_u \times n_v} \approx \mathbb{R}^{k \times n_u \times n_v}$, corresponding to $n_u$ input-channels and $n_v$ output-channels for that convolution.

We will give a general definition of (sparse) lifts first, the fully-connected example is recovered as a subcase later. Intuitively, from the same base module of Fig. 4a, we want to construct the MLP with underlying graph $G$ (from Figure 5a) as a perceptron module. This is done by "copying" nodes to construct a layer of hidden nodes in $G$ from the single node in $B$ that is copied over. How to perform this copy (i.e. how many times to copy every node and how to connect the copied nodes) is specified by $G$ and a homomorphism $\pi : G \to B$. The vertex map $\pi : V_G \to V_B$ indicates the number of copies ($v \in V_B$ yields nodes $\pi^{-1}(v) \subseteq V_G$), and the connectivity pattern is indicated by the structure of the graph $G$ (which may be sparse) and the induced edge map $\pi : E_G \to E_B$. This coincides with the transformation depicted

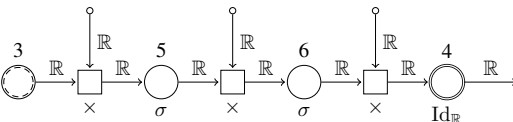

(a) Blueprint notation of the base module $\mathbb{B}$ and lift annotation $n : V_B \to \mathbb{N}$ above vertices (see Definition 3.6) for MLPs

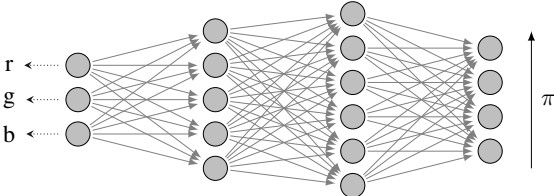

(b) Graph notation of the fully-connected lift G, the homomorphism $\pi : G \to B$ is given by vertical projection to (a)

Figure 4: MLP representing functions $\mathbb{R}^{\{r,g,b\}} \to \mathbb{R}^4$

in Fig. 4 (dense) and Fig. 5a (sparse). The condition that $\pi$ is a homomorphism of graphs ensures that this construction is well-defined. When there are many incoming connections to a vertex $v \in E_G$, mapped by $\pi$ to a single edge $e \in E_B$, such as in Fig. 4b and Fig. 5, the corresponding pre-activation values $z_{(u,v)}$ for $(u,v) \in \pi^{-1}(e)$ are summed before the computation of the non-linearity.

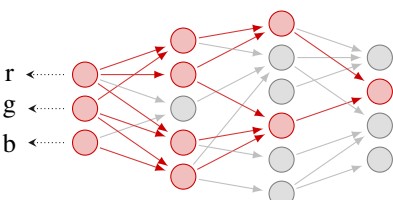

(a) Sparse lift similar to the multi-layer perceptron.

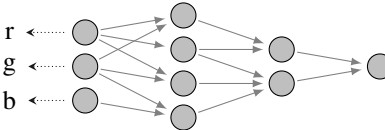

(b) Smaller sparse lift with the same base module $\mathbb{B}$ as above. This smaller network (b) is "contained" (*strongly*, extracted by fibration, in the vocabulary of Section 4) in the larger network (a). The inclusion is denoted by a (dark) red coloring of the nodes in (a).

Figure 5: Sparse lift graphs, denoted $G$ in the more general Def. 3.5, not covered by Def. 3.6.

**Definition 3.5** (Lift of perceptron modules). *Let $G = (V_G, E_G)$ and $B = (V_B, E_B)$ be graphs. A homomorphism of graphs $\pi : G \to B$ defines a function $\pi^* : \mathrm{PMod}(B) \to \mathrm{PMod}(G)$ called "lift",*

$$\pi^* : \Big( (I,T), (\mathcal{Y}, \mathcal{Z}, \mathcal{W}), (M,\sigma) \Big) \mapsto \Big( (\pi^{-1}(I), \pi^{-1}(T)), (\pi^*\mathcal{Y}, \pi^*\mathcal{Z}, \pi^*\mathcal{W}), (\bar{M}, \bar{\sigma}) \Big)$$

*where for $e \in E_G$, $\bar{M}_e = M_{\pi(e)}$, and for $v \in V_G \setminus \pi^{-1}(I)$, $\bar{\sigma}_v : \prod_{u \in G(-,v)} \mathcal{Z}_{\pi(u,v)} \to \mathcal{Y}_{\pi(v)}$ is*

$$\bar{\sigma}_v : z \mapsto \sigma_{\pi(v)} \left( \left( \sum_{u \in \pi^{-1}(a) \cap G(-,v)} z_{(u,v)} \right)_{a \in B(-,\pi(v))} \right)$$

*Let $\mathcal{C}$ be a finite set and $p_{\mathcal{C}} : \mathcal{C} \to I$. Define $\mathrm{LiftPMod}(\mathbb{B}, \mathcal{C})$ as the set of tuples $(G, \pi, c)$ where $\pi : G \to B$ is a graph homomorphism, and $c : \pi^{-1}(I) \to \mathcal{C}$ an injection such that $p_{\mathcal{C}} \circ c = \pi|_{\pi^{-1}(I)}$.*

*We extend the notation of forward functions to $\mathrm{G} = (G, \pi, c) \in \mathrm{LiftPMod}(\mathbb{B}, \mathcal{C})$ by defining the function $\mathrm{F}[\mathrm{G}] : \mathrm{Param}(\mathrm{G}) \times \prod_{u \in \mathcal{C}} \mathcal{Y}_{p_{\mathcal{C}}(u)} \to \prod_{v \in V_G} \mathcal{Y}_v$ as $\mathrm{F}[\mathrm{G}] : (w, x) \mapsto \mathrm{F}[\pi^*\mathbb{B}](w, c^*x)$.*

Recall that the bundle $\mathcal{Y}$ is equivalent to a family $(\mathcal{Y}_v)_{v \in V_B}$ of Euclidean spaces, so the the pullback $\pi^*\mathcal{Y} = \coprod_{v \in V_G} \mathcal{Y}_{\pi(v)}$ is equivalent to a family $((\pi^*\mathcal{Y})_v = \mathcal{Y}_{\pi(v)})_{v \in V_G}$, consistent with the intuition of "copying" activation spaces when replicating nodes (and similarly for $\mathcal{Z}$ and $\mathcal{W}$ indexed by edges). Validity of this definition (e.g. well-definition of the $z$-sum) is easy to check, details in Appendix K.1. Our later convergence theorem will only apply to particular sparse lifts, but most common deep learning architectures use a lift which is maximally connected, yielding dense weight tensors.

**Definition 3.6** (Fully-connected lift)**.** *Let $B = (V_B, E_B)$ be a directed acyclic graph.*
*Let $n : V_B \to \mathbb{N}$ be a function, called the "lifting dimension". Define the graph $C = (V_C, E_C)$ as*

$$V_C = \{ (b, i) \mid b \in V_B, i \in [n_b] \} \quad and \quad E_C = \{ (u, v) \in V_C \times V_C \mid (\pi(u), \pi(v)) \in E_B \}$$

*with $\pi : V_C \to V_B$ projection to the first coordinate. We call $C$ the fully-connected lift of $B$ along $n$.*

*Similarly if $\mathbb{B} \in \mathrm{PMod}(B)$, we call $\pi^*\mathbb{B} \in \mathrm{PMod}(C)$ the fully-connected lift of $\mathbb{B}$ along $n$.*
*If $\mathcal{C} = \coprod_{b \in I_B} [n_b]$ (i.e. an order of inputs has been chosen), then $(C, \pi, \mathrm{Id}_\mathcal{C}) \in \mathrm{LiftPMod}(\mathbb{B}, \mathcal{C})$.*

We can check that the forward function of the lifted module from Fig. 4 is the usual perceptron forward. Indeed, for a vertex $v \in V_B \setminus I_B$, and $i \in [n_v]$, we get the activation $f_{(v,i)} \in \mathcal{Y}_v = \mathbb{R}$ as

$$f_{(v,i)} = \sigma_v \left( \sum_{(u,j) \in C(-,(v,i))} M_{(u,v)} \left( w_{(u,j),(v,i)}, f_{(u,j)} \right) \right) = \sigma_v \left( \sum_{j \in [n_u]} f_{(u,j)} \, w_{(u,j),(v,i)} \right)$$

with $u$ the unique parent of $v$ in $B$ in the last part. In matrix notation, $f_v = \sigma_v \left( f_u \cdot W_{(u,v)} \right) \in \mathbb{R}^{n_v}$

**A more interesting example.** Using the operation $\mathrm{AddSoftMul} : (\mathbb{R}^n \times \mathbb{R}^n) \times \mathbb{R}^{n \times n} \to \mathbb{R}^n$, $((x, y), A) \mapsto x + \mathrm{softmax}(A) \cdot y$, where the softmax is taken over each row (and potentially masked), we recover modules similar to the transformer blocks of Vaswani et al. (2017) (see Figure 6).

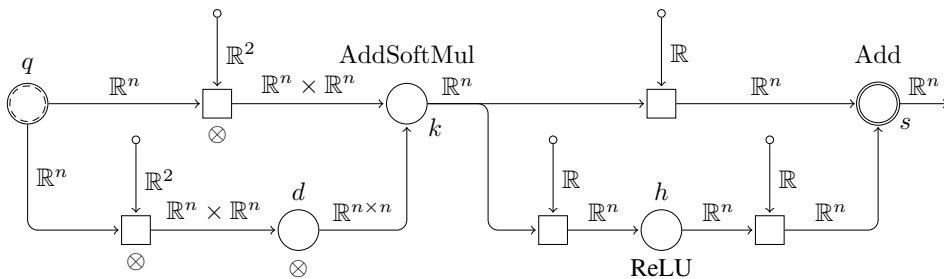

Figure 6: $\mathrm{MixingTransformer} : X \in \mathbb{R}^{n \times q} \mapsto Y \in \mathbb{R}^{n \times s}$, weights (top-to-bottom, left-to-right): $(M_A, W_V) \in \mathbb{R}^{q \times k} \times \mathbb{R}^{q \times k}$, $M_B \in \mathbb{R}^{k \times s}$, $(W_Q, W_K) \in \mathbb{R}^{q \times d} \times \mathbb{R}^{q \times d}$, $H_0 \in \mathbb{R}^{k \times h}$, $H_1 \in \mathbb{R}^{h \times s}$,
$$Z = X \cdot M_A + \mathrm{softmax}\left( (X \cdot W_Q) \cdot (X \cdot W_K)^T \right) \cdot (X \cdot W_V), \quad Y = Z \cdot M_B + \sigma(Z \cdot H_0) \cdot H_1$$
The first block $(X \mapsto Z)$ is a usual self-attention when $q = k$ and the mixing matrix is $M_A = I_q$, the second block $(Z \mapsto Y)$ is a residual MLP block when $k = s$ and the mixing matrix is $M_B = I_k$.

## 4  RANDOM SPARSE LIFTS AND CONVERGENCE OF SPARSE PERCEPTRONS

We start by reviewing the proof ideas for the convergence theorem, to give intuition for the definitions.

**Split setup & easy recombination.** To analyse a large neural network with linear last layer, we split it into a composition, first of a perceptron module computing deep features (whose number will scale) followed by a linear recombination function (whose input dimension will scale, but output dimension remains fixed). With fixed features, learning the last layer is easy by convexity, in particular if "optimal features" are present then the global optimum of the last layer must have optimal loss.

**Coverage property.** If the perceptron module representing the features is sufficiently large, it should "contain" a sub-network computing near-optimal features. The usual way to understand this term for a directed graph $G = (V_G, E_G)$ is to select a "subgraph", which is a graph $S = (V_S, E_S)$ such that $V_S \subseteq V_G$ and $E_S \subseteq E_G$. This is for instance the way such a "sub-network" is understood in the lottery ticket hypothesis (Frankle & Carbin, 2019). In the language of graph theory, this corresponds to an extraction by *homomorphism* (because the inclusion $\iota : V_S \to V_G$ induces a homomorphism of graphs from $S$ to $G$). These "sub-networks" are in general very different from the larger network, and do not necessarily compute the same functions (it is the lottery ticket hypothesis that they actually do learn similar functions, by a yet-unexplained mechanism). There is in graph theory (and algebraic topology more generally) a different way to extract what we could call a "strong sub-graph", by

*fibration* instead of *homomorphism*. The inclusion map $\iota : V_S \to V_G$ is a fibration of graphs if the following two conditions hold: first, if $(u, v) \in V_S \times V_S$ and $(u, v) \in E_G$ then $(u, v) \in E_S$ (no "removal of edges"); also, if $v \in V_S$, then $G(-, v) \subseteq V_S$ (no "removal of parents"). Intuitively, if $S$ and $G$ are viewed as computation graphs, then removing an incoming edge or removing a parent will change the computation, while removing a child does not. Thus if $S$ is "strong sub-graph" of $G$, extracted by *fibration*, we can extract the subgraph then perform a computation in the subgraph and isolate an activation, or we could equivalently perform the computation in the larger graph and then extract the activations associated to the subgraph and isolate the activation of the target vertex. When multi-layer perceptrons are constructed with dense linear layers, a network of width $2n$ does not contain (in the sense of "strong sub-graphs" extracted by *fibration*) a sub-network of width $n$. This is because the number of terms in each sum is $2n$ in the large nework, one for each parent of the node considered, which is fully connected to every node of the previous layer. However, fibrations preserve the number of parents, and the smaller network has $n$ terms in each sum, again one for each parent, but there are only $n$ such parents at width $n$. Nonetheless, we can easily construct networks that do have this property of containing smaller sub-networks by construction, by simply sparsifying the dense layers, similarly to a dropout masking operation that would be performed at initialization and whose mask is held constant throughout training. In these sparse networks, increasing model size cannot hurt (provably) because a large network "contains" (strongly) a small sub-network that is computing the same thing as a smaller model. The coverage property that we need is only very slightly stronger: there needs to be a certain fraction (or "volume") of sub-networks that are near-optimal, not just one, the precise details are in the formal statements of Appendix H.

**Local stability of coverage.** The existence of sub-networks computing interesting features at initialization is insufficient for a convergence proof, because we intend to train the weights of the large network, and thus we expect the weights to move and have no guarantee that the sub-network will continue to perform a similar computation throughout training. The last ingredient we require is that the large network has a low "outgoing degree" (i.e. each vertex in the directed graph has few children), such that there is no vertex whose activation is re-used too much by later vertices, no single point of failure that could perturb the entire network if its weights move slightly too much in the wrong direction. Coupled with a covering in volume, this low-degree property implies that there is an entire region of parameters around initialization where we can guarantee that there is a sub-network with near-optimal features, rather than only at the single point of initialization. This implies that at initialization of a very large sparse network, we expect a large number of sub-networks to be near-optimal, and throughout training the number of near-optimal sub-networks can only decrease slowly (due to the low outgoing degree): if a sub-network is perturbed enough to no longer be close to an optimal feature, then the last layer can simply switch the associated weight of this sub-network to a similar but un-perturbed sub-network to recover optimality (acceptable due to easy recombination).

## 4.1 Random sparse lifts of perceptrons modules

From now on, we will only ever consider a single base module with various lifts. Therefore, let $B = (V_B, E_B)$ be a finite directed acyclic graph. Let $\mathbb{B} \in \mathrm{PMod}(B)$ be a perceptron module, and write its components $\mathbb{B} = ((I_B, T_B), (\mathcal{Y}, \mathcal{Z}, \mathcal{W}), (M, \sigma))$. We will also keep fixed the number of lifted inputs (comparing modules with different inputs is not very useful anyway). Let $n^0 : I_B \to \mathbb{N}$, and $\mathcal{C} = \coprod_{b \in I_B} [n_b^0]$ (without loss of generality on a finite set $\mathcal{C} = \coprod_{b \in I_B} \mathcal{C}_b$ after ordering $\mathcal{C}_b \approx [n_b^0]$).

The first ingredient needed is the presence of a linear last layer with fixed output dimension even when the number of nodes in the perceptron module computing the "deep features" tends to infinity.

**Definition 4.1** (Perceptron with linear readout). *Let* $\mathrm{G} = (G, \pi, c) \in \mathrm{LiftPMod}(\mathbb{B}, \mathcal{C})$, *and* $k \in \mathbb{N}$. *Let* $\mathrm{LinReadout}(\mathrm{G}, k) = \prod_{v \in T(\mathrm{G})} \mathcal{Y}_{\pi(v)}^k$ *with* $\mathrm{LinReadout}(\mathrm{G}, k) \times \prod_{v \in V(\mathrm{G})} \mathcal{Y}_{\pi(v)} \to \mathbb{R}^k$

$$(a, x) \mapsto a \cdot x = \left[ \sum_{b \in T_B} \sqrt{\frac{1}{\#\pi^{-1}(b)}} \sum_{v \in \pi^{-1}(b)} \langle a_{i,v}, x_v \rangle_{\mathcal{Y}_b} \right]_{i \in [k]}$$

We can now describe the sparsification procedure. Note that the following definition is equivalent to using a fully-connected lift and a random dropout mask $m$, drawn at initialization and fixed during training. We expect the probability distribution chosen on weights $\mathcal{W}_e$ to be a normal distribution in all applications (thus write it $\mathcal{N}$), but we will only require the assumption that it has full support.

**Definition 4.2** (Random sparse lift). *Let $n : V_B \to \mathbb{N}$ be such that $n_b = n_b^0$ for all $b \in I_B$. Let $\lambda : E_B \to \mathbb{R}_+$, and for every $e \in E_B$, let $\mathcal{N}_e$ be a probability distribution on $\mathcal{W}_e$.*

*Let $C = (V_C, E_C)$ be the fully-connected lift of $B$ along $n$, with homomorphism $\pi : C \to B$. Let $(m : E_C \to \{0,1\}, w : E_C \to \pi^*\mathcal{W})$ be independent random variables with distributions $m_{(u,v)} \sim \mathrm{Bern}(\lambda_{\pi(u,v)}/n_{\pi(u)})$ Bernoulli for any $(u,v) \in E_C$, and with $w_e \sim \mathcal{N}_{\pi(e)}$ for $e \in E_C$.*

*The random sparse lift of $\mathbb{B}$ along $(n, \lambda)$ with distribution $\mathcal{N}$ is the tuple $(G, w|_E)$, where $G = (V, E)$ is the random graph with $V = V_C$, $E = \{e \in E_C \mid m_e = 1\}$, $G$ is the lifted perceptron module $G = (G, \pi, \mathrm{Id}_C) \in \mathrm{LiftPMod}(\mathbb{B}, \mathcal{C})$ and $w|_E \in \mathrm{Param}(G)$.*

For an architecture for which universal approximation has been established (i.e. the approximation "number" $h_0(f^\star, \varepsilon)$ is finite), we show that we can construct random sparse lifts for which the learning "number" $h_1(f^\star, \varepsilon, \delta)$ is also finite for failure probabilities $\delta > 0$ arbitrarily close to zero. However, in general this "number" is not an integer but an element of $\mathbb{N}^k$, where $k \in \mathbb{N}$ is the number of scaling dimensions (one per vertex of $B$). We refer to the network approximating the target function as the *witness network*, since it is a witness for the universal approximation theorem at a given precision.

## 4.2 Convergence of sparse perceptrons

Let $\mathcal{X} \subseteq \prod_{u \in \mathcal{C}} \mathcal{Y}_{p_C(u)}$ be compact, $\mathcal{D}$ a distribution on $\mathcal{X}$, and let $f^\star : \mathcal{X} \to \mathbb{R}^k$ with $\|f^\star\|_{\mathcal{D}}^2 < \infty$. For any $G \in \mathrm{LiftPMod}(\mathbb{B}, \mathcal{C})$, define the loss $\mathcal{L}[G] : \mathrm{Param}(G) \times \mathrm{LinReadout}(G, k) \to \mathbb{R}_+$ as

$$\mathcal{L}[G] : (w, a) \mapsto \mathbb{E}_{x \sim \mathcal{D}} \left[ \| a \cdot F[G](w, x) - f^\star(x) \|_2^2 \right]$$

Let $\lambda : E_B \to \mathbb{R}_+^*$. For $e \in E_B$, let $\mathcal{N}_e$ be a full-support distribution on $\mathcal{W}_e$. Additionally, define $\mathcal{S}_0 = \{n : V_B \to \mathbb{N} \mid \forall b \in I_B, n_b = n_b^0, \forall (a, b) \in E_B, n_b \geq n_a \log n_a\}$, where $n_b^0 = \#p_C^{-1}(b)$. We assume $\mathbb{B}$ has $(M, \sigma)$ maps that are continuously differentiable, to ensure gradients exist and have finite norm. Finally, we assume that for $(a, b) \in E_B$ if $a \in I_B$ then $\lambda_{(a,b)} \leq \min\{n_a^0/2, (n_a^0/3)^{1/2}\}$.

**Theorem 4.3** (Probable approximate correctness of random sparse lifts under gradient flow).
*Let $(\varepsilon, \delta) \in \mathbb{R}_+^* \times ]0, 1]$. If there exists $G^\star \in \mathrm{LiftPMod}(\mathbb{B}, \mathcal{C})$ a lifted perceptron module, and parameters $(w^\star, a^\star) \in \mathrm{Param}(G^\star) \times \mathrm{LinReadout}(G^\star, k)$ such that $\mathcal{L}[G^\star](w^\star, a^\star) < \varepsilon$, then there exists $\kappa \in \mathbb{R}_+^*$ and $N_1 : V_B \to \mathbb{N}$ (a size threshold) such that the following proposition holds:*

*For all $n \in \mathcal{S}_0$ such that $n \succeq N_1$, with probability at least $(1 - \delta)$ over $(G, w)$ a random sparse lift of $\mathbb{B}$ along $(n, \lambda)$ with distribution $\mathcal{N}$, and $a = 0 \in \mathrm{LinReadout}(G, k)$, the pair $(\mathcal{L}[G], (w, a))$ satisfies Convergence Criterion 1 with limit error $\varepsilon$ and constant $\kappa$.*

We give in Appendix J a quantitative version with bounds on $N_1$ and $\kappa$ as a function of the witness network structure and choice of parameters, allowing for future quantitative research in this direction. Note that this is immediately extended to a loss $f \mapsto \mathbb{E}_{(x,y)}[\|f(x) - y\|_2^2]$ with $\mathbb{E}_{(x,y)}[\|y\|_2^2] < \infty$. Indeed, that is equal to $\|f - f^\star\|_{\mathcal{D}}^2 + \mathbb{E}[\mathbb{V}[Y|X]]$ for $f^\star : x \mapsto \mathbb{E}_{(X,Y) \sim \mathcal{D}}[Y|X = x]$. Moreover, note that this theorem holds for any $\varepsilon \in \mathbb{R}_+^*$ and not only near zero. This statement thus truly matches what it means to converge to the infimum of the $\mathrm{LiftPMod}(\mathbb{B}, \mathcal{C})$ class with large lifts, which should prove most interesting when that architecture is restricted to respect certain symmetries by construction.

We believe that the conditions here are interesting because they are easy to verify. (1) The condition that the architecture is obtained as a random sparse lift can be satisfied by construction with little restriction on architecture search. (2) The condition that any lift (sparse or dense) with low loss exists can be obtained by universal approximation, or empirically because any network with low loss can serve as witness. Proving or disproving a similar result for dense lifts is left as future work.

We modeled dynamics in continuous-time, and considered only convergence in loss. Convergence of parameters will require more assumptions, see e.g. Patel et al. (2022, Section 2). Discrete time results will also require more assumptions, but are often obtained by extending continous-time results, see e.g. Even et al. (2023, Theorem 1). We did not assume that $\mathcal{D}$ is finitely supported, to facilitate extensions in online learning with test-loss bounds from stochastic gradient estimates on finite samples.

**Conclusion.** We have shown PAC-convergence of large random sparse lifts under gradient flow to the infimum over all lifts, by introducing a strong formalism to define a large class of neural networks and the tools to track their activations across training to show tangent approximation properties for a long enough time. We believe that this direction constitutes a promising route to a strong convergence theory for deep learning, with a principled architecture design and testable empirical predictions.

## ACKNOWLEDGEMENTS

The authors acknowledge support from the French government under management of the Agence Nationale de la Recherche, reference ANR-19-P3IA-0001 (PRAIRIE 3IA Institute).

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

# A SUMMARY OF NOTATIONS

Table 1: Summary of principal notations used in this document

| | |
|---|---|
| $B = (\mathrm{V}_B, \mathrm{E}_B)$ | The base graph, for the base perceptron module |
| $G = (\mathrm{V}_G, \mathrm{E}_G)$ | The lifted graph when there is only one, the graph of the random sparse lift in the last theorem |
| $G^\star = (\mathrm{V}^\star, \mathrm{E}^\star)$ | The graph of the *witness network*, the lifted perceptron module achieving the loss threshold. |
| $G(-, v) = \{u \in \mathrm{V}_G \mid (u, v) \in \mathrm{E}_G\}$ for $v \in \mathrm{V}_G$ | The parents (direct ancestors) of vertex $v$ in $G$. |
| $G(v, -) = \{u \in \mathrm{V}_G \mid (v, u) \in \mathrm{E}_G\}$ for $v \in \mathrm{V}_G$ | The children (direct descendants) of $v$ in $G$. |
| $\mathbb{B} = ((\mathrm{I}_B, \mathrm{T}_B), (\mathcal{Y}, \mathcal{Z}, \mathcal{W}), (M, \sigma))$ | The base perceptron module (Definition 3.3). The functions $(M, \sigma)$ are continuously differentiable. |
| $\mathcal{C} = \coprod_{b \in \mathrm{I}_B} \mathcal{C}_b \quad \text{or} \quad p_\mathcal{C} : \mathcal{C} \to \mathrm{I}_B$ | The "meaningful names" of input connections. This is "simplified" to $\mathcal{C}_b = \{0, \dots \#\mathcal{C}_b - 1\}$ in the proof without loss of generality, but leaves room for easier implementations. For instance, for single input type $\mathrm{I}_B = \{\mathrm{img}\}$, lifted to three channels (one per RGB color), meaningful names can be $\mathcal{C}_{\mathrm{img}} = \{r, g, b\}$, instead of $\{0, 1, 2\}$. |
| $\mathcal{X} \subseteq \prod_{b \in \mathrm{I}_B} \prod_{c \in \mathcal{C}_b} \mathcal{Y}_b$ | A compact subset of inputs accessed "by name". With two input types $\mathrm{I}_B = \{\mathrm{img}, \mathrm{text}\}$, lifted as above for the image and to $T \in \mathbb{N}$ tokens for the text with $\mathcal{C}_{\mathrm{text}} = [T]$, then $\prod_{b \in \mathrm{I}_B} \prod_{c \in \mathcal{C}_b} \mathcal{Y}_b = \mathcal{Y}_{\mathrm{img}}^{\{r,g,b\}} \times \mathcal{Y}_{\mathrm{text}}^T$. The spaces $\mathcal{Y}_b$ can be distinct, e.g. $\mathcal{Y}_{\mathrm{img}} = \mathbb{R}^{256 \times 256}$ and $\mathcal{Y}_{\mathrm{text}} = \mathbb{R}^k$ with $k \in \mathbb{N}$ an arbitrary embedding dimension. |
| $(G, \pi, c) \in \mathrm{LiftPMod}(\mathbb{B}, \mathcal{C})$ | A lifted perceptron module (Definition 3.5). The homomorphism $\pi : G \to B$ encodes the structure of the lift, the map $c$ provides the connection to inputs (e.g. which vertex is the $r$-channel, etc.) |
| $\mathrm{Param}(\mathbb{B}) = \prod_{e \in \mathrm{E}_B} \mathcal{W}_e$ | The parameter space of the base perceptron module $\mathbb{B} \in \mathrm{PMod}(B)$. There is one (multi-dimensional) weight per edge in the base module. All spaces $\mathcal{W}_e$ need not have the same dimension. |
| $\mathrm{Param}(\mathrm{G}) = \prod_{e \in \mathrm{E}_G} \mathcal{W}_{\pi(e)} = \prod_{e \in \mathrm{E}_B} \prod_{\substack{f \in \mathrm{E}_G \\ \pi(f)=e}} \mathcal{W}_e$ | The parameter space of a lifted perceptron module $\mathrm{G} = (G, \pi, c) \in \mathrm{LiftPMod}(\mathbb{B}, C)$. There is one copy of $\mathcal{W}_e$ per edge $f \in \mathrm{E}_G$ with $\pi(f) = e$. The number of edges will grow when scaling up, the dimension of the spaces $\mathcal{W}_e$ remain fixed. |
| $\mathrm{F}[\mathrm{G}] : \prod_{e \in \mathrm{E}_G} \mathcal{W}_{\pi(e)} \times \prod_{u \in \mathcal{C}} \mathcal{Y}_{p_\mathcal{C}(u)} \to \prod_{v \in \mathrm{V}_G} \mathcal{Y}_{\pi(v)}$ | Forward function of a lifted perceptron module $\mathrm{G} = (G, \pi, c) \in \mathrm{LiftPMod}(\mathbb{B}, \mathcal{C})$ over $G = (\mathrm{V}_G, \mathrm{E}_G)$ with homomorphism $\pi : G \to B$. |
| $\mathrm{LinReadout}(\mathrm{G}, k) = \prod_{v \in T(\mathrm{G})} \mathcal{Y}_{\pi(v)}^k$ | Linear readout of a perceptron module (i.e. linear last layer), see Definition 4.1. |

Throughout this paper, we use the french notation conventions, in particular $0 \in \mathbb{N}$, the open and half-open intervals are noted $]a, b[ = \{u \in \mathbb{R} \mid a < u < b\}$ and $[a, b[ = \{u \in \mathbb{R} \mid a \leq u < b\}$. We use liberally parenthesis or brackets to group factors in our proofs, for readability and without any particular distinction. We write "log" for the natural (or base-$e$) logarithm, i.e. $\forall u, \log \exp(u) = u$. For time-dependent quantities, we rarely distinguish between index notation and application, so for $\theta : \mathbb{R}_+ \to \mathbb{R}^d$ a function, we write $\theta_t = \theta(t)$ either in index or between parenthesis and use whichever is more readable depending on context (only exception is when both $\theta$ a function and $\theta_0$ a constant are defined at the same time, but we take both to coincide anyway). The norm notation $\|\cdot\|$ is used only in finite dimension and always indicates the Euclidean norm $\|v\|_V = \sqrt{\langle v, v \rangle_V}$, a.k.a. $\ell_2$-norm: $\|u\|^2 = \sum_i u_i^2$ in an orthonormal basis. On products of Euclidean spaces $V$ and $W$ the corresponding norm is the induced Euclidean norm on the product $\|(v, w)\|_{V \times W} = \sqrt{\|v\|_V^2 + \|w\|_W^2}$.

Table 2: Summary of advanced notations used mostly in the appendix

| | |
|---|---|
| $T(\mathrm{G}) = \pi^{-1}(\mathrm{T}_B)$ and $I(\mathrm{G}) = \pi^{-1}(\mathrm{I}_B)$ | The initial and terminal nodes of a lifted perceptron module $\mathrm{G} = (G, \pi, c) \in \mathrm{LiftPMod}(\mathbb{B}, \mathcal{C})$. |
| $V(\mathrm{G}) = \mathrm{V}_G$ and $E(\mathrm{G}) = \mathrm{E}_G$ | The vertices and edges of the underlying graph $G = (\mathrm{V}_G, \mathrm{E}_G)$ in a lifted perceptron module $\mathrm{G} = (G, \pi, c) \in \mathrm{LiftPMod}(\mathbb{B}, \mathcal{C})$. |
| $H = (\mathrm{V}_H, \mathrm{E}_H)$ | The other lifted graph when there are two. |
| $S = (\mathrm{V}_S, \mathrm{E}_S)$ | The subgraph of $G$ extracted by fibration. |
| $G_a(-, v) = \pi^{-1}(a) \cap G(-, v)$ | The "type-$a$" parents of $v \in \mathrm{V}_G$, with $a \in \mathrm{V}_B$ a vertex of the base graph, and $\pi : G \to B$ is the homomorphism indicating "types". In MLPs, the "type" is the index or depth of the corresponding layer. In more general perceptron modules, the notions of layer and depth cease to be well-defined, they are replaced by the pre-image by $\pi$ of a given vertex or edge in the base module. |
| $\pi^* \mathcal{Y} = \coprod\limits_{v \in \mathrm{V}_G} \mathcal{Y}_{\pi(v)}$ | Pullback bundle of activations for a lift $(G, \pi, c) \in \mathrm{LiftPMod}(\mathbb{B}, \mathcal{C})$, see Definition 3.2. |
| $\pi^* \mathcal{W} = \coprod\limits_{e \in \mathrm{E}_G} \mathcal{W}_{\pi(e)}, \quad \pi^* \mathcal{Z} = \coprod\limits_{e \in \mathrm{E}_G} \mathcal{Z}_{\pi(e)}$ | Pullback bundles for weights and pre-activations. |
| $(\pi^* w) \in \prod\limits_{e \in \mathrm{E}_G} \mathcal{W}_{\pi(e)}$ for $w \in \prod\limits_{b \in \mathrm{E}_B} \mathcal{W}_b$ | Pullback of sections (here, weights). This corresponds to an intuitive copy, i.e. $(\pi^* w)_e = w_{\pi(e)}$. |
| $\varphi^* : \prod_{h \in \mathrm{V}_H} \mathcal{Y}_h \to \prod_{g \in \mathrm{V}_G} \mathcal{Y}_{\varphi(g)}$ | Pullback of activations by a map $\varphi : \mathrm{V}_G \to \mathrm{V}_H$. |
| $(S, \varphi) : G \rightharpoonup H$ | The partial fibration from graph $G$ to graph $H$, i.e. a subgraph $S$ of $G$ such that the inclusion $\iota : S \to G$ is a fibration, together with a fibration $\varphi : S \to H$, see Definition H.1. |
| $(S, \varphi) : (G, \pi, c) \rightharpoonup (G^\star, \pi_\star, c_\star)$ | The $\mathrm{LiftPMod}(\mathbb{B}, \mathcal{C})$-morphism from the random sparse lift to the witness. See Definition F.1. |

## B ORGANISATION OF THE APPENDIX

Appendix C recalls the usual definitions for graphs and various types of morphisms between them. Appendix D studies the effect of the constants in the definition of Convergence Criterion 1 to give some early intuition. Appendix E gives the proof of the probably approximately correct convergence Theorem 2.1, showing how tangent approximation properties are sufficient to get convergence. Appendix F introduces the concept of morphisms of lifted perceptrons. Appendix G shows how to track activations even when weights can move, by leveraging the aforementionned morphisms. Appendix H shows how the tangent approximation property can be obtained from a covering property. Appendix I shows that the covering property previously defined is satisfied with high probability in random sparse lifts. Appendix J ties everything together in a quantitative version of Theorem 4.3. Appendix K provides all the more cumbersome proofs of well-definition of all concepts introduced in the document. Finally, Appendix L provides the proof of several small technical lemmas for bounding random variables that were used to alleviate the proofs of previous sections.

## C GRAPH DEFINITIONS AND LEMMAS

### C.1 GRAPH DEFINITIONS

We consider the language of graphs known to the reader, see Wilson (2010) for a thorough introduction otherwise. All graphs we consider are finite directed (see Wilson (2010, Chapter 7) more precisely) and acyclic. We use the following definitions in our proofs.

**Definition C.1** (Graph). *A (directed) graph is a couple $(V, E)$ of a finite set $V$ and a set $E \subseteq V \times V$.*

The set $V$ is called the set of "vertices", and $E$ the set of "edges". When $G = (V, E)$ is a graph, and $v \in V$, we write $G(v, -) = \{ u \in V \mid (v, u) \in E \}$ the set of out-neighbours of $v$ (also called *children*), and $G(-, v) = \{ u \in V \mid (u, v) \in E \}$ the set of in-neighbours of $v$ (also called *parents*).

**Definition C.2** (Homomorphism). *Let $G = (V_G, E_G)$ and $H = (V_H, E_H)$ be graphs. A function $\varphi : V_G \to V_H$ is a homomorphism of graphs if $(u, v) \in E_G \Rightarrow (\varphi(u), \varphi(v)) \in E_H$*

We write homomorphisms as $\varphi : G \to H$, by extending naturally the function of vertices to a function of edges $\varphi : E_G \to E_H, (u, v) \mapsto (\varphi(u), \varphi(v))$.

It is immediate from the definition that the composition of homomorphisms is also a homomorphism.

For our purposes, a directed graph $G = (V, E)$ is acyclic if there exists a total order $\prec$ on $V$ such that $(u, v) \in E \Rightarrow u \prec v$. This is equivalent to the definition of acyclicity as being "without cycles".

**Definition C.3** (Fibration). *A fibration from a graph $G = (\mathrm{V}, \mathrm{E})$ to a graph $H$ is a homomorphism $\varphi : G \to H$ such that for all $v \in \mathrm{V}$, the restriction $\varphi|_{G(-,v)} : G(-, v) \to H(-, \varphi(v))$ is bijective.*

In words, a fibration is a homomorphism that is also an isomorphism of in-neighbourhoods.

### C.2 FIBRATIONS ARE CLOSED UNDER COMPOSITION

**Proposition C.4.** *If $\varphi : G \to H$ and $\psi : H \to K$ are fibrations, then $\psi \circ \varphi : G \to K$ is a fibration.*

*Proof.* A fibration is a homomorphism that is an isomorphism of in-neighbourhoods. Therefore, since $\psi \circ \varphi$ is a homomorphism by composition, it suffices to show that it is an isomorphism of in-neighbourhoods. Write $G = (V_G, E_G)$, $H = (V_H, E_H)$ and $K = (V_K, E_K)$. Let $v_K \in V_K$, and $v_G \in V_G$ such that $(\psi \circ \varphi)(v_G) = v_K$. Write $v_H = \varphi(v_G) \in V_H$. By the fibration property of $\psi$, there exists a bijection $s : H(-, v_H) \to K(-, v_K)$ with $s = \psi|_{H(-,v_H)}$. By the fibration property of $\varphi$, there exists a bijection $t : G(-, v_G) \to H(-, v_H)$ with $t = \varphi|_{G(-,v_G)}$. The composition $s \circ t : G(-, v_G) \to K(-, v_K)$ is a bijection as a composition of bijections, it remains to check that it coincides with $(\psi \circ \varphi)$, which is immediate since $(s \circ t)(u) = s(t(u)) = s(\varphi(u)) = \psi(\varphi(u))$. □

# D CONVERGENCE CRITERION 1 : DEPENDENCY ON PARAMETERS

**Proposition D.1** (Convergence Criterion weakening).
*Let $\kappa_0 \in \mathbb{R}_+$ and $\varepsilon_0 \in \mathbb{R}_+$. Let $\kappa_1 \in \mathbb{R}_+$ and $\varepsilon_1 \in \mathbb{R}_+$ be such that $\kappa_1 \leq \kappa_0$ and $\varepsilon_1 \geq \varepsilon_0$.*

*If $(\mathcal{L} : \Theta \to \mathbb{R}_+, \theta_0 \in \Theta)$ satisfies Convergence Criterion 1 with limit error $\varepsilon_0$ and constant $\kappa_0$, then $(\mathcal{L}, \theta_0)$ also satisfies Convergence Criterion 1 with limit error $\varepsilon_1$ and constant $\kappa_1$.*

This is consistent with the intuition that in Convergence Criterion 1, the constant $\kappa$ is a form of "speed" (higher speed is more impressive) and $\varepsilon$ a form of "error" (lower error is more impressive).

*Proof of Proposition D.1.* For any $t \in \mathbb{R}_+$, let $S_t : \mathbb{R}_+ \times \mathbb{R}_+ \times \mathbb{R}_+^* \to \mathbb{R}_+$ be the function

$$S_t : (\kappa, \varepsilon, c) \mapsto \varepsilon + \frac{1}{\sqrt[3]{\kappa\, t + \frac{1}{c}}}$$

Let us show that $S_t$ is increasing with respect to $\varepsilon$, decreasing with respect to $\kappa$ and increasing with respect to $c$. Let $(\kappa, \varepsilon, c) \in \mathbb{R}_+^2 \times \mathbb{R}_+^*$. Compute the following derivatives and observe their sign

$$\partial_\varepsilon S_t(\kappa, \varepsilon, c) = +1 \qquad\qquad\qquad\qquad \geq 0$$

$$\partial_\kappa S_t(\kappa, \varepsilon, c) = -\frac{1}{3}\,(t)\,(\kappa\, t + 1/c)^{-4/3} \qquad \leq 0$$

$$\partial_c S_t(\kappa, \varepsilon, c) = -\frac{1}{3}\left(\frac{-1}{c^2}\right)(\kappa\, t + 1/c)^{-4/3} \quad \geq 0$$

Now to conclude, let $(\mathcal{L}, \theta_0)$ be a pair satisfying Convergence Criterion 1 with limit error $\varepsilon_0$ and constant $\kappa_0$. Let $\theta : \mathbb{R}_+ \to \Theta$ be such that $\theta(0) = \theta_0$ and $\partial_t \theta = -\nabla\mathcal{L}(\theta)$. Let $c_0 = \mathcal{L}(\theta_0)^3$ and $t \in \mathbb{R}_+$. By assumption, $\mathcal{L}(\theta_t) \leq S_t(\kappa_0, \varepsilon_0, c_0)$. By the study of variations above, $S_t(\kappa_0, \varepsilon_0, c_0) \leq S_t(\kappa_1, \varepsilon_0, c_0) \leq S_t(\kappa_1, \varepsilon_1, c_0)$. Thus for $t \in \mathbb{R}_+$, it holds $\mathcal{L}(\theta_t) \leq S_t(\kappa_1, \varepsilon_1, c_0)$, which is the definition of Convergence Criterion 1 with limit error $\varepsilon_1$ and constant $\kappa_1$, and concludes the proof. $\square$

*Bonus*: Note that we have also shown increase of $S_t$ with respect to $c$, so in later proofs it is sufficient to show $\mathcal{L}(\theta_t) \leq S_t(\kappa, \varepsilon, c)$ with an inequality $c \leq \mathcal{L}(\theta_0)^3$ rather than equality.

**Definition D.2** (Extended Convergence Criterion). *Let $\varepsilon \in \mathbb{R}_+$. The pair $(\mathcal{L} : \Theta \to \mathbb{R}_+, \theta_0 \in \Theta)$ satisfies the extended Convergence Criterion with limit error $\varepsilon$ and constant $\kappa = +\infty$ if for all $\theta : \mathbb{R}_+ \to \Theta$ such that $\theta(0) = \theta_0$, satisfying $\partial_t \theta = -\nabla\mathcal{L}(\theta)$, it holds $\forall t \in \mathbb{R}_+$, $\mathcal{L}(\theta_t) \leq \varepsilon$.*

It is immediately observed by the previous proposition — and the fact that $[S_t(\kappa, \varepsilon, c) \to \varepsilon]$ when $[\kappa \to +\infty]$ — that this statement is equivalent to "Convergence Criterion 1 is satisfied with constant $+\infty$ if it is satisfied with all finite constants $\kappa \in \mathbb{R}_+$". This is also consistent with the intuition that the statement of a flow reaching a loss $\varepsilon$ with "speed" $\kappa$ can be naturally extended to a statement with "infinite speed" if the flow just reaches loss $\varepsilon$ instantly. Because of this consistency with the previous criterion, we do not in the following distinguish this extension from Convergence Criterion 1.

# E  PROBABLE APPROXIMATE CORRECTNESS BY TANGENT APPROXIMATION

**Lemma E.1** (Convergence criterion by integration of a separable Kurdyka-Łojasiewicz inequality)**.**
*Let $\Theta$ be a vector space, $\mathcal{L} : \Theta \to \mathbb{R}_+$ a differentiable function, and $\theta_0 \in \Theta$. Let $\varepsilon \in \mathbb{R}_+^*$, $R \in \mathbb{R}_+$, $c \in \mathbb{R}_+^*$. If for all $\theta \in \Theta$ such that $\|\theta - \theta_0\| \leq R$, it holds $\|\nabla\mathcal{L}(\theta)\| \geq (\mathcal{L}(\theta) - \varepsilon)_+ / (\|\theta - \theta_0\| + c)$, Then the pair $(\mathcal{L}, \theta_0)$ satifies Convergence Criterion 1 with limit error $\varepsilon_0 = \tau \cdot \mathcal{L}(\theta_0) + (1-\tau) \cdot \varepsilon \in \mathbb{R}_+$ where $\tau = c/(R + c) \in \,]0, 1]$, and with constant $\kappa = 3\, c^{-2}(\mathcal{L}(\theta_0) - \varepsilon)_+^{-2} \in \mathbb{R}_+ \cup \{+\infty\}$,*

Since Convergence Criterion 1 uses a bound $\mathcal{L}(\theta_t) \leq \varepsilon_0 + 1/\sqrt[3]{\kappa\, t + 1/C_0}$ for which the right hand side is strictly decreasing with respect to $\kappa$, we can naturally extend its definition to $\kappa = +\infty$ if the convergence criterion is satisfied for all positive finite $\kappa$, which is also equivalent to simply a bound of $\forall t$, $\mathcal{L}(\theta_t) \leq \varepsilon_0$. Those are not the most interesting cases anyway, but this stresses the fact that this statement is well-defined in all edge cases. Note also that for fixed $c$, this statement has $\varepsilon_0 \xrightarrow[R \to \infty]{} \varepsilon$.

*Proof.* Let $\theta : \mathbb{R}_+ \to \Theta$ be a differentiable curve such that $\theta(0) = \theta_0$ and $\partial_t \theta = -\nabla\mathcal{L}(\theta)$. Note that if $\mathcal{L}(\theta_0) \leq \varepsilon$ then the conclusion is immediate because $\mathcal{L}(\theta_0) \leq \varepsilon_0$ and the loss is non-increasing since $\partial_t(\mathcal{L} \circ \theta)_t = \nabla\mathcal{L}(\theta_t) \cdot \partial_t\theta_t = -\|\nabla\mathcal{L}(\theta_t)\|^2 \leq 0$. It remains to tackle only the other case, thus for the remainder of the proof, let us assume $\mathcal{L}(\theta_0) > \varepsilon$, and in particular $\varepsilon \leq \varepsilon_0$, then define $T = \inf\left(\{\, t \in \mathbb{R}_+ \mid \|\theta_t - \theta_0\| \leq R \,\} \bigcap \{\, t \in \mathbb{R}_+ \mid \mathcal{L}(\theta_t) > \varepsilon \,\}\right)$.

**Radius upper-bound by separable KŁ integration.** Define $r : [0, T[\, \to \mathbb{R}_+$ as $r : t \mapsto \int_0^t \|\partial_t\theta_u\|\, \mathrm{d}u$. Observe that for all $t \leq T$, by triangle inequality it holds $\|\theta_t - \theta_0\| \leq r(t)$. Therefore,

$$\partial_t r_t = \|\partial_t\theta_t\| = \|\nabla\mathcal{L}(\theta_t)\| \underset{(S1)}{=} \frac{\|\nabla\mathcal{L}(\theta_t)\|^2}{\|\nabla\mathcal{L}(\theta_t)\|} \underset{(S2)}{\leq} \frac{\|\nabla\mathcal{L}(\theta_t)\|^2}{\frac{1}{r_t+c}\left(\mathcal{L}(\theta_t) - \varepsilon\right)_+} \underset{(S3)}{=} (r_t + c)\frac{-\partial_t(\mathcal{L}\circ\theta)_t}{\mathcal{L}(\theta_t) - \varepsilon}$$

where (S1) is valid because $\|\nabla\mathcal{L}(\theta_t)\| > 0$ by the initial assumption and restriction to $\mathcal{L}(\theta_t) > \varepsilon$, and (S2) is the initial assumption again, and (S3) uses $\partial_t(\mathcal{L}\circ\theta)_t = -\|\nabla\mathcal{L}(\theta_t)\|^2$ and $\mathcal{L}(\theta_t) > \varepsilon$.

This corresponds to the inequality $\partial_t(\Psi \circ r) \leq \partial_t(\Phi \circ \mathcal{L} \circ \theta)$ where $\Psi : s \mapsto \log(s + c)$ and $\Phi : s \mapsto -\log(s - \varepsilon)$. Integrating the inequality between $0$ and $t < T$ yields the inequality

$$\log\left(\frac{r_t + c}{0 + c}\right) = \Big[\log(r_u + c)\Big]_0^t \leq \Big[-\log\left(\mathcal{L}(\theta_u) - \varepsilon\right)\Big]_0^t = \log\left(\frac{\mathcal{L}(\theta_0) - \varepsilon}{\mathcal{L}(\theta_t) - \varepsilon}\right)$$

Thus after taking exponentials on both sides,

$$r_t + c \leq c\,\frac{\mathcal{L}(\theta_0) - \varepsilon}{\mathcal{L}(\theta_t) - \varepsilon} \tag{1}$$

**Radius bound injection and reintegration.** Define the $\varepsilon$-discounted loss $\mathcal{L}^\varepsilon : \Theta \to \mathbb{R}_+$ as $\mathcal{L}^\varepsilon : \theta \mapsto (\mathcal{L}(\theta) - \varepsilon)_+$. Injecting the radius bound inequality (1) into the initial assumption,

$$\partial_t(\mathcal{L}^\varepsilon \circ \theta)_t \underset{(S1)}{=} \partial_t(\mathcal{L}\circ\theta)_t = -\|\nabla\mathcal{L}(\theta_t)\|^2 \underset{(S2)}{\leq} -\frac{(\mathcal{L}^\varepsilon(\theta_t))^2}{(r_t + c)^2} \underset{(S3)}{\leq} -\frac{(\mathcal{L}^\varepsilon(\theta_t))^4}{(c\,\mathcal{L}^\varepsilon(\theta_0))^2}$$

where (S1) is because for all $t < T$, it holds $\mathcal{L}(\theta_t) = \mathcal{L}^\varepsilon(\theta_t) + \varepsilon$ by definition of $T$, (S2) is the initial assumption coupled with $r_t \geq \|\theta_t - \theta_0\|$, and (S3) is the injection of the radius bound inequality.

Therefore, let $\kappa = 3/\left(c\,\mathcal{L}^\varepsilon(\theta_0)\right)^2 \in \mathbb{R}_+^*$, and $\Xi : u \mapsto -1/u^3$. We have shown the inequality

$$\partial_t\left(\Xi \circ \mathcal{L}^\varepsilon \circ \theta\right)_t = +3\frac{\partial_t(\mathcal{L}^\varepsilon \circ \theta)_t}{(\mathcal{L}^\varepsilon(\theta_t))^4} \leq -\kappa$$

Integrating between $0$ and $t < T$, this yields $-\mathcal{L}^\varepsilon(\theta_t)^{-3} + \mathcal{L}^\varepsilon(\theta_0)^{-3} \leq -\kappa t$, thus after inverting $\Xi$,

$$\forall t < T, \quad \mathcal{L}^\varepsilon(\theta_t) \leq \left(\mathcal{L}^\varepsilon(\theta_0)^{-3} + \kappa t\right)^{-1/3} \tag{2}$$

**Conclusion.** We are now ready to conclude by case disjunction. If $T = +\infty$, Eq (2) immediately implies Convergence Criterion 1. If $T < +\infty$, there are two cases to tackle. First, if $\mathcal{L}(\theta_T) \leq \varepsilon$, then the bound of Eq (2) holds for $t < T$ and is extended to all $t \geq T$ by the non-increasing property $\mathcal{L}(\theta_t) \leq \mathcal{L}(\theta_T) \leq \varepsilon \leq \varepsilon_0$, which concludes. Lastly, if $\|\theta_T - \theta_0\| = R$, then by inequality (1) we get $\mathcal{L}(\theta_T) - \varepsilon \leq \frac{c}{R+c}(\mathcal{L}(\theta_0) - \varepsilon)$. Therefore $\mathcal{L}(\theta_T) \leq \tau \cdot \mathcal{L}(\theta_0) + (1-\tau) \cdot \varepsilon = \varepsilon_0$ where $\tau = c/(R+c)$. We conclude this case again by non-increasing extension, which completes the proof. $\qquad\square$

**Theorem E.2** (Quantitative version of Theorem 2.1)**.** *Let $(\varepsilon, \delta) \in \mathbb{R}_+^* \times \, ]0, 1]$.*
*If there exists $C \; : \; ]0, 1] \; \rightarrow \; \mathbb{R}_+$ such that for any $s \; \in \; \mathcal{S}_0$ and for any $\delta \; \in \; ]0, 1]$ it holds $\mathbb{P}_{(g, \theta) \sim \mathcal{N}_s} \left[ \mathcal{L} \circ F_{(s,g)}(\theta) \leq C(\delta) \right] \geq 1 - \delta$, and if there exists constants $\varepsilon_0 < \varepsilon$ and $\delta_0 < \delta$ such that Condition C1 is satisfied with parameters $(\varepsilon_0, \delta_0)$, then let $c \in \mathbb{R}_+$ be the corresponding intercept, let $s_1 \in \mathcal{S}$ be the threshold size associated by Condition C1 to radius $R = c\,(C(\delta - \delta_0) - \varepsilon_0) / (\varepsilon - \varepsilon_0)$, and let $\kappa = 3\, c^{-2}\, C(\delta - \delta_0)^{-2} \in \mathbb{R}_+^*$.*

*It holds for all $s \in \mathcal{S}_0$ satisfying $s \succeq s_1$, that if $(g, \theta_0) \sim \mathcal{N}_s$, then the pair $(\mathcal{L} \circ F_{(s,g)}, \theta_0)$ satisfies Convergence Criterion 1 with limit error $\varepsilon$ and constant $\kappa$ with probability at least $(1 - \delta)$.*

Note that Theorem 2.1 is a direct consequence of Theorem E.2, it thus suffices to prove the latter.

*Proof of Theorem E.2.* Let $s \in \mathcal{S}_0$ such that $s \succeq s_1$. Let $(g, \theta_0) \sim \mathcal{N}_s$. Let $A$ be the event: $[\forall r \leq R, B(\theta_0, r) \subseteq \mathcal{A}_{s,g}(c + r, \varepsilon_0)]$. By Condition C1, $\mathbb{P}(A) \geq (1 - \delta_0)$. Let $B$ be the event: $[\mathcal{L} \circ F_{(s,g)}(\theta_0) \leq C(\delta - \delta_0)]$. By boundedness assumption, $\mathbb{P}(B) \geq 1 - (\delta - \delta_0)$. Therefore, by union bound, $\mathbb{P}(A \cap B) \geq 1 - \mathbb{P}(\neg A) - \mathbb{P}(\neg B) \geq 1 - \delta$. Let $S$ be the event $[(\mathcal{L} \circ F_{(s,g)}, \theta_0)$ satisfies Convergence Criterion 1 with limit error $\varepsilon$ and constant $\kappa]$. It only remain to show that $S \subseteq A \cap B$, since that will imply $\mathbb{P}(S) \geq \mathbb{P}(A \cap B) \geq 1 - \delta$.

We will do so by leveraging Lemma E.1. Write for shortness $C = C(\delta - \delta_0)$. Let $(g, \theta_0) \in G_s \times \Theta_s$ be such that $\forall r \leq R, B(\theta_0, r) \subseteq \mathcal{A}_{s,g}(c + r, \varepsilon_0)$, and $\mathcal{L} \circ F_{(s,g)}(\theta_0) \leq C$ (i.e. both events $A$ and $B$ are realised). Let $\theta \in B(\theta_0, R)$, and let us show that the separable Kurdyka-Łojasiewicz bound $\|\nabla(\mathcal{L} \circ F_{(s,g)})(\theta)\| \geq (\mathcal{L} \circ F_{(s,g)}(\theta) - \varepsilon_0)_+ / (\|\theta - \theta_0\| + c)$ holds. Let $r = \|\theta - \theta_0\|$. Since $\theta \in \mathcal{A}_{s,g}(c + r, \varepsilon_0)$, let $u \in \Theta_s$ be such that $\|u\| \leq c + r$ and $\|F_{(s,g)}(\theta) + \mathrm{d}F_{(s,g)}(\theta) \cdot u - f^\star\|_{\mathcal{D}}^2 < \varepsilon_0$. First, note that if $\mathcal{L} \circ F_{(s,g)}(\theta) \leq \varepsilon$ then the bound holds immediately since the right-hand side is null, thus we can assume in the following that $\mathcal{L} \circ F_{(s,g)}(\theta) > \varepsilon_0$. By the variational form of the $\ell_2$-norm,

$$\|\nabla(\mathcal{L} \circ F_{(s,g)})(\theta)\|^2 = \sup_{v \in \Theta_s} \frac{(\mathrm{d}(\mathcal{L} \circ F(s, g))(\theta) \cdot v)^2}{\|v\|^2} \geq \frac{(-\mathrm{d}(\mathcal{L} \circ F(s, g))(\theta) \cdot u)^2}{\|u\|^2} \tag{3}$$

Then since $\mathcal{L}(f) = \|f - f^\star\|_{\mathcal{D}}^2$, we can expand the derivative to

$$
\begin{aligned}
-\mathrm{d}(\mathcal{L} \circ F_{(s,g)})(\theta) \cdot u &= -\mathrm{d}\mathcal{L}(F_{(s,g)}(\theta)) \cdot \big( \mathrm{d}F_{(s,g)}(\theta) \cdot u \big) \\
&= -2 \left\langle \mathrm{d}F_{(s,g)}(\theta) \cdot u, F_{(s,g)}(\theta) - f^\star \right\rangle_{\mathcal{D}} \\
&\underset{\text{(S1)}}{=} \mathcal{L} \circ F_{(s,g)}(\theta) + \big\| \mathrm{d}F_{(s,g)}(\theta) \cdot u \big\|_{\mathcal{D}}^2 - \big\| F_{(s,g)}(\theta) + \mathrm{d}F_{(s,g)}(\theta) \cdot u - f^\star \big\|_{\mathcal{D}}^2 \\
&\underset{\text{(S2)}}{\geq} \mathcal{L} \circ F_{(s,g)}(\theta) + 0 - \varepsilon_0
\end{aligned}
$$

where (S1) is the parallelogram indentity $-2\langle a, b \rangle = \|a\|^2 + \|b\|^2 - \|a + b\|^2$ with the definition of $\mathcal{L}$, and (S2) is the approximation inequality defining $u$. Using the positive part $(\cdot)_+ : s \mapsto \max\{0, s\}$, we can thus bound the square of this quantity irrespectively of its sign, as follows

$$\big( -\mathrm{d}(\mathcal{L} \circ F_{(s,g)})(\theta) \cdot u \big)^2 \geq \big( \mathcal{L} \circ F_{(s,g)}(\theta) - \varepsilon_0 \big)_+^2$$

Observing that $\|u\| \leq c + r$ and injecting both of these bounds into Eq (3) concludes the proof of the separable Kurdyka-Łojasiewicz bound.

Applying now Lemma E.1, this implies that the pair $(\mathcal{L} \circ F_{(s,g)}, \theta_0)$ satisfies Convergence Criterion 1 with limit error $\varepsilon_1 = \tau \cdot \mathcal{L} \circ F_{(s,g)}(\theta_0) + (1 - \tau) \cdot \varepsilon_0$ for $\tau = c/(R + c)$ and constant $\kappa_1 = 3\, c^{-2} \big( \mathcal{L} \circ F_{(s,g)}(\theta_0) - \varepsilon_0 \big)_+^2$. By decrease with respect to $\kappa$ and increase with respect to $\varepsilon$ of the Convergence Criterion 1 bound, it now only remains to show that $\varepsilon_1 \leq \varepsilon$ and $\kappa_1 \geq \kappa$. The latter is immediate because $\big( \mathcal{L} \circ F_{(s,g)}(\theta_0) - \varepsilon_0 \big)_+ \leq \mathcal{L} \circ F_{(s,g)}(\theta_0) \leq C$, thus $\kappa_1 \geq 3\, c^{-2}\, C^{-2} = \kappa$. Then, use the fact that $\tau = c/(R+c) = (\varepsilon - \varepsilon_0)/(C - \varepsilon_0)$ to get $\varepsilon_1 \leq \tau\, C + (1 - \tau)\, \varepsilon_0 = \tau(C - \varepsilon_0) + \varepsilon_0 = \varepsilon$. Thus the pair $(\mathcal{L} \circ F_{(s,g)}, \theta_0)$ satisfies Convergence Criterion 1 with limit error $\varepsilon$ and constant $\kappa$, which completes the proof that $S \subseteq A \cap B$ and thus concludes the proof of Theorem E.2. □

## F    MORPHISMS OF LIFTED PERCEPTRON MODULES TO TRACK ACTIVATIONS

**Definition F.1** (Morphism of lifted perceptrons). *Let* $(G, \pi_G, c_G) \in \text{LiftPMod}(\mathbb{B}, \mathcal{C})$ *and* $(H, \pi_H, c_H) \in \text{LiftPMod}(\mathbb{B}, \mathcal{C})$. *A morphism of lifted perceptrons* $\varphi : (G, \pi_G, c_G) \to (H, \pi_H, c_H)$ *is a fibration* $\varphi : G \to H$ *such that* $\pi_G = \pi_H \circ \varphi$ *and* $c_G = c_H \circ \varphi$.

In other words, a morphism of lifted perceptrons G to H is a correspondance $\varphi$ between the underlying graphs $G$ and $H$ that preserves the activation-defining "types" $\pi$ of vertices and edges, and the input connections $c$, i.e. such that the following three diagrams commute

$$
\begin{array}{ccc}
V_G \xrightarrow{\varphi} V_H & \quad E_G \xrightarrow{\varphi} E_H & \quad I_G \xrightarrow{\varphi} I_H \\
\pi_G \searrow \quad \swarrow \pi_H & \quad \pi_G \searrow \quad \swarrow \pi_H & \quad c_G \searrow \quad \swarrow c_H \\
V_B & \quad E_B & \quad \mathcal{C}
\end{array}
$$

In the example of Figure 4 and Figure 5 of multi-layer perceptron with three layers, the base graph has $V_B = \{0, 1, 2, 3\}$, and a lifted vertex $v \in V$ has $\pi(v) = k \in V_B$ if the vertex $v$ is located in layer $k$. The condition that $\varphi$ preserves $\pi$ means that a vertex of layer $k$ in G must be mapped to a vertex of layer $k$ in H. Mapping across layers would not make much sense, because it would not be compatible with the definition of the forward function. Similarly for this example, we have $\mathcal{C} = \{r, g, b\}$ indicating which of the lifted inputs corresponds to each channel. The mapping $\varphi$ must also preserve these connections (map the $r$ channel of $G$ to the $r$ channel of $H$, etc.).

**Proposition F.2** (LiftPMod-morphisms preserve activations).
*Let* $G \in \text{LiftPMod}(\mathbb{B}, \mathcal{C})$ *and* $H \in \text{LiftPMod}(\mathbb{B}, \mathcal{C})$. *If* $\varphi : G \to H$ *is a* $\text{LiftPMod}(\mathbb{B}, \mathcal{C})$-*morphism, then for all* $w_G \in \text{Param}(G)$ *and* $w_H \in \text{Param}(H)$, *it holds*

$$
\left[ w_G = \varphi^* w_H \right] \Rightarrow \left[ F[G](w_G, \cdot) = \varphi^* \circ F[H](w_H, \cdot) \right]
$$

Thus one can transform the graph and weights, then compute the forward, or compute the forward first, then transform the activations and input connections. This property of compatibility of the forward operator and the morphisms of LiftPMod corresponds to the following commutative diagram

$$
\begin{array}{ccc}
(G, w_G) & \xleftarrow{\varphi^*} & (H, w_H) \\
F[\cdot](\cdot, -) \downarrow & & \downarrow F[\cdot](\cdot, -) \\
F[G](w_G, \cdot) & \xleftarrow{\varphi^*} & F[H](w_H, \cdot)
\end{array}
$$

The proof of this proposition is deferred to Appendix F.1. Intuitively, this proposition states that a fibration of graphs compatible with the lift and preserving the weights means that the lifted perceptron modules "compute the same thing". Fibrations are crucial for this property to hold, a homomorphism of graphs for $\varphi$ would not suffice, thus the restriction to fibrations in the previous definition.

### F.1    PROOF THAT MORPHISMS PRESERVE ACTIVATIONS

*Proof of Proposition F.2.* Let $G = (G, \pi_G, c_G) \in \text{LiftPMod}(\mathbb{B}, \mathcal{C})$ and $H = (H, \pi_H, c_H) \in \text{LiftPMod}(\mathbb{B}, \mathcal{C})$. Let $\varphi : G \to H$ be a fibration such that $(\pi_G = \pi_H \circ \varphi) : V_G \to V_B$ and $(c_G = c_H \circ \varphi) : I(G) \to C$. Let $w_G \in \text{Param}(G)$ and $w_H \in \text{Param}(H)$, be such that $w_G = \varphi^* w_H$. Let us show that $F[G](w_G, \cdot) = \varphi^* \circ F[H](w_H, \cdot) : \prod_{u \in \mathcal{C}} \mathcal{Y}_{p_C(u)} \to \prod_{v \in V_G} \mathcal{Y}_{\pi_G(v)}$.

Let us check first that these expressions are all well defined. As a function of sets $\varphi : V_G \to V_H$ induces a pullback $\varphi^* : \prod_{u \in V(H)} \mathcal{Y}_{\pi_H(u)} \to \prod_{v \in V(G)} \mathcal{Y}_{\pi_H(\varphi(v))} = \prod_{v \in V(G)} \mathcal{Y}_{\pi_G(v)}$. Therefore, the expression $\varphi^* \circ F[H](w_H, \cdot)$ is well-defined and has the correct type signature. Similarly for the weights, $w_H \in \text{Param}(H) = \prod_{e \in E_H} \mathcal{W}_{\pi_H(e)}$. As a function of sets, $\varphi : E_G \to E_H$ defines a pullback $\varphi^* : \prod_{e \in E_H} \mathcal{W}_{\pi_H(e)} \to \prod_{e \in E_G} \mathcal{W}_{\pi_H(\varphi(e))} = \prod_{e \in E_G} \mathcal{W}_{\pi_G(e)} = \text{Param}(G)$, where the first equality follows from $\pi_G = \pi_H \circ \varphi$. Therefore the assumption $w_G = \varphi^* w_H$ is well-formed.

Let $x \in \prod_{u \in \mathcal{C}} \mathcal{Y}_{p_{\mathcal{C}}(u)}$. We will proceed by induction on a topological ordering $\prec$ over $V_G$ (available since $G$ is acyclic). Let $U \subseteq V_G$ be the set of vertices such that

$$v \in U \Leftrightarrow [\mathrm{F}[\mathrm{G}](w_G, x)]_v \neq [\varphi^* \, \mathrm{F}[\mathrm{H}](w_H, x)]_v$$

Let us show that $U = \varnothing$. By contradiction, if $U$ is not empty, then let $v \in U$ be minimal for the total order $\prec$. Proceed by case disjunction on $(\mathrm{I}_G, V_G \setminus \mathrm{I}_G)$. For the first case, if $v \in U \cap \mathrm{I}_G$, then $\varphi(v) \in \mathrm{I}_H$. Therefore by definition of the activation map, we get the contradiction

$$[\mathrm{F}[\mathrm{G}](w_G, x)]_v \underset{(\mathrm{S1})}{=} [c_G^* x]_v \underset{(\mathrm{S2})}{=} x_{c_G(v)} \underset{(\mathrm{S3})}{=} x_{c_H(\varphi(v))} \underset{(\mathrm{S4})}{=} [\mathrm{F}[\mathrm{H}](w_H, x)]_{\varphi(v)} \underset{(\mathrm{S5})}{=} [\varphi^* \mathrm{F}[\mathrm{H}](w_H, x)]_v$$

where (S1) is the definition of the activation map of lifts (Definition 3.5 and Definition 3.4) for $v \in \mathrm{I}_G$, (S2) is the definition of the pullback $c_G^*$, (S3) follows from $c_G = c_H \circ \varphi$, (S4) is the same argument as (S1) for $\varphi(v) \in \mathrm{I}_H$, and (S5) the definition of $\varphi^*$, which concludes this first case.

For the second case, assume $v \in U \cap (V_G \setminus \mathrm{I}_G)$. Write for shortness $g = \mathrm{F}[\mathrm{G}](w_G, x)$ and $h = \mathrm{F}[\mathrm{H}](w_H, x)$. The fact that $v \in U$ implies that $g_v \neq h_{\varphi(v)}$, but the assumption that $v$ is minimal in $U$ implies that for all $u \in V_G$, if $u \prec v$, then $g_u = h_{\varphi(u)}$ (otherwise $u \in U$ by definition, and thus $v$ is not minimal). Writing $G_a(-, v) = \pi_G^{-1}(a) \cap G(-, v)$ for shortness (resp. $H_a(-, t) = \pi_H^{-1}(a) \cap H(-, t)$), by definition of activation maps (Def. 3.4) and lift (Def. 3.5),

$$g_v = \sigma_{\pi_G(v)}\left(\left(\sum_{u \in G_a(-, v)} M_{\pi_G(u, v)}\left([w_G]_{(u, v)}, g_u\right)\right)_{a \in B(-, \pi_G(v))}\right)$$

Now, by definition, $\varphi : G \to H$ being a fibration implies that $\varphi|_{G(-, v)} : G(-, v) \to H(-, \varphi(v))$ is a bijection (Definition C.3). Moreover, the fact that $\pi_G = \pi_H \circ \varphi$ implies that for all $a \in V_B$, the restriction of $\varphi$ is also a bijection $G_a(-, v) \to H_a(-, \varphi(v))$. Therefore, we can replace the terms in the sum one-by-one (note that this is not possible with a homomorphism, we need a fibration to do this), as follows. Write $\psi = \varphi|_{G(-, v)}$ the bijection, and rewrite the previous equality

$$g_v \underset{(\mathrm{S1})}{=} \sigma_{\pi_G(v)}\left(\left(\sum_{u \in G_a(-, v)} M_{\pi_G(u, v)}\left([w_G]_{(u, v)}, g_u\right)\right)_{a \in B(-, \pi_G(v))}\right)$$

$$\underset{(\mathrm{S2})}{=} \sigma_{\pi_G(v)}\left(\left(\sum_{u \in H_a(-, \varphi(v))} M_{\pi_G(\psi^{-1}(u), v)}\left([w_G]_{(\psi^{-1}(u), v)}, g_{\psi^{-1}(u)}\right)\right)_{a \in B(-, \pi_G(v))}\right)$$

$$\underset{(\mathrm{S3})}{=} \sigma_{\pi_H(\varphi(v))}\left(\left(\sum_{u \in H_a(-, \varphi(v))} M_{\pi_H(\varphi(\psi^{-1}(u), v))}\left([w_G]_{(\psi^{-1}(u), v)}, g_{\psi^{-1}(u)}\right)\right)_{a \in B(-, \pi_H(\varphi(v)))}\right)$$

$$\underset{(\mathrm{S4})}{=} \sigma_{\pi_H(\varphi(v))}\left(\left(\sum_{u \in H_a(-, \varphi(v))} M_{\pi_H(u, \varphi(v))}\left([w_H]_{(u, \varphi(v))}, h_u\right)\right)_{a \in B(-, \pi_H(\varphi(v)))}\right)$$

$$\underset{(\mathrm{S5})}{=} h_{\varphi(v)}$$

where (S1) is by definition of $g$, (S2) is the rewriting of terms reindexed by the bijection, (S3) uses $\pi_G = \pi_H \circ \varphi$, (S4) is the assumption that $\psi^{-1}(u) \notin U$ by minimality of $v$ and $\varphi(\psi^{-1}(u)) = u$, applied to $g_{\psi^{-1}(u)} = h_{\varphi(\psi^{-1}(u))} = h_u$ and rewriting of $w_G = \varphi^* w_H$. Finally (S5) is the definition of $h$, constitutes the contradiction for the second case, thus concludes the induction and the proof. $\square$

## G  TRACKING ACTIVATIONS ACROSS DEFORMATIONS OF WEIGHTS

**Definition G.1** (Quantitative continuity). *Let $G = (\mathrm{V}_G, \mathrm{E}_G)$ be a graph.*
*Let $\mathrm{G} = ((I, T), (\mathcal{Y}, \mathcal{W}, \mathcal{Z}), (M, \sigma)) \in \mathrm{PMod}(G)$ and $w^0 \in \mathrm{Param(G)}$. Let $\mathcal{X} \subseteq \prod_{v \in I} \mathcal{Y}_v$.*

*For any $v \in \mathrm{V}_G$, write $F_v = \sigma_v \circ \big(M_{(u,v)}\big)_{u \in G(-,v)} : \prod_{u \in G(-,v)} \mathcal{W}_{(u,v)} \times \mathcal{Y}_u \to \mathcal{Y}_v$.*
*Write $g^0 = \mathrm{F}[\mathrm{G}](w^0, \cdot)$ such that for $x \in \mathcal{X}$ and $v \in \mathrm{V}_G$, we have $g_v^0(x) = \big[\mathrm{F}[\mathrm{G}](w^0, x)\big]_v \in \mathcal{Y}_v$.*

*For any $\eta \in \mathbb{R}_+^*$ and $x \in \mathcal{X}$, define $L(\eta, x) : \mathrm{V}_G \mapsto \mathbb{R}_+$ inductively as follows from $[L(\eta, x)]_b = 0$*
*for $b \in I$ and propagated to $v \in \mathrm{V}_G \setminus I$ as*

$$[L(\eta, x)]_v = \sup_{\substack{w \in \prod_{u \in G(-,v)} \mathcal{W}_{(u,v)} \\ \forall u, \|w_{(u,v)} - w_{(u,v)}^0\| \leq \eta}} \quad \sup_{\substack{g \in \prod_{u \in G(-,v)} \mathcal{Y}_v \\ \forall u, \|g_u - g_u^0(x)\| \leq [L(\eta,x)]_u}} \big\| F_v(w, g) - F_v(w^0, g^0(x)) \big\|$$

*Then write $L[\mathrm{G}, w^0](\eta, \mathcal{X}) = \sup_{x \in \mathcal{X}} \max_{v \in \mathrm{V}_G} [L(\eta, x)]_v$. If $\mathcal{X}$ is compact, this constant is finite.*

This notion will be used in conjunction with Proposition F.2, which is the reason this convoluted definition is chosen instead of just $\Big[\sup_{x \in \mathcal{X}} \sup_{\|w - w^0\|_\infty \leq \eta} \|\mathrm{F}[\mathrm{G}](w, x) - \mathrm{F}[\mathrm{G}](w^0, x)\|_\infty\Big]$, which is much more simply stated. The construction of Definition G.1 is such that the constant $L$ is preserved by morphisms, thus if we start with two networks with similar weights and a morphism between them, we can track the evolution of all activations with a single constant.

**Proposition G.2** (Tracking activations through morphisms and across weight deformations).
*Let $\mathrm{G} = (G, \pi, c) \in \mathrm{LiftPMod}(\mathbb{B}, \mathcal{C})$ and $w_G \in \mathrm{Param(G)}$ for underlying graph $G = (\mathrm{V}_G, \mathrm{E}_G)$.*
*Let $\mathrm{H} \in \mathrm{LiftPMod}(\mathbb{B}, \mathcal{C})$ and $w_G \in \mathrm{Param(H)}$. Let $\mathcal{X} \subseteq \prod_{u \in \mathcal{C}} \mathcal{Y}_{\pi_C(u)}$.*

*Let $\eta \in \mathbb{R}_+$. Assume $\varphi : \mathrm{G} \to \mathrm{H}$ is a $\mathrm{LiftPMod}(\mathbb{B}, \mathcal{C})$-morphism such that*

$$\forall e \in \mathrm{E}_G, \ \|[w_G]_e - [\varphi^* w_H]_e\| \leq \eta$$

*Then for any $v \in \mathrm{V}_G$ it holds $\sup_{x \in \mathcal{X}} \Big\| [\mathrm{F}[\mathrm{G}](w_G, x)]_v - [\mathrm{F}[\mathrm{H}](w_H, x)]_{\varphi(v)} \Big\| \leq L[\mathrm{H}, w_H](\eta, \mathcal{X})$*

*Proof of Proposition G.2.* To avoid confusions, write the lifted modules $\mathrm{G} = (G, \pi_G, c_G)$ and $\mathrm{H} = (H, \pi_H, c_H)$, for the underlying graphs $G = (\mathrm{V}_G, \mathrm{E}_G)$ and $H = (\mathrm{V}_H, \mathrm{E}_H)$. For $v_H \in \mathrm{V}_H$, we use the notation $[L(\eta, x)]_{v_H}$ associated to vertices $\mathrm{V}_H$ by Definition G.1 in the construction of $L[\mathrm{H}, w_H](\eta, \mathcal{X})$. Let $x \in \mathcal{X}$. For shortness, define $g = \mathrm{F}[\mathrm{G}](w_G, x)$ and $h = \mathrm{F}[\mathrm{H}](w_H, x)$. Let us show by induction on $v \in \mathrm{V}_G$ that $\|g_v - h_{\varphi(v)}\| \leq [L(\eta, x)]_{\varphi(v)}$.

To that end, let $v \in \mathrm{V}_G$ be the minimal vertex satisfying $\|g_v - h_{\varphi(v)}\| > [L(\eta, x)]_{\varphi(v)}$, for a topological ordering of $G$. If $\pi_G(v) \in \mathrm{I}_B$, then by definition of a lift and forward (resp. Def. 3.5 and Def. 3.4) it holds $g_v = x_{c_G(v)} = x_{c_H(\varphi(v))} = h_{\varphi(v)}$ which constitutes the contradiction. For the remaining case of $v \in \mathrm{V}_G \setminus \pi_G^{-1}(\mathrm{I}_B)$, let $\psi = \varphi|_{G(-,v)}$ be the bijection between parents $\psi : G(-,v) \to H(-, \varphi(v))$ (existence and bijectvity follow from the fact that $\varphi$ is a fibration). Thus,

$$g_v = \sigma_{\pi_G(v)} \left( \left( M_{\pi_G(u,v)} \left( [w_G]_{(u,v)}, g_u \right) \right)_{u \in G(-,v)} \right)$$

$$= \sigma_{\pi_H(\varphi(v))} \left( \left( M_{\pi_H(s, \varphi(v))} \left( [w_G]_{(\psi^{-1}(s), v)}, g_{\psi^{-1}(s)} \right) \right)_{s \in H(-, \varphi(v))} \right)$$

However by assumption we have $\|[w_G]_{\psi^{-1}(s), v} - [w_H]_{s, \varphi(v)}\| \leq \eta$, and by minimality of $v$ it holds $\|g_{\psi^{-1}(s)} - h_s\| \leq [L(\eta, x)]_s$, thus by definition of $[L(\eta, x)]_{\varphi(v)}$ as the supremum over such values, we get $\|g_v - h_{\varphi(v)}\| \leq [L(\eta, x)]_{\varphi(v)}$, which consitutes the contradiction and concludes the induction. Finally, observing that $[L(\eta, x)]_{\varphi(v)} \leq L[\mathrm{H}, w_H](\eta, \mathcal{X})$ concludes the proof. $\qquad \square$

# H FROM A COVERING PARTIAL MORPHISM TO TANGENT APPROXIMATION

## H.1 PARTIAL MORPHISMS, A STRONG MEANING OF "SUB-NETWORK"

**Definition H.1** (Partial fibration). *A partial fibration $(S, \varphi) : G \rightharpoonup H$ is a subgraph $S$ of $G$ such that the inclusion map $\iota : S \to G$ is a fibration, together with a fibration $\varphi : S \to H$.*

A subgraph $S$ of $G$ such that the inclusion $\iota : S \to G$ is a fibration will lead to an "extraction" of a perceptron module from a larger module, compatible with forward functions.

**Definition H.2** (Partial morphisms of LiftPMod).
*Let $(G, \pi_G, c_G) \in \mathrm{LiftPMod}(\mathbb{B}, \mathcal{C})$ and $(H, \pi_H, c_H) \in \mathrm{LiftPMod}(\mathbb{B}, \mathcal{C})$. A partial morphism of $\mathrm{LiftPMod}(\mathbb{B}, \mathcal{C})$ is a partial fibration of graphs $(S, \varphi) : G \rightharpoonup H$, such that the inclusion $\iota : S \to G$ induces a $\mathrm{LiftPMod}(\mathbb{B}, \mathcal{C})$-morphism $\iota : (S, \pi_G|_S, c_G|_S) \to (G, \pi_G, c_G)$, and the fibration $\varphi : S \to H$ induces a $\mathrm{LiftPMod}(\mathbb{B}, \mathcal{C})$-morphism $\varphi : (S, \pi_G|_S, c_G|_S) \to (H, \pi_H, c_H)$.*

**Definition H.3** (Covering partial matchings).
*Let $\mathrm{G} = (G, \pi_G, c_G) \in \mathrm{LiftPMod}(\mathbb{B}, \mathcal{C})$ and $w_G \in \mathrm{Param}(\mathrm{G})$.
Let $\mathrm{H} = (H, \pi_H, c_H) \in \mathrm{LiftPMod}(\mathbb{B}, \mathcal{C})$ and $w_H \in \mathrm{Param}(\mathrm{H})$, over graph $H = (\mathrm{V}_H, \mathrm{E}_H)$.*

*Let $\alpha : \mathrm{V}_H \to [0, 1]$ and $\eta \in \mathbb{R}_+^*$. A partial $\mathrm{LiftPMod}(\mathbb{B}, \mathcal{C})$-morphism $(S, \varphi) : \mathrm{G} \rightharpoonup \mathrm{H}$ is said to be an $(\alpha, \eta)$-covering if it satsfies the following conditions:*

$$\forall e \in E(S), \quad \|[\iota^* w_G]_e - [\varphi^* w_H]_e\| \leq \eta$$
$$\forall v \in \mathrm{V}_H, \quad \#\varphi^{-1}(v) \geq \alpha(v) \cdot \#\pi_G^{-1}(\pi_H(v))$$

*We write $\mathcal{M}_\eta^\alpha[(\mathrm{G}, w_G), (\mathrm{H}, w_H)]$ the set of such partial morphisms from $\mathrm{G}$ to $\mathrm{H}$. Additionally, we say that $(\mathrm{G}, w_G)$ has $(\alpha, \eta)$-cover of $(\mathrm{H}, w_H)$ if the set of such partial morphisms is not empty.*

## H.2 TANGENT APPROXIMATION BY SUBNETWORK-MATCHING

Let us show that the tangent approximation property at a point can be reduced to a covering property.

**Proposition H.4.**
*Let $\mathrm{G} \in \mathrm{LiftPMod}(\mathbb{B}, \mathcal{C})$, and let $\theta_0 = (w^0, a^0) \in \mathrm{Param}(\mathrm{G}) \times \mathrm{LinReadout}(\mathrm{G}, k)$. Define the shorthands $\Theta = \mathrm{Param}(\mathrm{G}) \times \mathrm{LinReadout}(\mathrm{G}, k)$ and $F : \Theta \to (\mathcal{X} \to \mathbb{R}^k)$ the forward function $F : (w, a) \mapsto a \cdot \mathrm{F}[\mathrm{G}](w, -)$, with norm $\|(w, a)\|_\Theta^2 = \sum_{e \in \mathrm{E}_G} \|w_e\|^2 + \sum_{i \in [k], v \in T(\mathrm{G})} \|a_{i,v}\|^2$.*

*Let $\mathrm{G}^\star = (G^\star, \pi_\star, c_\star) \in \mathrm{LiftPMod}(\mathbb{B}, \mathcal{C})$ and $(w^\star, a^\star) \in \mathrm{Param}(\mathrm{G}^\star) \times \mathrm{LinReadout}(\mathrm{G}^\star, k)$ over $G^\star = (\mathrm{V}^\star, \mathrm{E}^\star)$. Let $\eta \in \mathbb{R}_+$, and $\alpha : \mathrm{V}^\star \to ]0, 1]$. If $(\mathrm{G}, w^0)$ has $(\alpha, \eta)$-cover of $(\mathrm{G}^\star, w^\star)$, then*

$$\inf_{\substack{u \in \Theta \\ \|u\|_\Theta \leq \kappa}} \left\| F(\theta_0) + \mathrm{d}F(\theta_0) \cdot u - f^\star \right\|_\mathcal{D} \leq \left\| a^\star \cdot \mathrm{F}[\mathrm{G}^\star](w^\star, \cdot) - f^\star \right\|_\mathcal{D} + C^\star \cdot L[\mathrm{G}^\star, w^\star](\eta, \mathcal{X})$$

*with $C^\star = \sum_{i \in [k]} \sum_{v \in V^\star} \frac{\|a_{i,v}^\star\|}{\sqrt{\#\pi_\star^{-1}(\pi_\star(v))}}$ and $\kappa = \|a^0\| + \sqrt{\sum_{i \in [k]} \sum_{v \in V^\star} \frac{\|a_{i,v}^\star\|^2}{\alpha(v) \cdot \#\pi_\star^{-1}(\pi_\star(v))}}$.*

Since $L[\mathrm{G}^\star, w^\star](\eta, \mathcal{X}) \xrightarrow[\eta \to 0]{} 0$, this bound tends to $\|a^\star \cdot \mathrm{F}[\mathrm{G}^\star](w^\star, \cdot) - f^\star\|_\mathcal{D}$ in the limit $\eta \to 0$.

This should be fairly straightforwardly connected to the tangent approximation property at $\theta_0$. Indeed if $\|\mathrm{F}[\mathrm{G}^\star](w^\star, \cdot) - f^\star\|_\mathcal{D}^2 \leq \varepsilon$ then this property will constitute tangent approximation at $\theta_0$ with any error $\varepsilon_1 > \varepsilon$ provided $\eta$ is taken sufficiently small. The tangent radius $\kappa$ also has a manageable form.

Note that this is consistent with the intuition given by Frankle & Carbin (2019): the *Lottery Ticket Hypothesis* postulates that if a large enough network contains a smaller network which achieves low loss (a *Lottery Ticket*), then the larger network will achieve low loss. The principal distinction with the present discussion — apart from the added generality of generic perceptron modules instead of just multi-layer perceptrons — is that Frankle & Carbin (2019) considered dense networks whose "sub-networks" were extracted by *homomorphism*, whereas our proof is restricted to extractions by *fibration*, which is much more restrictive. The overarching intuition carries over even if the details don't match, largely because these early postulates were intentionally fuzzy in nature to guide intuition, whereas our proofs are rigid by design in order to yield precise empirical predictions.

*Proof of Proposition H.4.* Note first that the function $F : (w, a) \mapsto a \cdot \mathrm{F[G]}(w, -)$ is linear with respect to $a$, therefore for any $b \in \mathrm{LinReadout}(\mathrm{G}, k)$, it holds $\mathrm{d}F(w, a) \cdot (0, b) = b \cdot \mathrm{F[G]}(w, -)$. Then, from $(S, \varphi) : G \rightharpoonup G^\star$ a partial fibration with $(\alpha, \eta)$-cover, construct $u \in \mathrm{LinReadout}(\mathrm{G}, k)$ defined for $v \in T(\mathrm{G})$ by $u_v = a^\star_{\varphi(v)}/s_v$, where $s_v = \#\varphi^{-1}(\varphi(v))\sqrt{\#\pi_\star^{-1}(\pi(v))/\#\pi^{-1}(\pi(v))}$.

$$\begin{aligned}
&\|F(w^0, a^0) + \mathrm{d}F(w^0, a^0) \cdot (0, u - a^0) - f^\star\|_\mathcal{D} \\
&\underset{(S1)}{=} \|\mathrm{d}F(w^0, a^0) \cdot (0, u) - f^\star\|_\mathcal{D} \underset{(S2)}{=} \|u \cdot \mathrm{F[G]}(w^0, -) - f^\star\|_\mathcal{D} \\
&\underset{(S3)}{\leq} \|a^\star \cdot \mathrm{F[G^\star]}(w^\star, -) - f^\star\|_\mathcal{D} + \|u \cdot \mathrm{F[G]}(w^0, -) - a^\star \cdot \mathrm{F[G^\star]}(w^\star, -)\|_\mathcal{D}
\end{aligned} \tag{4}$$

where (S1) is the fact that for any $a$, we have $F(w^0, a) = a \cdot \mathrm{F[G]}(w^0, -) = \mathrm{d}F(w^0, a^0) \cdot (0, a)$, together with linearity of the differential, (S2) is the simplification of the differential, and (S3) is subadditivity of the semi-norm $\|\cdot\|_\mathcal{D}$ on the space $(\mathcal{X} \to \mathbb{R}^k)$ (see Proposition K.2). Let us show that this is valid upper-bound for the infimum in the claim, because $\|(0, u - a^0)\|_\Theta \leq \kappa$. Indeed, by subadditivity of the norm $\|(0, u - a^0)\|_\Theta \leq \|(0, u)\|_\Theta + \|(0, a^0)\|_\Theta = \|u\| + \|a^0\|$. Moreover,

$$\begin{aligned}
\|u\|^2 &= \sum_{i \in [k]} \sum_{v \in T(\mathrm{G})} \|u_{i,v}\|^2 = \sum_{i \in [k]} \sum_{u \in T(\mathrm{G}^\star)} \sum_{v \in \varphi^{-1}(u)} \|a^\star_u\|^2/s_v^2 \\
&= \sum_{i \in [k]} \sum_{u \in T(\mathrm{G}^\star)} \sum_{v \in \varphi^{-1}(u)} \frac{\|a^\star_u\|^2 \, \#\pi^{-1}(\pi_\star(u))}{\#\varphi^{-1}(\varphi(v))^2 \, \#\pi_\star^{-1}(\pi_\star(u))} \\
&= \sum_{i \in [k]} \sum_{u \in T(\mathrm{G}^\star)} \frac{\|a^\star_u\|^2 \, \#\pi^{-1}(\pi_\star(u))}{\#\varphi^{-1}(u) \, \#\pi_\star^{-1}(\pi_\star(u))} \underset{(S1)}{\leq} \sum_{i \in [k]} \sum_{u \in T(\mathrm{G}^\star)} \frac{\|a^\star_u\|^2}{\alpha(u) \, \#\pi_\star^{-1}(\pi_\star(u))}
\end{aligned}$$

where (S1) is the covering property ensuring that $\#\varphi^{-1}(\varphi(v)) \geq \alpha_{\varphi(v)} \#\pi^{-1}(\pi(v))$. Now to bound the second term in Eq. (4), let us show a more convenient rewriting of $u \cdot \mathrm{F[G]}(w^0, x)$. Write for shortness for any $u \in T(\mathrm{G}^\star)$ the average $F_u(x) = \sum_{v \in \varphi^{-1}(u)} \mathrm{F[G]}(w^0, x)_v/\#\varphi^{-1}(u)$.

$$\begin{aligned}
u \cdot \mathrm{F[G]}(w^0, x) &\underset{(S1)}{=} \left[\sum_{b \in \mathrm{T}_B} \frac{1}{\sqrt{\#\pi^{-1}(b)}} \sum_{v \in \pi^{-1}(b)} \langle u_{i,v}, \mathrm{F[G]}(w^0, x)_v \rangle \right]_{i \in [k]} \\
&\underset{(S2)}{=} \left[\sum_{b \in \mathrm{T}_B} \sum_{u \in \pi_\star^{-1}(b)} \sum_{v \in \varphi^{-1}(u)} \frac{1}{\sqrt{\#\pi_\star^{-1}(b)}} \frac{\langle a^\star_{i,u}, \mathrm{F[G]}(w^0, x)_v \rangle}{\#\varphi^{-1}(u)} \right]_{i \in [k]} \\
&\underset{(S3)}{=} \left[\sum_{b \in \mathrm{T}_B} \sum_{u \in \pi_\star^{-1}(b)} \frac{1}{\sqrt{\#\pi_\star^{-1}(b)}} \langle a^\star_{i,u}, F_u(x) \rangle \right]_{i \in [k]}
\end{aligned}$$

where (S1) is by defintion of the action, (S2) by definition of $u_v$, and (S3) is by linearity. Next, by *Proposition G.2* and subadditivity, we have $\|F_u(x) - \mathrm{F[G^\star]}(w^\star, x)\| \leq L[\mathrm{G}^\star, w^\star](\eta, \mathcal{X})$. Thus,

$$\begin{aligned}
\|u \cdot \mathrm{F[G]}(w^0, -) - a^\star \cdot \mathrm{F[G^\star]}(w^\star, -)\|_\mathcal{D}^2 &= \mathbb{E}_x \left[\sum_{i \in [k]} \left|\sum_{b \in \mathrm{T}_B} \sum_{u \in \pi_\star^{-1}(b)} \frac{\langle a^\star_{i,u}, F_u - \mathrm{F[G^\star]}(w^\star, x) \rangle}{\sqrt{\#\pi_\star^{-1}(b)}}\right|^2 \right] \\
&\underset{(S1)}{\leq} \mathbb{E}_x \left[\sum_{i \in [k]} \left|\sum_{b \in \mathrm{T}_B} \sum_{u \in \pi_\star^{-1}(b)} \frac{\|a^\star_{i,u}\| \cdot \|F_u - \mathrm{F[G^\star]}(w^\star, x)\|}{\sqrt{\#\pi_\star^{-1}(b)}}\right|^2 \right] \\
&\leq \mathbb{E}_x \left[\sum_{i \in [k]} \left|L[\mathrm{G}^\star, w^\star](\eta, X) \sum_{b \in \mathrm{T}_B} \sum_{u \in \pi_\star^{-1}(b)} \frac{\|a^\star_{i,u}\|}{\sqrt{\#\pi_\star^{-1}(b)}}\right|^2 \right]
\end{aligned}$$

where (S1) is the Cauchy-Schwarz inequality in $\mathcal{Y}_{\pi_\star(u)}$. Thus taking square roots and using the inequality $\|\cdot\|_2 \leq \|\cdot\|_1$, we get $\|u \cdot \mathrm{F[G]}(w^0, -) - a^\star \cdot \mathrm{F[G^\star]}(w^\star, -)\|_\mathcal{D} \leq C^\star \cdot L[\mathrm{G}^\star, w^\star](\eta, \mathcal{X})$. Reinjecting this inequality into the first bound Eq. (4) concludes the proof. $\qquad\square$

# I   COVERAGE OF RANDOM SPARSE LIFTS

**Definition I.1** ($\alpha$ parameter of a perceptron module)**.**
Let $G^\star = (G^\star, \pi_\star, c_\star) \in \mathrm{LiftPMod}(\mathbb{B}, \mathcal{C})$ and $w^\star \in \mathrm{Param}(G^\star)$, with graph $G^\star = (V^\star, E^\star)$.
Let $\lambda : E_B \to \mathbb{R}_+^*$. For $e \in E_B$, let $\mathcal{N}_e$ be a full support distribution on $\mathcal{W}_e$. Let $\eta \in \mathbb{R}_+^*$.

Define $\alpha_\eta : V^\star \to ]0, 1]$, inductively over $v \in V^\star$, from $\alpha_\eta(u) = 1/\#\pi^{-1}(\pi(u))$ for $u \in I(G^\star)$, and
propagated, with the notation $K_v = 2^{1+\#B(-,\pi_\star(v))} \cdot \#\pi_\star^{-1}(\pi_\star(v))$, as follows to $v \in V^\star \setminus I(G^\star)$

$$\alpha_\eta(v) = \frac{1}{K_v} \left( \prod_{a \in B(-,\pi_\star(v))} \mathcal{P}\left(k_a(v); \lambda_{(a,\pi_\star(v))}\right) \right) \left( \prod_{u \in G^\star(-,v)} p_{\pi_\star(u,v)}\left(w_{(u,v)}^\star, \eta\right) \alpha_\eta(u) \right)$$

where $k_a(v) = \#\left(G^\star(-,v) \cap \pi_\star^{-1}(a)\right) \in \mathbb{N}$ is the number of "type-$a$" parents of vertex $v$ in $G^\star$,
$\mathcal{P}(k; \lambda) = e^{-\lambda}\lambda^k/k!$ is the Poisson density, and for $e \in E_B$, we write $p_e : \mathcal{W}_e \times \mathbb{R}_+^* \to \mathbb{R}_+^*$ the
volume of a ball as measured by $\mathcal{N}_e$, i.e. $p_e : (s, \tau) \mapsto \mathbb{P}_{w \sim \mathcal{N}_e}\left(\|w - s\|_{\mathcal{W}_e} \leq \tau\right)$.

Validity of this definition (e.g. image of the function $\alpha$) is detailed in Appendix K.2. The form of
$\alpha$ at vertex $v$ (forgetting $K_v$) is intuitively the probability of guessing the correct degree, then the
probability that for each parent, both the weight and the corresponding subnetwork match the witness.

Based on the results of Appendix E, we know that it is sufficient for convergence to show tangent
approximation on a large ball around initialization. For a given lift and choice of parameters, we
now know from Appendix H.2 that it is sufficient to show existence of a witness extraction (a partial
morphism such that weights approximatively match) on that ball. Let us start by showing that at the
initial point, there exists a witness extraction with high probability, we will extend this to balls later.

**Proposition I.2** (Large random sparse lifts are covering with high probability)**.**
Let $\lambda : E_B \to \mathbb{R}_+^*$. For $e \in E_B$, let $\mathcal{N}_e$ be a distribution on $\mathcal{W}_e$ with full support.

Let $G^\star = (G^\star, \pi_\star, c_\star) \in \mathrm{LiftPMod}(\mathbb{B}, \mathcal{C})$ and $w^\star \in \mathrm{Param}(G^\star)$, over graph $G^\star = (V^\star, E^\star)$.
Let $\eta \in \mathbb{R}_+$. Define $\alpha_\eta : V^\star \to ]0, 1]$ the $\alpha$-parameter associated to $(G^\star, w^\star)$ by Definition I.1

Let $n : V_B \to \mathbb{N}$ such that if $b \in I_B$ then $n_b = \#p_\mathcal{C}^{-1}(b)$. Let $(G, w)$ be a random sparse lift of $\mathbb{B}$
along $(n, \lambda)$ with distribution $\mathcal{N}$. If for all $(a, b) \in E_B$, $n_a \geq 2\max(\lambda_{(a,b)}, \lambda_{(a,b)}^2/\log(2))$, then

$$\mathbb{P}\left(\mathcal{M}_\eta^{\alpha_\eta}\left[(G, w), (G^\star, w^\star)\right]\right) \geq \prod_{b \in V_B \setminus I_B} \left(1 - \sum_{v \in \pi_\star^{-1}(b)} \exp\left(-\frac{\tau(v)^2}{4} n_b \cdot \alpha(v)\right)\right)_+$$

where $\tau : V^\star \to [0, 1]$ is $\tau : v \mapsto \left(1 - \frac{1}{n_{\pi_\star(v)}\alpha(v)}\right)_+$ which tends to 1 when $\left(n_{\pi_\star(v)}\alpha(v)\right) \to +\infty$.

We will write $G_a^\star(-,v) = G^\star(-,v) \cap \pi_\star^{-1}(a)$ with $a \in V_B$ in index because this will appear often
in this proof. Similarly for the graph $G$, we write $G_a(-,v) = G(-,v) \cap \pi^{-1}(a)$ for $v \in V$.

For any vertex $v \in V_B$, define the set $A_v = \{u \in V_B, u \preceq v\}$ (of "ancestors" of $v$). Define the
filtration $(\mathcal{F}_b)_{b \in V_B}$ induced by the strict total order $\prec$ on $V_B$ as the $\sigma$-algebra

$$\mathcal{F}_b = \sigma\left( \bigcup_{\substack{(x,y) \in E_B \\ y \prec b}} \bigcup_{\substack{e \in E_C \\ \pi(e)=(x,y)}} \{m_e, w_e\} \right)$$

Conditioning on the $\sigma$-algebra $\mathcal{F}_b$ will "freeze" the random variables corresponding to the subset
$\{v \in V \mid \pi(v) \in A_b \setminus \{b\}\}$, and allow the study in isolation of what happens when adding $b \in V_B$.

*Proof of Proposition I.2.* We will proceed by induction on $V_B$, the vertices of the base graph, using a
strict total topological order $\prec$ on $V_B$ (i.e. such that $(u, v) \in E_B \Rightarrow u \prec v$), available by acyclicity.

We use the notations from Definition 4.2 of random sparse lifts, with the bernoulli random variables
$m_{(u,v)} \sim \mathcal{B}(\lambda_{\pi(u,v)}/n_{\pi(u)})$ and weights $w_{(u,v)} \sim \mathcal{N}_{\pi(u,v)}$ for all $(u, v) \in E_C$, where $C = (V_C, E_C)$
is the fully-connected lift of $B$ along $n$ (Definition 3.6) to define the random sparse lift $(G, w, a)$.

**Induction hypothesis** . For any subset $U \subseteq V_B$, define $\mathcal{M}(U)$ the set of $\mathrm{LiftPMod}(\mathbb{B}, \mathcal{C})$-partial-morphism $(S, \varphi) : G \to G^\star$ such that $V_S \subseteq \pi^{-1}(U)$, and the following two conditions hold:

$$\forall e \in E(S), \quad \|w_e - w^\star_{\varphi(e)}\|_{\mathcal{W}_{\pi(e)}} \leq \eta$$
$$\forall v \in \pi_\star^{-1}(U), \quad \#\varphi^{-1}(v) \geq \alpha(v) \cdot \#\pi^{-1}\left(\pi_\star(v)\right)$$

The first condition is identical to the definition of covering matchings, the second is quantified differently (on $\pi_\star^{-1}(U)$ not $V^\star$), but coincides with the definition when $U = V_B$.

By contradiction, if the claimed inequality does not hold, then let $b \in V_B$ be the minimal vertex (for the $\prec$ topological order) such that the following inequality holds

$$\mathbb{P}\left(\mathcal{M}(A_b) \neq \varnothing\right) \;<\; \prod_{a \in A_b \setminus I_B} \left(1 - \sum_{v \in \pi_\star^{-1}(a)} \exp\left(-\frac{\tau(v)^2}{4} n_a \cdot \alpha(v)\right)\right)_+ \tag{5}$$

Let us show a contradiction (thus that there is no vertex such that this condition holds, thus the inverse inequality holds for any $b$ and in particular it holds for $\mathcal{M}(V_B)$ which matches the claim exactly).

**Induction intialization.** Let us proceed by case disjunction. If $b \in I_B$ first, then let us show a contradiction. First, to tackle the edge case, note that $\mathcal{M}(\varnothing) \neq \varnothing$ almost surely because it contains $(S, \varphi)$ a partial morphism with $S$ the empty graph (no vertices and no edges). Then, let us show that $\mathbb{P}(\mathcal{M}(A_b) \neq \varnothing) \geq \mathbb{P}(\mathcal{M}(A_b \setminus \{b\}) \neq \varnothing)$ by the almost-sure extension that follows. Recall for that matter that the function $p_{\mathcal{C}} : \mathcal{C} \to I_B$ partitions $\mathcal{C}$ into subsets $p_{\mathcal{C}}^{-1}(u) \subseteq \mathcal{C}$ indexed by $u \in I_B$. Moreover, by construction of a random sparse lift, for any $u \in I_B$, we have $\pi^{-1}(u) = p_{\mathcal{C}}^{-1}(u)$. Thus, if $(S, \varphi) \in \mathcal{M}(A_b \setminus \{b\})$ for a graph $S = (V_S, E_S)$, construct the extension $((V_S^+, E_S), \varphi^+)$ as $V_S^+ = V_S \cup c_\star(\pi_\star^{-1}(b))$ with $\varphi^+|_{V_S} = \varphi$ and for $v \in V_S^+ \setminus V_S$, let $\varphi^+(v) = c_\star^{-1}(v) \in \pi_\star^{-1}(b)$. This construction is well-defined by injectivity of $c_\star : \pi_\star^{-1}(I_B) \to \mathcal{C}$. It is also immediately verified that $\varphi^+$ is a fibration because elements of $V_S^+ \setminus V_S$ have no parents (more formally, $v \in V_S^+ \setminus V_S \Rightarrow \pi(v) = b \Rightarrow G(-, v) = \varnothing$). The homomorphism $\pi_S : V_S^+ \to V_B$ is obtained by restriction $\pi_S = \pi|_{V_S^+}$. Additionally, for every $u \in \pi_\star^{-1}(u)$, we have $\#(\varphi^+)^{-1}(u) = 1 = \alpha(u) \cdot \#\pi_\star^{-1}(\pi_\star(u)) = \alpha(u) \cdot \#\pi_S^{-1}(\pi_\star(u))$ by definition of $\alpha$. Thus $((V_S^+, E_S), \varphi^+) \in \mathcal{M}(A_b)$. This concludes the proof that $\mathbb{P}(\mathcal{M}(A_b) \neq \varnothing) \geq \mathbb{P}(\mathcal{M}(A_b \setminus \{b\}) \neq \varnothing)$. Let us now show why this contradicts the definition of $b$. If $A_b \setminus \{b\} = \varnothing$, then the inequality Eq. (5) contradicts $\mathbb{P}(\mathcal{M}(\varnothing) \neq \varnothing) = 1$. If on the other hand there is $a \in A_b$ such that $a \prec b$, then $a$ must also satisfy the condition Eq. (5), which contradicts minimality of $b$. This concludes the case $b \in I_B$.

**Induction propagation by morphism extension.** The case $b \in I_B$ has been tackled. Therefore let us assume for the remainder of the proof that $b \in V_B \setminus I_B$. The idea is roughly the same as for the previous case, we will work conditionally on $\mathcal{F}_b$ and show that we can extend partial morphisms from $\mathcal{M}(A_b \setminus \{b\})$ to $\mathcal{M}(A_b)$ with a high-enough probability, and use it to form a contradiction.

Intuitively, if $(S, \varphi) \in \mathcal{M}(A_b \setminus \{b\})$, we want to construct for every $i \in \pi_\star^{-1}(b)$ a set $E_i(\varphi) \subseteq \pi^{-1}(b)$ such that $j \in E_i(\varphi)$ if we can extend $\varphi$ into a partial fibration by mapping $j$ to $i$, and if this extension continues to satisfy both the type conditions and the weight-matching condition. We do so by considering parents one at a time, then take intersections, and finally show that $E_i(\varphi)$ is sufficiently large with a sufficiently high probability.

**Extension (a): single-parent compatibility.** For every $(S, \varphi) \in \mathcal{M}(A_b \setminus \{b\})$ a partial morphism with $S = (V_S, E_S)$, and every $a \in B(-, b)$, let $E_i^a(\varphi) \subseteq \pi^{-1}(b)$ be the subset of $V$ defined as

$$j \in E_i^a(\varphi) \;\Leftrightarrow\; \begin{cases} G_a(-, j) \subseteq V_S \\ \varphi(G_a(-, j)) = G_a^\star(-, i) \\ \forall u \in G_a(-, j), \; \|w_{(u,j)} - w^\star_{(\varphi(u), i)}\| \leq \eta \end{cases}$$

With the previous notation of $\mathcal{P}(k; \lambda) = e^{-\lambda} \lambda^k / k!$ and $p_e(w, \eta) = \mathbb{P}_{u \sim \mathcal{N}_e}(\|u - w\| \leq \eta)$, define

$$\mu_{i,a} = \frac{1}{2} \mathcal{P}\left(\#G_a^\star(-, i); \lambda_{(a,b)}\right) \prod_{u \in G_a^\star(-, i)} p_{(a,b)}\left(w^\star_{(u,i)}, \eta\right) \cdot \alpha(u)$$

Observe that this definition is chosen such that $\alpha(i) = \frac{1}{2}\left(\prod_{a \in B(-,b)} \mu_{i,a}\right)/\#\pi_\star^{-1}(\pi_\star(i))$.

Let us show that $\mathbb{P}\left(j \in E_i^a(\varphi) \mid (S, \varphi) \in \mathcal{M}(A_b \setminus \{b\}), \mathcal{F}_b\right) \geq \mu_{i,a}$. For shortness in the writing of this proof, let $k = \#G_a^\star(-,i) \in \mathbb{N}$ and $q_{(a,b)} = \lambda_{(a,b)}/n_a \in [0,1]$. For a subset $I \subseteq \pi^{-1}(a) \subseteq V_G$ such that $\#I = k$, write $Z(I)$ the set of bijections from $I$ to $G_a^\star(-,i)$. Now, observe that by disjunction over such subsets $I$ and average over $Z(I)$, it holds

$$\mathbb{1}_{j \in E_i^a(\varphi)} \geq \sum_{\substack{I \subseteq \pi^{-1}(a) \\ \#I = k}} \mathbb{1}_{I = G_a(-,j)} \mathbb{1}_{I \subseteq V_S} \frac{1}{\#Z(I)} \sum_{\zeta \in Z(I)} \mathbb{1}_{\varphi|_I = \zeta} \prod_{u \in I} \mathbb{1}_{\|w_{(u,j)} - w_{(\zeta(u),i)}^\star\| \leq \eta}$$

By independence and linearity, we can take the conditional expectation

$$\mathbb{E}\left[\mathbb{1}_{j \in E_i^a(\varphi)} \mid (S, \varphi) \in \mathcal{M}(A_b \setminus \{b\}), \mathcal{F}_b\right]$$

$$\underset{(S1)}{\geq} \sum_{\substack{I \subseteq \pi^{-1}(a) \\ \#I = k}} \mathbb{1}_{I \subseteq V_S} \cdot q_{(a,b)}^k \left(1 - q_{(a,b)}\right)^{n_a - k} \frac{1}{k!} \sum_{\zeta \in Z(I)} \mathbb{1}_{\varphi|_I = \zeta} \prod_{u \in I} p_{(a,b)}\left(w_{(\zeta(u),i)}^\star, \eta\right)$$

$$\underset{(S2)}{\geq} q_{(a,b)}^k \left(1 - q_{(a,b)}\right)^{n_a - k_{i,a}} \frac{1}{k!} \sum_{\substack{I \subseteq \pi^{-1}(a) \\ \#I = k}} \mathbb{1}_{I \subseteq V_S} \cdot \sum_{\zeta \in Z(I)} \mathbb{1}_{\varphi|_I = \zeta} \prod_{u \in I} p_{(a,b)}\left(w_{(\zeta(u),i)}^\star, \eta\right)$$

$$\underset{(S3)}{\geq} \frac{1}{2} \frac{\lambda_{(a,b)}^k}{k!} e^{-\lambda_{(a,b)}} \frac{1}{n_a^k} \sum_{\substack{I \subseteq \pi^{-1}(a) \\ \#I = k}} \mathbb{1}_{I \subseteq V_S} \cdot \sum_{\zeta \in Z(I)} \mathbb{1}_{\varphi|_I = \zeta} \prod_{u \in I} p_{(a,b)}\left(w_{(\zeta(u),i)}^\star, \eta\right)$$

$$\underset{(S4)}{\geq} \frac{1}{2} \frac{\lambda_{(a,b)}^k}{k!} e^{-\lambda_{(a,b)}} \prod_{u \in G_a^\star(-,i)} p_{(a,b)}\left(w_{(u,i)}^\star, \eta\right) \cdot \frac{1}{n_a^k} \sum_{\substack{I \subseteq \pi^{-1}(a) \\ \#I = k}} \mathbb{1}_{I \subseteq V_S} \cdot \sum_{\zeta \in Z(I)} \mathbb{1}_{\varphi|_I = \zeta}$$

$$\underset{(S5)}{\geq} \frac{1}{2} \frac{\lambda_{(a,b)}^k}{k!} e^{-\lambda_{(a,b)}} \prod_{u \in G_a^\star(-,i)} p_{(a,b)}\left(w_{(u,i)}^\star, \eta\right) \cdot \alpha(u)$$

$$= \mu_{i,a}$$

where (S1) is linearity of the expectation and independence of $m$ and $w$, (S2) is a factorization of constants related to $m$, (S3) is the lower-bound of Proposition L.1 because by assumption $n_a \geq \max(2\lambda_{(a,b)}, 2\lambda_{(a,b)}^2/\log(2))$, (S4) is a factorization of constants related to $w$, and (S5) is the following counting trick:

$$\frac{1}{n_a^k} \sum_{\substack{I \subseteq \pi^{-1}(a) \\ \#I = k}} \mathbb{1}_{I \subseteq V_S} \cdot \sum_{\zeta \in Z(I)} \mathbb{1}_{\varphi|_I = \zeta} \underset{(S1)}{\geq} \frac{1}{n_a^k} \sum_{\substack{\psi: G_a^\star(-,i) \to V_S \\ \forall u, \psi(u) \in \varphi^{-1}(u)}} 1$$

$$\underset{(S2)}{\geq} \frac{1}{n_a^k} \prod_{u \in G_a^\star(-,i)} \left(\#\varphi^{-1}(u)\right)$$

$$\underset{(S3)}{\geq} \frac{1}{n_a^k} \prod_{u \in G_a^\star(-,i)} \left(\alpha(u) \cdot n_a\right)$$

$$\geq \prod_{u \in G_a^\star(-,i)} \alpha(u)$$

where (S1) is a lower-bound by exhibition of a subset of terms satisfying the indicator conditions, (S2) is the counting of such terms, and (S3) is the hypothesis $(S, \varphi) \in \mathcal{M}(A_b \setminus \{b\})$ with the $\alpha$-cover property applied to $a \in B(-,b) \subseteq (A_b \setminus \{b\})$.

This concludes the proof of the inequality $\mathbb{E}\left[\mathbb{1}_{j \in E_i^a(\varphi)} \mid (S, \varphi) \in \mathcal{M}(A_b \setminus \{b\}), \mathcal{F}_b\right] \geq \mu_{i,a}$.

**Extension (b): multi-parent compatibility.** Define now $E_i(\varphi) = \bigcap_{a \in B(-,b)} E_i^a(\varphi)$. By conditional independence given $\mathcal{F}_b$ of the events $(j \in E_i^a(\varphi))_a$ indexed by $a \in B(-,b)$, we get

$$\mathbb{E}\left[\mathbb{1}_{j \in E_i(\varphi)} \,\middle|\, (S,\varphi) \in \mathcal{M}(A_b \setminus \{b\}), \mathcal{F}_b\right] = \mathbb{E}\left[\prod_{a \in B(-,b)} \mathbb{1}_{j \in E_i^a(\varphi)} \,\middle|\, (S,\varphi) \in \mathcal{M}(A_b \setminus \{b\}), \mathcal{F}_b\right]$$

$$= \prod_{a \in B(-,b)} \mathbb{E}\left[\mathbb{1}_{j \in E_i^a(\varphi)} \,\middle|\, (S,\varphi) \in \mathcal{M}(A_b \setminus \{b\}), \mathcal{F}_b\right]$$

$$\geq \prod_{a \in B(-,b)} \mu_{i,a}$$

Write for shortness $\mu_i = \prod_{a \in B(-,b)} \mu_{i,a}$, recall that $\alpha(i) = \frac{1}{2}\mu_i/\#\pi_\star^{-1}(b)$, and equipped with the inequality $\mathbb{E}\left[\mathbb{1}_{j \in E_i(\varphi)} \,\middle|\, (S,\varphi) \in \mathcal{M}(A_b \setminus \{b\}), \mathcal{F}_b\right] \geq \mu_i$, let us show that $E_i(\varphi)$ is large enough.

**Extension (c): probability amplification.** The events $(j \in E_i(\varphi))_j$ indexed by $j \in \pi^{-1}(b)$ are independent conditionally on $\mathcal{F}_b$. Thus the random variables $X_j = \mathbb{1}_{j \in E_i(\varphi)}$ for $j \in \pi^{-1}(b)$ have a Bernoulli distribution and conditional expectation

$$\mathbb{P}\left(j \in E_i(\varphi) \,\middle|\, (S,\varphi) \in \mathcal{M}(A_b \setminus \{b\}), \mathcal{F}_b\right) \geq \mu_i > 0$$

Let $\delta_i = \frac{1}{2}\tau(i) \in [0,1]$. Since $\#E_i(\varphi) = \sum_j X_j$, we get by amplification (Appendix-Lemma L.4)

$$\mathbb{P}\left(\#E_i(\varphi) \leq (1-\delta_i)\mu_i \cdot n_b \,\middle|\, (S,\varphi) \in \mathcal{M}(A_b \setminus \{b\}), \mathcal{F}_b\right)$$

$$\leq \exp\left(-\frac{1}{2}\delta_i^2 n_b \cdot \mu_i\right) = \exp\left(-\frac{\tau(i)^2}{8} n_b \cdot \mu_i\right)$$

$$\leq \exp\left(-\frac{\tau(i)^2}{4} n_b \cdot \alpha(i)\right)$$

We can then get a lower-bound on the probability that all candidate preimage sets $E_i(\varphi)$ are sufficiently large, by union bound (since their cardinals are not independent)

$$\mathbb{E}\left[\prod_{i \in \pi_\star^{-1}(b)} \mathbb{1}_{\#E_i(\varphi) \geq (1-\delta_i)n_b\mu_i} \,\middle|\, (S,\varphi) \in \mathcal{M}(A_b \setminus \{b\}), \mathcal{F}_b\right]$$

$$\geq 1 - \sum_{i \in \pi_\star^{-1}(b)} \mathbb{E}\left[\mathbb{1}_{\#E_i(\varphi) < (1-\delta_i)n_b\mu_i} \,\middle|\, (S,\varphi) \in \mathcal{M}(A_b \setminus \{b\}), \mathcal{F}_b\right]$$

$$\geq 1 - \sum_{i \in \pi_\star^{-1}(b)} \exp\left(-\frac{\tau(i)^2}{4} n_b \cdot \alpha(i)\right)$$

We will use positive parts $(\cdot)_+$ in the following to handle nicely the case where this is strictly negative.

**Extension (d): tackling overlap.** Let us start by showing that for any partial morphism $(S,\varphi)$,

$$\mathbb{1}_{\mathcal{M}(A_b) \neq \varnothing} \geq \mathbb{1}_{(S,\varphi) \in \mathcal{M}(A_b \setminus \{b\})} \cdot \prod_{i \in \pi_\star^{-1}(b)} \mathbb{1}_{\#E_i(\varphi) \geq (1-\delta_i)n_b\mu_i} \tag{6}$$

If $(S,\varphi) \in \mathcal{M}(A_b \setminus \{b\})$ and if for every $i \in \pi_\star^{-1}(b)$ it holds $\#E_i(\varphi) \geq (1-\delta_i)n_b\mu_i$ then let us construct an extension $(S^+, \varphi^+) \in \mathcal{M}(A_b)$. The difficulty is that the sets $(E_i(\varphi))_i$ may not be disjoint, thus we must select disjoint subsets of sufficient size to satisfy the covering condition. Choose for every $i \in \pi_\star^{-1}(b)$ a subset $V_i \subseteq E_i(\varphi)$ such that $(V_i)_i$ are disjoint, and $\#V_i \geq \frac{1}{2}n_b\mu_i/\#\pi_\star^{-1}(b)$.

Proof that such choice is possible is given in Proposition L.6, since if $n_b\alpha(i) \geq 1$, for $P = \#\pi_\star^{-1}(b)$,

$$\left\lfloor \frac{\#E_i(\varphi)}{P} \right\rfloor \geq (1-\delta_i)\frac{n_b\mu_i}{P} - 1 = \left(\frac{1}{2} + \frac{1}{2n_b\alpha_i}\right)\frac{n_b\mu_i}{P} - 1 = \frac{n_b\mu_i}{2P} = n_b\alpha(i)$$

And if $n_b\alpha(i) < 1$, then Eq. (5) is immediately contradicted because the right-hand side is zero.

Then, let $V_S{}^+ = \bigcup_i V_i$ be the extended vertex set and $E_S{}^+ = \bigcup_{v \in V_S{}^+} \bigcup_{u \in G(-,v)} (u,v)$ the corresponding edges. Define $((V_S \cup V_S{}^+, E_S \cup E_S{}^+), \varphi^+)$ the morphism obtained as $\varphi^+|_{V_S} = \varphi$ and for $v \in V_i \subseteq V_S{}^+$ as $\varphi^+(v) = i$. By construction of $(V_i)_i$, this morphism is a fibration of graphs. Since for all $i \in \pi_\star^{-1}(b)$, we have $\#(\varphi^+)^{-1}(i) = \#V_i \geq \frac{1}{2} n_b \mu_i / \#\pi_\star^{-1}(b) = n_b \alpha(i)$, thus the covering condition is satisfied, and therefore $((V_S \cup V_S{}^+, E_S \cup E_S{}^+), \varphi^+) \in \mathcal{M}(A_b)$, which conclues the proof of Eq. (6).

**Towering.**    We can now proceed by using the property that $\mathbb{1}_{\mathcal{M}(A_b) \neq \varnothing} = \sup_{(S,\varphi)} \mathbb{1}_{(S,\varphi) \in \mathcal{M}(A_b)}$.

$$
\begin{aligned}
\mathbb{E}\left[\mathbb{1}_{\mathcal{M}(A_b) \neq \varnothing} \,\middle|\, \mathcal{F}_b\right] &\geq \mathbb{E}\left[\sup_{(S,\varphi)} \mathbb{1}_{(S,\varphi) \in \mathcal{M}(A_b \setminus \{b\})} \cdot \prod_{i \in \pi_\star^{-1}(b)} \mathbb{1}_{\#E_i(\varphi) \geq (1-\delta_i) n_b \mu_i} \,\middle|\, \mathcal{F}_b\right] \\
&\geq \sup_{(S,\varphi)} \mathbb{E}\left[\mathbb{1}_{(S,\varphi) \in \mathcal{M}(A_b \setminus \{b\})} \cdot \prod_{i \in \pi_\star^{-1}(b)} \mathbb{1}_{\#E_i(\varphi) \geq (1-\delta_i) n_b \mu_i} \,\middle|\, \mathcal{F}_b\right] \\
&= \sup_{(S,\varphi)} \mathbb{1}_{(S,\varphi) \in \mathcal{M}(A_b \setminus \{b\})} \cdot \mathbb{E}\left[\prod_{i \in \pi_\star^{-1}(b)} \mathbb{1}_{\#E_i(\varphi) \geq (1-\delta_i) n_b \mu_i} \,\middle|\, \mathcal{F}_b\right] \\
&\geq \sup_{(S,\varphi)} \mathbb{1}_{(S,\varphi) \in \mathcal{M}(A_b \setminus \{b\})} \cdot \left(1 - \sum_{i \in \pi_\star^{-1}(b)} \exp\left(-\frac{\tau(i)^2}{4} n_b \cdot \alpha(i)\right)\right)_+ \\
&\geq \mathbb{1}_{\mathcal{M}(A_b \setminus \{b\}) \neq \varnothing} \cdot \left(1 - \sum_{i \in \pi_\star^{-1}(b)} \exp\left(-\frac{\tau(i)^2}{4} n_b \cdot \alpha(i)\right)\right)_+
\end{aligned}
$$

Finally, we can conclude by a towering argument

$$
\begin{aligned}
\mathbb{E}\left[\mathbb{1}_{\mathcal{M}(A_b) \neq \varnothing}\right] &= \mathbb{E}\left[\mathbb{E}\left[\mathbb{1}_{\mathcal{M}(A_b) \neq \varnothing} \,\middle|\, \mathcal{F}_b\right]\right] \\
&\geq \mathbb{E}\left[\left(1 - \sum_{i \in \pi_\star^{-1}(b)} \exp\left(-\frac{\tau(i)^2}{4} n_b \cdot \alpha(i)\right)\right)_+ \cdot \mathbb{1}_{\mathcal{M}(A_b \setminus \{b\}) \neq \varnothing}\right] \\
&= \left(1 - \sum_{i \in \pi_\star^{-1}(b)} \exp\left(-\frac{\tau(i)^2}{4} n_b \cdot \alpha(i)\right)\right)_+ \cdot \mathbb{E}\left[\mathbb{1}_{\mathcal{M}(A_b \setminus \{b\}) \neq \varnothing}\right] \\
&\underset{(S1)}{\geq} \prod_{a \in A_b \setminus I_B} \left(1 - \sum_{i \in \pi_\star^{-1}(a)} \exp\left(-\frac{\tau(i)^2}{4} n_a \cdot \alpha(i)\right)\right)_+
\end{aligned}
$$

where (S1) is the reverse of inequality Eq. (5) for any $a \in V_B$ with $a \prec b$ by minimality of $b$. This contradicts the inequality Eq. (5) defining $b$, thus concludes the case $b \in V_B \setminus I_B$ and the induction, which concludes the proof. $\qquad\square$

## I.1 BALLWISE RESISTANCE OF COVERINGS WITH LOW OUT-DEGREE

Recall that we write $n^0 : I_B \to \mathbb{N}$ the function $n^0 : b \mapsto \#p_C^{-1}(b)$.

**Proposition I.3.**
*Let $\lambda : E_B \to \mathbb{R}_+^*$. For $e \in E_B$, let $\mathcal{N}_e$ be a distribution on $\mathcal{W}_e$ with full support.*

*Let $G^\star = (G^\star, \pi_\star, c_\star) \in \mathrm{LiftPMod}(\mathbb{B}, \mathcal{C})$ and $w^\star \in \mathrm{Param}(G^\star)$, over graph $G^\star = (V^\star, E^\star)$.*
*Let $\eta \in \mathbb{R}_+$. Define $\alpha_\eta : V^\star \to \,]0, 1]$ the $\alpha$-parameter associated to $(G^\star, w^\star)$ and $\eta$ by Def. I.1.*

*Let $n : V_B \to \mathbb{N}$ such that if $b \in I_B$ then $n_b = n_b^0$. Let $(G, w^0)$ be a random sparse lift of $\mathbb{B}$ along $(n, \lambda)$ with distribution $\mathcal{N}$.*

*Assume that for all $(a, b) \in E_B$, if $a \in I_B$ then $\lambda_{(a,b)} \leq \min\{n_a^0/2, (n_a^0/3)^{1/2}\}$, and that*

$$\forall b \in V_B \setminus I_B, \; n_b \geq 8 \, \frac{(R+\eta)^2}{\eta^2} \, \frac{(1+\Lambda)^{1+\#V_B}}{\inf_{v \in V^\star} \alpha_{\eta/2}(v)}$$

*where $\Lambda = \sum_{(a,b) \in E_B} \left( 7\, \lambda_{(a,b)} + 1 + \log(2) + \log(\#V^\star + \#E_B) - \log \delta \right)$*

*Then,*

$$\mathbb{P}\Big( \forall w \in B(w^0, R), \; \mathcal{M}_\eta^{\frac{1}{2}\alpha_{\eta/2}} [(G, w), (G^\star, w^\star)] \neq \varnothing \Big) \geq 1 - \delta$$

The proof is deferred to Appendix I.1.2. Together with Proposition H.4, it should be straightforward to see how this proposition will be used to prove the tangent approximation property on an $R$-ball.

### I.1.1 DECOMPOSITION OF BALLWISE COVER VIA LOW-DEGREE PROPERTIES

**Lemma I.4** (Resistance of cover in low out-degree graphs). *Let $\eta \in \mathbb{R}_+^*$ and $R \in \mathbb{R}_+$.*
*Let $G^\star = (G^\star, \pi_\star, c_\star) \in \mathrm{LiftPMod}(\mathbb{B}, \mathcal{C})$ and $w^\star \in \mathrm{Param}(G^\star)$ over graph $G^\star = (V^\star, E^\star)$.*
*Let $\alpha : V^\star \to \,]0, 1]$ and let $G = (G, \pi, c) \in \mathrm{LiftPMod}(\mathbb{B}, \mathcal{C})$ and $w^0 \in \mathrm{Param}(G)$.*

*Write $n : V_B \to \mathbb{N}$ and $d : E_B \to \mathbb{N}$ (vertex count and maximal out-degree of $G$) defined as*

$$n : b \in V_B \mapsto \#\pi^{-1}(b) \quad and \quad d : (a,b) \in E_B \mapsto \sup_{v \in \pi^{-1}(a)} \# \left( G(v, -) \cap \pi^{-1}(b) \right)$$

*Assume that $\forall (a,b) \in E_B, n_b \geq n_a \log n_a$, and that there is a constant $\Lambda \in \mathbb{R}_+$ such that*

$$\left[ \sup_{b \in V_B} \sum_{a \in B(-,b)} d_{(a,b)} \frac{n_a}{n_b} \right] \leq \Lambda \quad and \quad \left[ \inf_{b \in V_B \setminus I_B} n_b \right] \geq \frac{2R^2}{\eta^2} \frac{(1+\Lambda)^{\#V_B}}{\inf_{v \in V^\star} \alpha_v}$$

*Then for every $w \in \mathrm{Param}(G)$ such that $\|w - w^0\| \leq R$ it holds*

$$\mathcal{M}_\eta^\alpha [(G, w^0), (G^\star, w^\star)] \neq \varnothing \quad \Rightarrow \quad \mathcal{M}_{2\eta}^{\alpha/2} [(G, w), (G^\star, w^\star)] \neq \varnothing$$

*Proof of Lemma I.4.* Under the assumption that this set is non-empty, choose a partial morphism $(S, \varphi) \in \mathcal{M}_\eta^\alpha \left[ (G, w^0), (G^\star, w^\star) \right]$ and let us construct an an element of $\mathcal{M}_{2\eta}^{\alpha/2} [(G, w), (G^\star, w^\star)]$.

Write $S = (V_S, E_S)$. We will show that we can extract a subgraph $U = (V_U, E_U)$ of $S$ such that the inclusion $\iota_U : U \to S$ is a fibration, and the restriction of $\varphi$ to $U$ preserves weights (up to $2\eta$).

We proceed by induction on $b \in V_B$, by defining $U_b \subseteq V_S \cap \pi^{-1}(b)$ as follows:

$$U_b = \left\{ v \in \pi^{-1}(b) \cap V_S \;\middle|\; G(-, v) \subseteq \bigcup_{\substack{a \in V_B \\ a \prec b}} U_a, \; \sup_{u \in G(-,v)} \|w_u^0 - w_u\| \leq \eta \right\}$$

In words, $U_b$ is the set of vertices of $S$ selected as follows: a vertex is selected if all its parents have been selected already, and weights on links to its parents have moved no more than $\eta$ from $w^0$ to $w$.

The extracted graph is then $U = (V_U, E_U)$, where $V_U = \bigcup_{b \in V_B} U_b$, and $E_U = E_S \cap (V_U \times V_U)$. First, let us check that the inclusion $\iota_U : U \to S$ is a fibration, and that the weight-approximation

condition is satisfied. It is immediately verified that it is an isomorphism of graphs. Moreover, by definition of $U$, if $v \in V_U$ then $U(-, v) = G(-, v) = S(-, v)$, therefore $\iota_U$ is an isomorphism of in-neighborhoods, thus a fibration. To check the weight-approximation condition, let $e \in E_U$, and observe that it holds by triangle inequality in $\mathcal{W}_{\pi(e)}$ that

$$\|(\iota^* w)_e - (\varphi^* w^\star)_e\| = \|w_e - w_{\varphi(e)}^\star\| \leq \|w_e - w_e^0\| + \|w_e^0 - w_{\varphi(e)}^\star\| \leq \eta + \eta = 2\eta$$

Thus, it only remains to show that the restriction $\varphi|_U : V_U \to V^\star$ has sufficient volume. This is a little more tedious. We will switch points of view, to count the vertices from $V_S$ that we reject instead of those that we keep. Define (again inductively on $b \in V_B$):

$$C_b = \left\{ v \in \pi^{-1}(b) \cap V_S \ \middle| \ \exists u \in G(-, v), u \notin \bigcup_{\substack{a \in V_B \\ a \prec b}} U_a \right\}$$

$$W_b = \left\{ v \in \pi^{-1}(b) \cap V_S \ \middle| \ \exists u \in G(-, v), \|w_{(u,v)}^0 - w_{(u,v)}\| > \eta \right\}$$

In words, $W_b$ is the set of vertices in $V_S$ of "type $b$" that have been rejected because one the corresponding weights had moved to much. Similarly, $C_b$ is the set of vertices in $V_S$ that have been rejected because one of their parents had been rejected ($C$ stands for "chain" rejection). Observe that

$$U_b = \left(\pi^{-1}(b) \cap V_S\right) \setminus (C_b \cup W_b)$$

Therefore $\#U_b \geq \# \left(\pi^{-1}(b) \cup V_S\right) - \#C_b - \#W_b$. Let us bound each term separately.

First, note that if $v \in I(G) \subseteq V$, and $v \in V_S$, then $v \in U_{\pi(v)}$ because $G(-, v) = \varnothing$.

Regarding $\#W_b$, since we know by assumption $\sum_{e \in E_S} \|w_e^0 - w_e\|^2 \leq \sum_{e \in E} \|w_e^0 - w_e\|^2 \leq R^2$, but also by definition of $W_b$ that $\#W_b \cdot \eta^2 \leq \sum_{e \in E_S} \|w_e^0 - w_e\|^2$, we can deduce that $\#W_b \leq R^2/\eta^2$.

Regarding $\#C_b$, we will rely on the following forward inclusion:

$$C_b \subseteq \bigcup_{a \in B(-,b)} \bigcup_{v \in C_a \cup W_a} G(v, -)$$

therefore we can leverage outgoing degree bounds to obtain

$$\#C_b \leq \sum_{a \in B(-,b)} (\#C_a + \#W_a) \cdot d_{(a,b)}$$

At this point, it becomes easier to stop thinking in integers and cardinals, and to switch to rates. Define $r : V_B \to [0, 1]$ the "rejection rate" $r : a \mapsto (\#C_a + \#W_a)/n_a$. We have shown so far that if $b \in I_B$, then $r_b = 0$. Then, dividing the previous inequality by $n_b$ on both sides, and rewriting

$$r_b - \frac{\#W_b}{n_b} = \frac{\#C_b}{n_b} \leq \sum_{a \in B(-,b)} \frac{\#C_a + \#W_a}{n_a} \cdot \frac{n_a}{n_b} \cdot d_{(a,b)} = \sum_{a \in B(-,b)} r_a \cdot d_{(a,b)} \frac{n_a}{n_b}$$

Write $s = \frac{1}{2} \inf \alpha / (1 + \Lambda)^{\#V_B}$. Using the second assumption, that $n_b \geq \frac{R^2}{\eta^2} \cdot \frac{1}{s}$, and the first assumption $\sum_{(a,b) \in E_B} d_{(a,b)} \frac{n_a}{n_b} \leq \Lambda$, we can deduce the following bound for $r$

$$r_b \leq \frac{\#W_b}{n_b} + \sum_{a \in B(-,b)} r_a \cdot d_{(a,b)} \frac{n_a}{n_b} \leq \frac{R^2}{\eta^2} \frac{1}{n_b} + \left(\sum_{a \in B(-,b)} d_{(a,b)} \frac{n_a}{n_b}\right) \cdot \left(\max_{a \in B(-,b)} r_a\right)$$

$$r_b \leq s + \Lambda \cdot \left(\max_{a \in B(-,b)} r_a\right)$$

Hence by Appendix-Lemma L.5, we get $\max_{b \in V_B} r_b \leq s \cdot (1 + \Lambda)^{\#V_B} \leq \frac{1}{2} \inf \alpha$.

We are now ready to check volume conditions on $\varphi|_U : U \to G^\star$. Let $v \in V^\star$, and $b = \pi_\star(v) \in V_B$.

$$(\varphi|_U)^{-1}(v) = \varphi^{-1}(v) \cap U_b = \left(\varphi^{-1}(v)\right) \setminus (C_b \cup W_b)$$

Thus, by using the $\alpha$-covering property of $\varphi : S \to G^\star$ and the previous bound $r_b \le \alpha(v)/2$,

$$\# \left( (\varphi|_U)^{-1}(v) \right) \ge \# \left( \varphi^{-1}(v) \right) - (\#W_b + \#C_b)$$
$$\ge \alpha(v) \cdot n_b - r_b \cdot n_b$$
$$\ge \frac{1}{2} \alpha(v) \cdot n_b$$

Thus $(U, \varphi|_U) \in \mathcal{M}_{2\eta}^{\alpha/2}[(G, w), (G^\star, w^\star)]$, which concludes the proof. $\qquad \square$

**Lemma I.5** (Low out-degree of random sparse lifts)**.**
*Let $n : V_B \to \mathbb{N}$ such that if $b \in I_B$ then $n_b = \# p_{\mathcal{C}}^{-1}(b)$. Let $(G, w)$ be a random sparse lift of $\mathbb{B}$ along $(n, \lambda)$ with distribution $\mathcal{N}$.*

*For any $\delta \in \,]0, 1]$, and for $(a, b) \in \#E_B$, let $D_{(a,b)} = 7 \frac{n_b}{n_a} \lambda_{(a,b)} + \log n_a + \log \#E_B - \log \delta$.*

*It holds*

$$\mathbb{P} \left( \forall (a, b) \in E_B, \sup_{v \in \pi^{-1}(a)} \# \left( G(v, -) \cap \pi^{-1}(b) \right) \le D_{(a,b)} \right) \ge 1 - \delta$$

Write $d : E_B \to \mathbb{N}$ the random variable $d : (a, b) \in E_B \mapsto \sup_{v \in \pi^{-1}(a)} \#(G(v, -) \cap \pi^{-1}(b))$.

*Proof of Lemma I.5.* Write for shortness $c = -\log(\delta/\#E_B) \in \mathbb{R}_+$. Let $(a, b) \in E_B$, and $p = \lambda_{(a,b)}/n_a$. Define for all $(i, j) \in [n_a] \times [n_b]$, independent random Bernoulli variables $m_{(i,j)} \sim \mathrm{Bern}(p)$. For $i \in [n_a]$, let $X_i = \sum_{j \in [n_b]} m_{(i,j)}$. Note that these variables have a Binomial distribution $X_i \sim \mathcal{B}(n_b, p)$. Thus from Appendix-Lemma L.2,

$$\mathbb{P} \left( X_u \ge D_{(a,b)} \right) \le \exp \left( -n_b \mathcal{D} \left( \frac{D_{(a,b)}}{n_b} \, \middle\| \, \frac{\lambda_{(a,b)}}{n_a} \right) \right)$$

where $\mathcal{D}(a \,\|\, p) = a \log \frac{a}{p} + (1 - a) \log \frac{1-a}{1-p}$. Observe that $D_{(a,b)}/n_b \ge \lambda_{(a,b)}/n_a$ because

$$\rho = \frac{D_{(a,b)}}{n_b} \frac{n_a}{\lambda_{(a,b)}} = \frac{7 \lambda_{(a,b)}}{\lambda_{(a,b)}} + \frac{n_a \log n_a}{n_b} + \frac{c}{\lambda_{(a,b)}} \frac{n_a}{n_b} \ge \frac{7 \lambda_{(a,b)}}{\lambda_{(a,b)}} = 7 > 1$$

Therefore, we get (S1) by Appendix-Lemma L.3 and (S2) by the previous lower-bound on $\rho$, in

$$\mathcal{D} \left( \frac{D_{(a,b)}}{n_b} \, \middle\| \, \frac{\lambda_{(a,b)}}{n_a} \right) \underset{(\mathrm{S1})}{\ge} \frac{D_{(a,b)}}{n_b} \left( \log(\rho) + \frac{1}{\rho} - 1 \right) \underset{(\mathrm{S2})}{\ge} \frac{D_{(a,b)}}{n_b}$$

because $\log(7) + \frac{1}{7} - 1 \approx 1.09 \pm 0.01 \ge 1$.

Hence so far, we have thus shown that $n_b \mathcal{D} \left( \frac{D_{(a,b)}}{n_b} \, \middle\| \, \frac{\lambda_{(a,b)}}{n_a} \right) \ge M \ge \log(n_a) + c$. Thus,

$$\mathbb{P} \left( X_u \ge D_{(a,b)} \right) \le \exp \left( -n_b \mathcal{D} \left( \frac{M}{n_b} \, \middle\| \, \frac{\lambda_{(a,b)}}{n_a} \right) \right) \le \exp \left( -\log n_a - c \right) = \frac{1}{n_a} e^{-c}$$

Then let $X = \sup_{u \in [n_a]} X_u$, and observe that by definition of a random sparse lift (Def. 4.2), The random variables $X$ and $d_{(a,b)}$ have the same distribution. By union bound,

$$\mathbb{P} \left( d_{(a,b)} \ge D_{(a,b)} \right) = \mathbb{P} \left( X \ge D_{(a,b)} \right) \le \sum_{u \in [n_a]} \mathbb{P} \left( X_u \ge D_{(a,b)} \right) \le \sum_{u \in [n_a]} \frac{1}{n_a} e^{-c} = e^{-c}$$

Then by union bound again, and by definition of the shorthand $c = -\log \left( \frac{\delta}{\#E_B} \right)$

$$\mathbb{P} \left( \exists (a, b) \in E_B, d_{(a,b)} \ge D_{(a,b)} \right) \le \sum_{(a,b) \in E_B} \mathbb{P} \left( d_{(a,b)} \ge D_{(a,b)} \right) \le \#E_B \cdot e^{-c} = \delta$$

By negation, it follows $\mathbb{P} \left( \forall (a, b) \in E_B, d_{(a,b)} \le D_{(a,b)} \right) \ge 1 - \delta$. $\qquad \square$

### I.1.2 Proof of ballwise cover resistance

*Proof of Proposition I.3.*
Write $d : \mathrm{E}_B \to \mathbb{N}$ the function $d : (a,b) \in \mathrm{E}_B \mapsto d_{(a,b)} = \sup_{v \in \pi^{-1}(a)} \#(G(v,-) \cap \pi^{-1}(b))$.

Let $A^{\mathrm{cov}}$ be the event $\{\mathcal{M}_{\eta/2}^{\alpha_{\eta/2}}[(\mathrm{G}, w^0), (\mathrm{G}^\star, w^\star)] \neq \varnothing\}$ (a covering exists at initialization). Then, let $A^{\mathrm{deg}}$ be the event $\{\forall b \in \mathrm{V}_B, \sum_{a \in B(-,b)} d_{(a,b)} \frac{n_a}{n_b} \leq \Lambda\}$ (the lift has relatively low out-degree).

**Degree control.** Let us start by showing that $\mathbb{P}(A^{\mathrm{deg}}) \geq 1 - \delta/2$. In order to use Lemma I.5, let us show first that for any $(a,b) \in \mathrm{E}_B$, it holds $n_a \geq 2\max\{\lambda_{(a,b)}, \lambda_{(a,b)}^2 / \log(2)\}$. By case disjunction on $a$, first if $a \in \mathrm{I}_B$, then on one hand $\lambda_{(a,b)} \leq n_a^0/2 = n_a/2$, and on the other hand $\lambda_{(a,b)} \leq (n_a^0/3)^{1/2} = (n_a/3)^{1/2}$ thus $2\lambda_{(a,b)}^2 / \log(2) \leq (2/\log(2))(n_a/3) \leq n_a$ because $2/\log(2) \approx 2.88 \pm 0.01 \leq 3$. For the other case, if $a \notin \mathrm{I}_B$, then by dropping extraneous factors in the assumption, $n_a \geq 2(1 + \Lambda)^2$ since $\#\mathrm{V}_B \geq 1$, which concludes because $\Lambda \geq \lambda_{(a,b)}$.

For every edge $(a,b) \in \mathrm{E}_B$, define $D_{(a,b)} = 7\lambda_{(a,b)} n_b/n_a + \log n_a + \log \#\mathrm{E}_B - \log(\delta/2)$. By virtue of Lemma I.5, we have
$$\mathbb{P}\left(\forall (a,b) \in \mathrm{E}_B,\, d_{(a,b)} \leq D_{(a,b)}\right) \geq 1 - \delta/2$$
Moreover, for any $b \in \mathrm{V}_B$,

$$\sum_{a \in B(-,b)} \frac{n_a}{n_b} D_{(a,b)} \underset{(\mathrm{S1})}{=} \sum_{a \in B(-,b)} 7\lambda_{(a,b)} + \frac{n_a \log n_a}{n_b} + \frac{n_a}{n_b}\left(\log \#\mathrm{E}_B - \log(\delta/2)\right)$$

$$\underset{(\mathrm{S2})}{\leq} \sum_{a \in B(-,b)} 7\lambda_{(a,b)} + 1 + \log \#\mathrm{E}_B - \log(\delta/2)$$

$$\leq \sum_{e \in \mathrm{E}_B} 7\lambda_e + 1 + \log(\#\mathrm{V}^\star + \#\mathrm{E}_B) - \log(\delta/2) \underset{(\mathrm{S3})}{=} \Lambda$$

where (S1) is the definition of $D_{(a,b)}$, (S2) uses twice the assumption that $n_b \geq n_a \log n_a$ for every $a \in B(-,b)$, and (S3) is the definition of $\Lambda$. Thus
$$\mathbb{P}\left(A^{\mathrm{deg}}\right) \geq \mathbb{P}\left(\forall (a,b) \in \mathrm{E}_B,\, d_{(a,b)} \leq D_{(a,b)}\right) \geq 1 - \delta/2$$

**Initial cover.** Let us show now that $\mathbb{P}(A^{\mathrm{cov}}) \geq 1 - \delta/2$. By leveraging Proposition I.2 as (S1), and writing for shortness, when $\pi_\star(v) = b$, as in the proposition, $\tau(v) = \left(1 - \frac{1}{n_b\,\alpha_{\eta/2}(v)}\right)_+$, we have

$$\mathbb{P}\left(\mathcal{M}_{\eta/2}^{\alpha_{\eta/2}}[(\mathrm{G}, w^0), (\mathrm{G}^\star, w^\star)] \neq \varnothing\right) \underset{(\mathrm{S1})}{\geq} \prod_{b \in \mathrm{V}_B \setminus \mathrm{I}_B} \left(1 - \sum_{v \in \pi_\star^{-1}(b)} \exp\left(-\frac{\tau(v)^2}{4} n_b \cdot \alpha_{\eta/2}(v)\right)\right)_+$$

$$\underset{(\mathrm{S2})}{\geq} 1 - \sum_{b \in \mathrm{V}_B \setminus \mathrm{I}_B} \sum_{v \in \pi_\star^{-1}(b)} \exp\left(-\frac{\tau(v)^2}{4} n_b \cdot \alpha_{\eta/2}(v)\right)$$

$$\underset{(\mathrm{S3})}{\geq} 1 - \sum_{b \in \mathrm{V}_B \setminus \mathrm{I}_B} \sum_{v \in \pi_\star^{-1}(b)} \exp\left(-\frac{1}{8} n_b \cdot \alpha_{\eta/2}(v)\right)$$

$$\underset{(\mathrm{S4})}{\geq} 1 - \sum_{v \in \mathrm{V}^\star} \exp\left(+\log\left(\frac{\delta}{2\#\mathrm{V}^\star}\right)\right)$$

$$\geq 1 - \delta/2$$

where (S2) is a union bound, then (S3) uses $n_b \cdot \alpha_{\eta/2}(v) \geq 4$ thus $\tau(v)^2 \geq (3/4)^2 = 9/16 \geq 1/2$, and (S4) uses $n_b \cdot \alpha_{\eta/2}(v) \geq 8\frac{(R+\eta)^2}{\eta^2}(1+\Lambda)^{1+\#\mathrm{V}_B} \geq 8\Lambda \geq -8\log\frac{\delta}{2\#\mathrm{V}^\star}$.

**Conclusion.** Finally, by using Lemma I.4 for (S1), and a union bound for (S2), we get the bound
$$\mathbb{P}(\forall w \in B(w^0, R),\, \mathcal{M}_\eta^{\frac{1}{2}\alpha_{\eta/2}}[(\mathrm{G}, w), (\mathrm{G}^\star, w^\star)] \neq \varnothing) \underset{(\mathrm{S1})}{\geq} \mathbb{P}(A^{\mathrm{deg}} \cap A^{\mathrm{cov}}) \underset{(\mathrm{S2})}{\geq} 1 - \delta$$

$\square$

## J  QUANTITATIVE PAC CONVERGENCE OF RANDOM SPARSE LIFTS

The statement of this theorem references the definitions of Section 4.2.

**Theorem J.1.** *Let $(\varepsilon, \delta) \in \mathbb{R}_+^* \times\ ]0,1]$.*
*Assume that there exists $G^\star = ((V^\star, E^\star), \pi_\star, c_\star) \in \mathrm{LiftPMod}(\mathbb{B}, \mathcal{C})$ a lifted perceptron module, and parameters $(w^\star, a^\star) \in \mathrm{Param}(G^\star) \times \mathrm{LinReadout}(G^\star, k)$ such that $\mathcal{L}[G^\star](w^\star, a^\star) < \varepsilon$.*

*Let $\mathcal{L}[G^\star](w^\star, a^\star) = \varepsilon_0$, and define the constants*

$$C^\star = \sum_{i \in [k]} \sum_{v \in T(G^\star)} \frac{\|a_{i,v}^\star\|}{\sqrt{\#\pi_\star^{-1}(\pi_\star(v))}}$$

$$\eta = \sup\left\{ \eta \in \mathbb{R}_+^* \,\middle|\, L[G^\star, w^\star](\eta, \mathcal{X}) < \frac{1}{C^\star}\left(\sqrt{\frac{\varepsilon + \varepsilon_0}{2}} - \sqrt{\varepsilon_0}\right)\right\}$$

$$\Lambda = \sum_{(a,b) \in E_B} \left(7\,\lambda_{(a,b)} + 1 + \log(2) + \log(\#V^\star + \#E_B) - \log \delta\right)$$

*Let $\alpha_{\eta/2} : V^\star \to\ ]0,1]$ be the $\alpha$-parameter associated to $(G^\star, w^\star)$ and $\frac{\eta}{2}$ by Def. I.1. Define*

$$c = \sqrt{\sum_{i \in [k]} \sum_{v \in T(G^\star)} \frac{2\,\|a_{i,v}^\star\|^2}{\alpha_{\eta/2}(v) \cdot \#\pi_\star^{-1}(\pi_\star(v))}}$$

*Let $\kappa = 3\,c^{-2}/\|f^\star\|_{\mathcal{D}}^4$. Let $N_1 = V_B \to \mathbb{N}$ be defined for $b \in I_B$ as $N_1(b) = \#p_{\mathcal{C}}^{-1}(b)$ and*

$$\forall b \in V_B \setminus I_B, \quad N_1(b) = 8\left(1 + \frac{4\,c}{\eta}\frac{\|f^\star\|_{\mathcal{D}}^2}{\varepsilon - \varepsilon_0}\right)^2 \frac{(1+\Lambda)^{1+\#V_B}}{\inf_{v \in V^\star} \alpha_{\eta/2}(v)}$$

*It holds, for all $n \in \mathcal{S}_0$ such that $n \succeq N_1$, that if $(G, w)$ is a random sparse lift of $\mathbb{B}$ along $(n, \lambda)$ with distribution $\mathcal{N}$, and $a = 0 \in \mathrm{LinReadout}(G, k)$, then the pair $(\mathcal{L}[G], (w, a))$ satisfies Convergence Criterion 1 with limit error $\varepsilon$ and constant $\kappa$, with probability at least $(1 - \delta)$.*

Theorem 4.3 is a direct consequence of Theorem J.1, it is thus sufficient to prove the latter.

*Proof of Theorem J.1.* Define $\varepsilon_0 = \mathcal{L}[G^\star](w^\star, a^\star) \in \mathbb{R}_+^*$ the error achieved by the witness. Then, define $\varepsilon_1 = (\varepsilon_0 + \varepsilon)/2 \in\ ]\varepsilon_0, \varepsilon[$ the midpoint between the witness error and the target limit error.

**Outline.** The idea for the proof is as follows. First, we use the witness network to get tangent approximation to error $\varepsilon_1 > \varepsilon_0$ on a ball around initialization (by Proposition H.4), provided the network is large enough to have the witness as subnetwork. This is possible because we take $\eta$ small enough to ensure that the gap between the witness error and tangent approximation error is no more than $(\varepsilon_1 - \varepsilon_0)$. Then, for a yet unspecified constant $R$, we leverage Proposition I.3 to set $N_1$ large enough such that the witness is a subnetwork on an $R$-ball with probability at least $(1 - \delta)$, thus we have tangent approximation with error $\varepsilon_1 < \varepsilon$ with high probability. Finally, we set $R$ large enough with respect to the gap $(\varepsilon - \varepsilon_1)$ and the tangent approximation intercept (as per Theorem E.2 and more precisely Lemma E.1) to ensure the limit error of gradient flows is at most $\varepsilon$, and propagate that choice of radius to the definition of $N_1$, which will be sufficient to conclude.

**Proof of Condition C1.** Let us reconstruct the setting of Section 2 with the definitions of perceptron modules. Let $(\mathcal{S}, \preceq)$ be the set $\mathcal{S} = \{n : V_B \to \mathbb{N} \mid \forall b \in I_B, n_b = \#p_{\mathcal{C}}^{-1}(b)\}$, with the order $n^0 \leq n^1$ if and only if $\forall b \in V_B, n_b^0 \leq n_b^1$. For $n \in \mathcal{S}$, define $C = (V_C, E_C)$ the fully-connected lift of $B$ along $n$, and the set $G_n$ of lifted modules[2] $((V_G, E_G), \pi, c) \in \mathrm{LiftPMod}(\mathbb{B}, \mathcal{C})$ with vertices $V_G = V_C$. Let $\Theta_n = \prod_{e \in E_C}(\pi_C^* \mathcal{W})_e$. Define $F_{(n,G)} : \Theta_n \to (\mathcal{X} \to \mathbb{R}^k)$ the forward function $F_{(n,G)}(w, a) = a \cdot F[G](w, -)$. Let us try to show that Condition C1 is satisfied with parameters $(\varepsilon_1, \delta)$ and with intercept $c$. Let $R \in \mathbb{R}_+$, and let $s_1(R) \in \mathcal{S}$ be defined as follows

$$\forall b \in V_B \setminus I_B, \quad [s_1(R)]_b = 8\frac{(R+\eta)^2}{\eta^2}\frac{(1+\Lambda)^{1+\#V_B}}{\inf_{v \in V^\star} \alpha_{\eta/2}(v)}$$

---

[2]Note that there are distinct elements of $G_n$ which may be isomorphic as graphs.

Let $s \succeq s_1$, let $(G, w^0)$ be a random sparse lift along $(s, \lambda)$ and and $a^0 = 0 \in \mathrm{LinReadout}(G, k)$. By Proposition I.3 and by definition of $s_1(R)$, with probability at least $(1 - \delta)$ over $(G, w^0)$, for every $w \in \mathrm{Param}(G)$ such that $\|w - w^0\| \leq R$, it holds that $(G, w)$ has $(\frac{1}{2}\alpha_{\eta/2}, \eta)$-cover of $(G^\star, w^\star)$. Furthermore, by Proposition H.4, this implies that for every $r < R$ and for every $a \in \mathrm{LinReadout}(G, k)$ such that $\|(w, a) - (w^0, a^0)\| \leq r$, writing $\theta = (w, a) \in \Theta_s$, the proposition implies that there exists $u \in \Theta_s$ with $\|u\| \leq c + \|a\| \leq c + r$ and such that

$$\|F_{(s, G)}(\theta) + \mathrm{d}F_{(s, G)}(\theta) \cdot u - f^\star\|_D^2 \leq (\sqrt{\varepsilon_0} + C^\star \cdot L[G^\star, w^\star](\eta, \mathcal{X}))^2 \leq \varepsilon_1$$

This is nearly the $(\varepsilon_1, \delta)$ tangent approximation stated in Condition C1, but the assumption uses a strict inequality. We could proceed by extending the proof of Theorem E.2, but since our bound on the threshold $N_1$ is not tight anyway, we will instead choose to match the assumption exactly. Therefore, define $\varepsilon_2 = (\varepsilon + \varepsilon_1)/2$. Note that $\varepsilon_2 > \varepsilon_1$, therefore we have shown that Condition C1 is satisfied with parameters $(\varepsilon_2, \delta)$. Moreover, by an immediate calculation, $\varepsilon - \varepsilon_2 = (\varepsilon - \varepsilon_0)/4$.

**Proof of PAC-convergence.** Note that it holds $\mathcal{L}[G](w^0, a^0) \leq \|f^\star\|_D^2$ almost surely because $a^0 = 0$. Thus, as per Theorem E.2, set $R_0 = c\|f^\star\|_D^2/(\varepsilon - \varepsilon_2)$ and inject that constant in $N_1 = s_1(R_0)$. Indeed, this matches the definition of $N_1$ because $R_0 = c\|f^\star\|_D^2/(\varepsilon - \varepsilon_2) = 4\,c\|f^\star\|_D^2/(\varepsilon - \varepsilon_0)$ This concludes the proof. $\qquad\square$

## K  PROOFS OMMITTED FROM THE MAIN TEXT

Here we provide several straightforward checks that the definitions introduced in this document are properly defined. All ideas are relatively simple, but the very general notation can sometimes obscure this simplicity. Lifting in particular requires going back to the definition and carefully checking that every type coincides with what we expect. The MLP examples of Fig. 4 and Fig. 5 can help.

### K.1  LIFTING BY HOMOMORPHISM IS WELL DEFINED

**Proposition K.1** (Def. 3.5 is well-defined). *Let $B = (V_B, E_B)$ be a finite directed acyclic graph. Let $\mathbb{B} = ((I_B, T_B), (\mathcal{Y}, \mathcal{Z}, \mathcal{W}), (M, \sigma)) \in \mathrm{PMod}(B)$ be a perceptron module. Let $G = (V, E)$ be a graph, and $\pi : G \to B$ a homomorphism of graphs.*

*Then $G = ((\pi^{-1}(I_B), \pi^{-1}(T_B)), (\pi^*\mathcal{Y}, \pi^*\mathcal{Z}, \pi^*\mathcal{W}), (\bar{M}, \bar{\sigma}))$ is a perceptron module.*

*Proof.* Let us check the conditions of Def. 3.3 one by one. First, $G$ is a directed acyclic graph, because $\pi : G \to B$ is a homomorphism and $B$ is a directed acyclic graph. Then, $I_G = \pi^{-1}(I_B) \subseteq V$, and if $v \in I_G$, then $\pi(v) \in I_B$ therefore $B(-, \pi(v)) = \varnothing$, thus $G(-, v) = \varnothing$ because $\pi$ is a homomorphism. Then $T_G = \pi^{-1}(T_B) \subseteq V$ is immediate. The bundles $\pi^*\mathcal{Y}, \pi^*\mathcal{Z}, \pi^*\mathcal{W}$ over respectively $V, E, E$ are well-defined by pullback (Def. 3.2) through $\pi : V \to V_B$ viewed as a function of sets (respectively $\pi : E \to E_B$ as a function of sets). It remains to check the domains and co-domains of $(\bar{M}, \bar{\sigma})$. First let $(u, v) \in E$, and observe that

$$\bar{M}_{(u,v)} = M_{\pi(u,v)} \in \left(\mathcal{W}_{\pi(u,v)} \times \mathcal{Y}_{\pi(u)} \to \mathcal{Z}_{\pi(u,v)}\right) = \left((\pi^*\mathcal{W})_{(u,v)} \times (\pi^*\mathcal{Y})_u \to (\pi^*\mathcal{Z})_{(u,v)}\right)$$

Then, let $v \in V \setminus I_G$. We have $\pi(v) \in V_B \setminus I_B$ because $I_G = \pi^{-1}(I_B)$, thus $\sigma_{\pi(v)}$ has type signature $\sigma_{\pi(v)} : \prod_{b \in B(-, \pi(v))} \mathcal{Z}_{\pi(u,v)} \to \mathcal{Y}_{\pi(v)}$. Let $z \in \prod_{u \in G(-, v)} (\pi^*\mathcal{Z})_{(u,v)}$, and let us show that $\bar{\sigma}_v(z) \in (\pi^*\mathcal{Y})_v$. For any $a \in B(-, \pi(v))$, we can define $Z_a = \sum_{u \in G_a(-, v)} z_u$, where $G_a(-, v) = \pi^{-1}(a) \cap G(-, v)$. This sum is well-defined because $z_u \in (\pi^*\mathcal{Z})_{(u,v)} = \mathcal{Z}_{(\pi(u), \pi(v))} = \mathcal{Z}_{(a, \pi(v))}$ and $\mathcal{Z}_{(a, \pi(v))}$ is a vector space. Since $G_a(-, v) \subseteq G(-, v)$, the quantity $Z_a$ is a function of $z$. Then the check concludes by $\bar{\sigma}_v(z) = \sigma_{\pi(v)}\left((Z_a(z))_{a \in B(-, \pi(v))}\right) \in \mathcal{Y}_{\pi(v)} = (\pi^*\mathcal{Y})_v$ by definition. $\quad\square$

### K.2  THE PARAMETER $\alpha$ OF A PERCEPTRON MODULE IS WELL DEFINED

Let us check that the parameter $\alpha$ defined in Definition I.1 is well-defined. First, the expression is well-formed because the finite directed graph $G^\star$ is acyclic since it admits a homomorphism $\pi_\star : G^\star \to B$ to an acyclic graph. Then, for all $v \in V^\star$, as a product or non-negative factors lower than 1, $\alpha_\eta(v) \in [0, 1]$. Finally, by induction on $v$, it is immediate that $\alpha_\eta(v) > 0$ provided all factors $p_{\pi_\star(u,v)}(w^\star_{(u,v)}, \eta)$ are strictly positive, which is granted by the full-support assumption on $\mathcal{N}_{\pi_\star(u,v)}$.

### K.3 Subadditivity of the $\mathcal{D}$-seminorm

**Proposition K.2.** *Let $\mathcal{D}$ be a distribution on $\mathcal{X}$ with compact support, and define for $f : \mathcal{X} \to \mathbb{R}^k$*

$$\|\cdot\|_{\mathcal{D}} : f \mapsto \sqrt{\mathbb{E}_{x \sim \mathcal{D}}\left[\|f(x)\|_2^2\right]}$$

*Let $a : \mathcal{X} \to \mathbb{R}^k$ and $b : \mathcal{X} \to \mathbb{R}^k$. Then it holds $\|a + b\|_{\mathcal{D}} \leq \|a\|_{\mathcal{D}} + \|b\|_{\mathcal{D}}$*

*Proof.* Since both sides of the inequality are non-negative, we work directly with squares. By a straightforward computation, using Cauchy-Schwarz inequality in $\mathbb{R}^k$, then $\mathbb{E}[AB]^2 \leq \mathbb{E}[A^2]\mathbb{E}[B^2]$,

$$\begin{aligned}
\mathbb{E}\left[\|a_x + b_x\|_2^2\right] &= \mathbb{E}\left[\|a_x\|_2^2 + 2\langle a_x, b_x\rangle_{\mathbb{R}^k} + \|b_x\|_2^2\right] \\
&\leq \mathbb{E}\left[\|a_x\|_2^2 + 2\|a_x\|_2\|b_x\|_2 + \|b_x\|_2^2\right] \\
&= \|a\|_{\mathcal{D}}^2 + 2\,\mathbb{E}\left[\|a_x\|_2\|b_x\|_2\right] + \|b\|_{\mathcal{D}}^2 \\
&\leq \|a\|_{\mathcal{D}}^2 + 2\sqrt{\mathbb{E}\left[\|a_x\|_2^2\right]}\sqrt{\mathbb{E}\left[\|b_x\|_2^2\right]} + \|b\|_{\mathcal{D}}^2 \\
&= \left(\|a\|_{\mathcal{D}} + \|b\|_{\mathcal{D}}\right)^2
\end{aligned}$$

$\square$

## L  Technical lemmas

**Proposition L.1.** *Let $\lambda \in \mathbb{R}_+^*$, $n \in \mathbb{N}$, $k \in [n]$. If $n \geq \max(2\lambda, 2\lambda^2/\log(2))$, then*

$$\left(1 - \frac{\lambda}{n}\right)^{n-k} \geq \frac{1}{2}e^{-\lambda}$$

*Proof.* Let $g : [0, 1/2] \to \mathbb{R}, x \mapsto \log(1 - x) + x$. We will start by showing that $g(x) \geq -2x^2$. Indeed by computing the derivatives, $g(0) = 0$ and $g'(x) = 1 - 1/(1 - x)$ so $g'(0) = 0$ and $g''(x) = -1/(1 - x)^2$. Observing that $g''(x) \geq -4$ for $x \in [0, 1/2]$, and integrating twice following Taylor-Laplace's theorem with integral remainder, we get

$$\begin{aligned}
g(x) &= g(0) + g'(0)(x - 0) + \int_0^x g''(s)(x - s)\,\mathrm{d}s \\
&\geq \int_0^x -4(x - s)\,\mathrm{d}s = \left[-4\left(xs - \frac{s^2}{2}\right)\right]_0^x \\
&= -2x^2
\end{aligned}$$

Let us now apply this result to $x = \lambda/n \leq 1/2$.

$$n\left(\log\left(1 - \frac{\lambda}{n}\right) + \frac{\lambda}{n}\right) = ng\left(\frac{\lambda}{n}\right) \geq -2\frac{\lambda^2}{n}$$

Therefore, moving the $\lambda$ term from the left to the right hand side, and taking exponentials,

$$\left(1 - \frac{\lambda}{n}\right)^{n-k} \underset{\text{(S1)}}{\geq} \left(1 - \frac{\lambda}{n}\right)^n \underset{\text{(S2)}}{\geq} e^{-2\lambda^2/n} \cdot e^{-\lambda} \underset{\text{(S3)}}{\geq} \frac{1}{2}e^{-\lambda} \tag{7}$$

where (S1) uses $0 < 1 - \lambda/n \leq 1$, (S2) is the exponential of the previous inequality, and (S3) is the assumption $n \geq 2\lambda^2/\log(2)$ which implies $-2\lambda^2/n \geq -\log(2)$. $\square$

**Lemma L.2.** *Let $X \sim \mathrm{Bern}(n, p)$ be a binomial random variable, and $M \in \mathbb{R}_+$.*

$$\mathbb{P}(X \geq M) \leq \exp\left(-n\,\mathcal{D}\left(\frac{M}{n}\,\middle\|\,p\right)\right)$$

*where $\mathcal{D}(a \| p) = a\log\frac{a}{p} + (1 - a)\log\frac{1-a}{1-p}$.*

This lemma is Mulzer (2018, Theorem 2.1), which presents and discusses several proofs.

**Lemma L.3.** *For all* $(a, p) \in {]0, 1[}^2$*, let* $\mathcal{D}\left(\,a\,\|\,p\,\right) = a \log \frac{a}{p} + (1-a) \log \frac{1-a}{1-p}$*. It holds*

$$a \geq p \;\Rightarrow\; \mathcal{D}\left(\,a\,\|\,p\,\right) \geq a \left( \log \frac{a}{p} + \frac{p}{a} - 1 \right)$$

*Proof.* Note that it is sufficient to show that $\log \frac{1-a}{1-p} \geq p - a$. Let $f : [p, a] \to \mathbb{R}$, $u \mapsto \log(1-u)$. This function has derivative $f'(u) = -1/(1-u)$. For $u \leq a$, it holds $f'(u) \geq -1/(1-a)$. Thus,

$$\log \frac{1-a}{1-p} = \Big[ f(u) \Big]_p^a = \int_p^a f'(u) \, \mathrm{d}u \geq \int_p^a \frac{-1}{1-a} \, \mathrm{d}u = \frac{p-a}{1-a}$$

Multiplying both members by $(1-a) \geq 0$ concludes the proof. $\qquad\square$

**Lemma L.4.** *Let* $(X_i)_{i \in [n]}$ *be independent identically distributed random variables with values in* $\{0, 1\}$ *such that* $\mathbb{E}[X_i] \geq \mu \in [0, 1]$*. Let* $X = \sum_i X_i$*. For all* $\delta \in {]0, 1[}$,

$$\mathbb{P}\left( X \leq (1-\delta)n\mu \right) \leq \exp\left( -\frac{1}{2}\delta^2 n\mu \right)$$

This lemma is Mulzer (2018, Corollary 4.3).

**Proposition L.5.** *Let* $S$ *be a finite set with a strict total order* $\prec$*. Let* $r : S \to \mathbb{R}_+$ *be a function such that* $r(\min S) = 0$ *and such that there exist two constants* $(p, q) \in \mathbb{R}_+^2$ *such that*

$$\forall s \in S, \quad r(s) \leq p + q \cdot \left( \max_{a \prec s} r(a) \right)$$

*Then, it holds*

$$\max_{s \in S} r(s) \leq p \cdot (1+q)^{\#S}$$

*Proof.* Let $x : \mathbb{N} \to \mathbb{R}_+$ be the sequence defined by $x_0 = 0$ and $x_{k+1} = p + q \cdot x_k$. By a quick induction, $x_k = p \sum_{i < k} q^i$. The total ordering of $S$ induces $\psi : S \to [\#S]$ an increasing map, called a numbering of $S$. By induction on $k \in [\#S] \subseteq \mathbb{N}$, we will show that $r(\psi^{-1}(k)) \leq \max_{i \leq k} x_i$. The case $k = 0$ is immediate because both are null, and for $s = \psi^{-1}(k+1)$ we have

$$r(s) \leq p + q \cdot \max_{a \prec s} r(a) = p + q \cdot \max_{i \leq k} r(\psi^{-1}(i)) \leq p + q \cdot \max_{i \leq k} x_i \leq \max_{i \leq k} p + q \cdot x_i \leq \max_{i \leq k} x_{i+1}$$

Now observe that for all $k \in \mathbb{N}$, it holds $x_k \leq p \sum_{i \leq k} q^i \leq p \sum_{i \leq k} \binom{k}{i} q^i = p \cdot (1+q)^k$. Thus

$$\max_{s \in S} r(s) \leq \max_{i \leq \#S} x_i \leq \max_{i \leq \#S} p \cdot (1+q)^i \leq p \cdot (1+q)^{\#S}$$

$$\square$$

**Proposition L.6.** *Let* $P \in \mathbb{N}$ *and let* $\{E_i \mid i \in [P]\}$ *be a collection of finite sets. There exists a disjoint collection* $\{V_i \mid i \in [P]\}$ *such that for all* $i \in [P]$*, it holds both* $V_i \subseteq E_i$ *and* $\#V_i \geq \lfloor \#E_i / P \rfloor$*.*

*Proof.* The proof is an application of Hall's marriage theorem. Define the bipartite graph $G$ as follows. Let $c : [P] \to \mathbb{N}$ be $c : i \mapsto \lfloor \#E_i / P \rfloor$. Define the left part of the graph as $X = \coprod_{i \in [P]} [c_i]$, and the right part as $Y = \bigcup_{i \in [P]} E_i$. For any $(i, u) \in X$ (i.e. $i \in [P]$ and $u \in [c_i]$) and $e \in Y$, there is an edge $((i, u), e)$ in $G$ if and only if $e \in E_i$. An $X$-perfect matching in $G$ is an injective function $M : X \to Y$ such that $(x, M(x))$ is an edge of $G$ for every $x \in X$. Such a matching yields a choice of subsets $(V_i = M(\{i\} \times [c_i]))_i$ satisfying the conditions, therefore, let us show that such a matching exists by proving Hall's condition: for every $W \subseteq X$, it holds $\#W \leq \#N_G(W)$ where $N_G(W) \subseteq Y$ are the neighbours of $W$ in $G$. Let $W \subseteq X$. Let $I = \{i \in [P] \mid \exists u, (i, u) \in W\}$. Observe that $\#W \leq \sum_{i \in I} c_i \leq \max_{i \in I} \#E_i$. However by definition of $G$, $N_G(W) = \bigcup_{i \in I} E_i$ thus we get $\#W \leq \max_{i \in I} \#E_i \leq \#(\bigcup_{i \in I} E_i) = \#N_G(W)$, which concludes the proof. $\quad\square$

## M  CONVOLUTIONAL EXAMPLE

We give in Figure 7 an example of fully-convolutional neural network to compute functions $\mathbb{R}^{16 \times 16 \times 3} \to \mathbb{R}$. The $\max$ operation has signature $\max : \mathbb{R}^{8 \times 8} \to \mathbb{R}$ and computes a maximum across the feature map, wich is typical of fully-convolutional networks. The lift dimension of the input is 3, which corresponds to the three color channels of RGB images, and the lift dimension of 6 for the hidden layer corresponds to 6 "channels" of activations in the hidden layer.

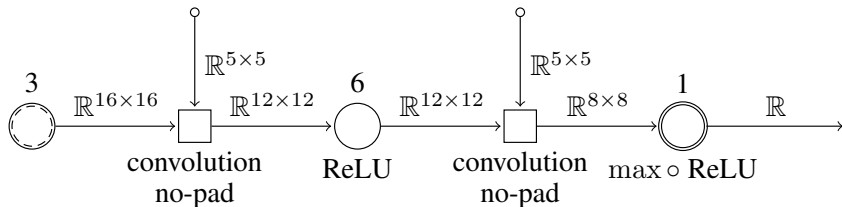

Figure 7: Convolutional perceptron module. Lift dimensions correspond to channels.

## N  EMPIRICAL OBSERVATIONS RELATED TO THE PRESENT THEORY

We include in this section several preliminary experiments to evaluate empirical predictions based on this theory. None of these experiments are sufficient in scale or diversity to claim anything consistent across architectures or tasks. However, we hope that these experiments suffice to show that the theory presented here is not entirely vacuous, and that conducting larger and broader experiments to verify and challenge its predictions is a direction worth pursuing.

The present theory is limited to continuous-time gradient flow for very large lifts, but we hope that future extensions will tackle time-discretization, smaller lifts, and different optimizers. For these reasons, we perform these experiments using an empirical setting closer to the common practices of deep learning, despite the discrepancies that this induces with the theory, to show that it still produces valuable insights even outside its limited application domain at this early stage.

### N.1  QUANTILES OF THE FINAL TRAINING LOSS

In the context of one-dimensional regression, we train random sparse lifts, with two distinct base modules, and plot quantiles of the final training loss reached as a function of the lift dimension.

#### N.1.1  DETAILS OF THE EXPERIMENT SETUP

We consider the one-dimensional regression task of learning the target function $f^\star : \mathbb{R} \to \mathbb{R}$, defined as $x \mapsto a \sin(\omega\,x + \varphi)$, with target parameters $a = 2$, $\omega = 0.5$ and $\varphi = 0.42$.

The training set consists of $n = 10^4$ input points independently sampled uniformly at random from the interval $[0, 100]$, together with the corresponding value for $f^\star$. We use the quadratic loss on $\mathbb{R}$ and the models described in the following section. We train each model for $10^5$ iterations with training samples grouped by batches of 10, taken uniformly at random in the training set with replacement, with the Adam optimizer for a step size of $10^{-2}$. We consider base perceptron modules describing two-layer networks with biases, but with two different activations. For each lift dimension, we plot — over twenty training runs for each lift dimension — the median of the final training loss in orange, the first and third quartile with a box, and the first and last decile with its whiskers.

#### N.1.2  BASE PERCEPTRON MODULES STUDIED

We use as models the random sparse lifts defined by Figure 8 for $\sigma = (\cdot)_+ : x \mapsto \max(0, x)$ and $\sigma = \sin : \mathbb{R} \to [-1, 1]$. Note that for two layers, since the input dimension is one and the last layer is a fully-connected linear readout, neither of those models is effectively "sparse", despite being defined as random sparse lifts. This coincides with the usual definition of a multi-layer perceptron for two layers and activation $\sigma : \mathbb{R} \to \mathbb{R}$, with the additional node above being used to create a bias term in the language of perceptron modules. Lift annotations of vertices are depicted above the vertex.

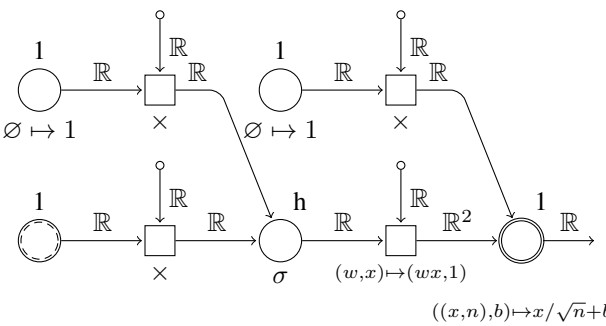

Figure 8: One-dimensional two-layer perceptron blueprint with activation $\sigma$

### N.1.3 EXPERIMENTAL RESULTS AND INTERPRETATION

We plot in Figure 9 the results of this experiment. We observe that, as predicted by the theory, when the lift dimension increases then all quantiles of the final loss decrease (it is unclear whether they will tend to zero as predicted by the theory, see the discussion with the next figure). We do not observe an "incompressible failure probability", of for instance 10% of networks not moving below 1.90 train error regardless of size. We do not observe a significant chance of small networks reaching near-zero loss, despite universality results with low lift dimension. We do not observe across blueprints the same speed of convergence of the error to zero with respect to the lift dimension (note the differences in scales in Figure 9), on the contrary the two blueprints define two very different families of networks in terms of effective learnability at tractable sizes. We do not observe a "sweet spot" in lift dimension, past which large networks would consistently fail to learn or have diverging training loss. All of these observations, despite the notable differences between the experimental setting and the theoretical model (discrete vs continuous time, change of optimizer, etc.), are relatively consistent with the theoretical predictions. This suggests that this type of experiment could be conducted at a larger scale to challenge the predictions of this theory.

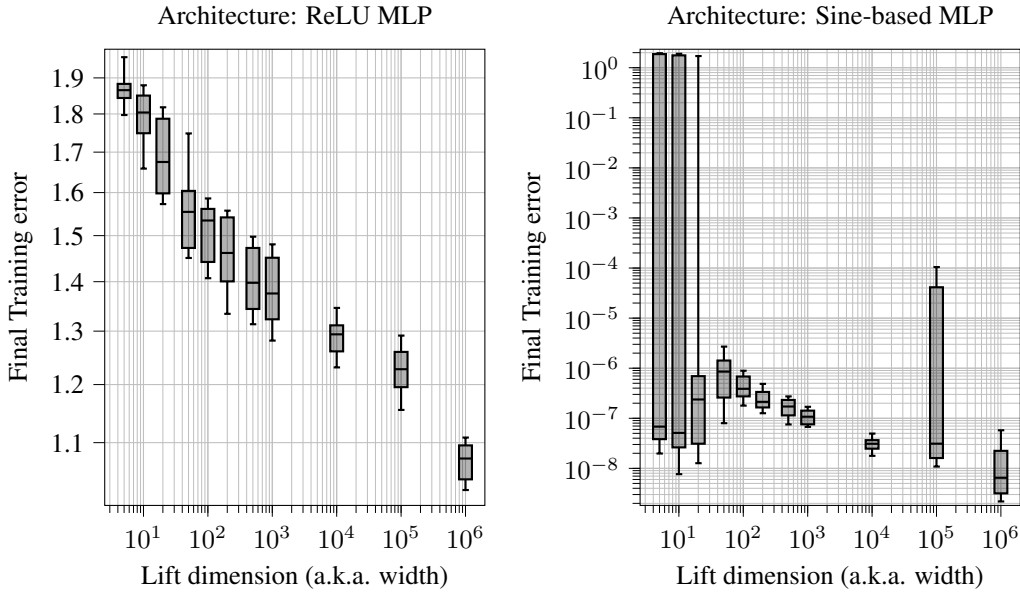

Figure 9: Median, inter-quartile range, first and last deciles of final training loss ($10^6$ iterations).

Since the measurements in Figure 9 are used as estimates of the limit value reached by the loss, we assess the quality of convergence in Figure 10. We plot the same quantiles of the training loss at 80% and 100% of the total allocated training interations: on one hand, for networks that have converged to a critical point, we expect these quantiles to be identical since the parameters should not move when

gradients are null; on the other hand, if they differ significantly, then the experiment would need to be repeated with an increased training budget to rule these considerations out.

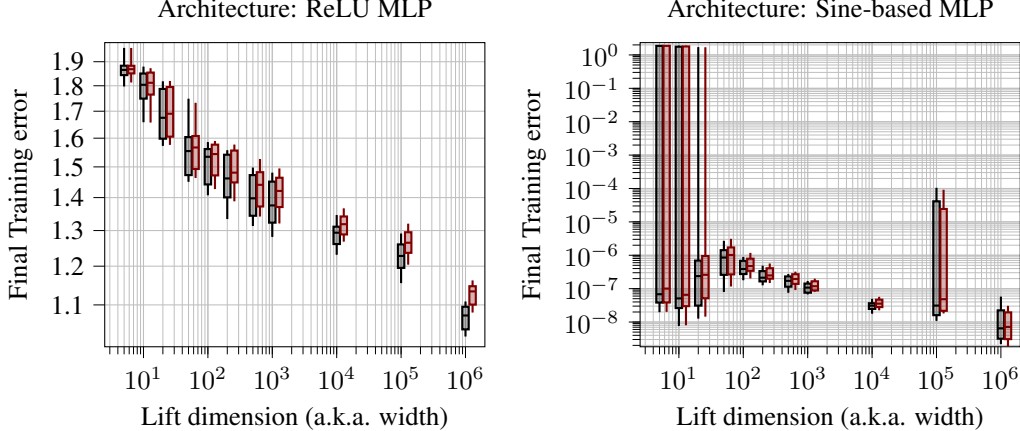

Figure 10: Superposition of quantiles at 100% and 80% (in red, shifted right) of training.

For lift dimensions lower than $10^2$, the results are similar enough that the measured loss can be understood as the final limit. For higher lift dimensions, this conclusion is not as clear, and other replication experiments would be needed to confidently conclude that the observations on these plots are representative of a general behavior of neural networks, rather than only artefacts of this very particular experimental setting. Given the very simple setting, and comparatively very large lift dimensions and training times explored here, these observations remain promisingly similar to the theory's predictions.

## N.2 SPARSE VERSUS DENSE LIFTS IN CLASSIFICATION

In the context of classification, for the MNIST digit-recognition dataset, we perform experiments with dense and sparse multi-layer perceptrons to check whether there exists a fundamental difference which would prevent the extension of this theory to the dense setting.

### N.2.1 DETAILS OF THE EXPERIMENT SETUP

We train several models on the MNIST digit recognition dataset, using the cross-entropy loss and the Adam optimizer with a step-size of $10^{-3}$ by batches of 100 samples taken with replacement, and measure the test accuracy of the resulting models after $10^5$ training iterations.

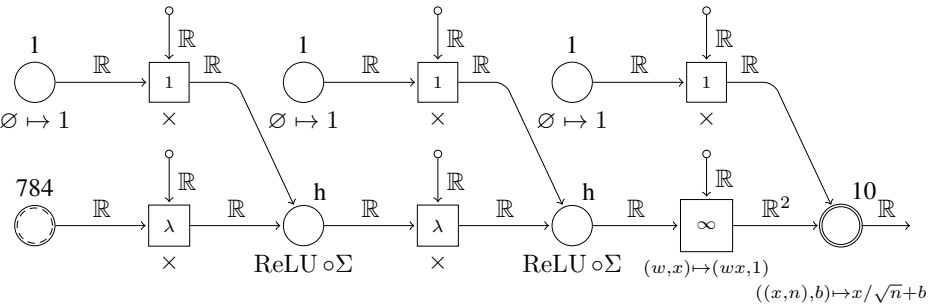

Figure 11: Blueprint of the base perceptron module used for the MNIST experiment. Lift annotations are depicted above the vertices, and inside the boxes for edges ($\infty$ indicates dense connection)

For the dense networks, we use three-layer fully connected perceptrons with hidden widths $h \in \mathbb{N}$ and ReLU activation. We initialize each weight matrix $W \in \mathbb{R}^{n \times m}$ with independent identically distributed entries, with a gaussian distribution of mean zero and variance $1/n$, also known as

"Kaiming He's normal initialization with fan-in mode", and biases with normal distributions. Each "linear" layer computes the function $(W, b, x) \mapsto W \cdot x + b$ (i.e. without a renormalizing factor).

For sparse networks, we use random sparse lifts derived from the blueprint of Figure 11, to have nearly-identical architectures. We use a lift dimension $h \in \mathbb{N}$ identical for all layers, and a degree parameter $\lambda \in \mathbb{R}_+$. We initialize all weights with a normal distribution. Consistently with the definition of linear readouts, the last "linear" operation computes the function $(W, b, x) \mapsto (W \cdot x)/\sqrt{h} + b$ (i.e. with a renormalization factor of $\sqrt{h}$ for the matrix $W \in \mathbb{R}^{h \times 10}$). We vary $\lambda \in \mathbb{R}_+$ the expected incoming degree of hidden nodes, and for both dense and sparse models, we use various powers of two for the lift dimension $h \in \mathbb{N}$.

### N.2.2 Experiment results

The results of this experiment are depicted in Figure 12. In the absence of any regularization or data augmentation, or even large hyperparameter scans, we do not expect particularly impressive performance for either family of models, but only look for differences between the two families. At this small scale, differences in accuracy lower than 1% are to be interpreted with care.

We observe in Figure 12 that for similar lift dimensions, sparse networks perform slightly worse and their performance degrades when the sparsity is more extreme. However this drop remains relatively small, for instance at $h = 128$ there is only a 3% drop from the dense model to the sparse model with $\lambda = 10$. For scale, in the first hidden layer of the dense model, there are $784 \times h \approx 10^5$ parameters, while in the corresponding sparse model there are $\lambda \times h \approx 10^3$. Overall the sparse models appear very similar to their dense counterparts, we do not observe a plateau at a very low (e.g. 70%) accuracy, and we do not observe failures to learn (e.g. order of 10% accuracy) even at large lift dimensions. We also plot in Figure 12 the accuracy as a function of the total number of floating-point parameters, for which the performance is even closer Sparse models appear to outperform dense models with the same number of floating-point parameters, however this gap is too small for conclusions at this scale. In particular, the number of floating-point parameters does not account for the total storage space of the network, since the connectivity of the sparse network also has to be stored (e.g. using one bit per corresponding dense-network parameter, with a naive encoding), which should be accounted for in future larger-scale experiments. We did not include large lift dimensions in sparse networks in this experiment, due to our inefficient implementation of sparse operations with few non-zero entries.

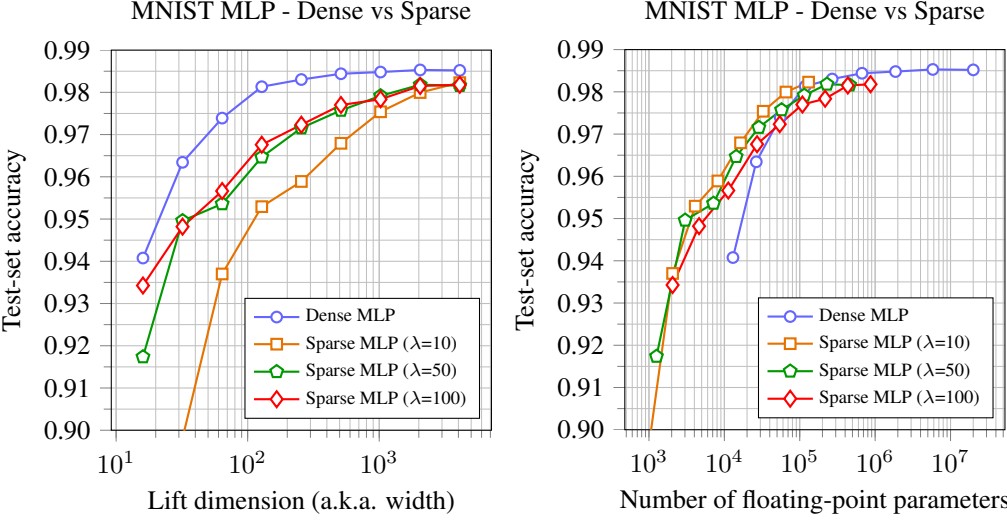

Figure 12: MNIST accuracy of dense and sparse Multi-Layer Perceptrons with ReLU activations

This experiment is overall inconclusive in uncovering a fundamental difference between the behaviors of sparse and dense lifts. This leaves hope that the current proof of convergence, tailored to the sparse case at the moment, will admit extensions to explain the performance of similar dense networks.

