# OpenReview forum: "Random Sparse Lifts: Construction, Analysis and Convergence of finite sparse networks"
_ICLR.cc/2024/Conference — ICLR 2024 poster_

### Official Review · Reviewer_ZupD · 2023-10-18

**Soundness:** 4 excellent
**Presentation:** 4 excellent
**Contribution:** 3 good
**Rating:** 8
**Confidence:** 3

**Summary:**

This work proposes a "lifting" procedure on a general computational graph. Randomizing over these lifts, the authors show that training converges with high probability.

**Strengths:**

1. The paper proposes a really interesting idea which to the best of my knowledge is novel.

2. The authors tackle a notationally dense topic with care and precision, even including a notational glossary in the appendix.

3. Presentation and logical flow are excellent, paper is easy to follow.

4. The theory seems well fleshed out (e.g., showing well-definedness in appendix K), although I did not check the proofs.

**Weaknesses:**

1. It's a shame that the authors did not include concrete experiments training their lifted models. I would be interested to see empirical tests validating the theory: i.e., actually do $n$ random lifts, train via sgd, and see what percent of the time it converges.

2. I have reservations about assumption A1. The set $A_{s,g}(\kappa, \epsilon)$ echoes ideas from neural tangent kernel approximation -- as the authors note, it includes not only weights which are close to $f^*$, but also weights which can be made close to $f^*$ by moving a little bit $u$ in the tangent approximation to the network. The NTK argument justifies this by showing that, under some random initialization, networks are close to their linear approximation as their width goes to infinity. Assumption 1 essentially starts by assuming that the tangent assumption is reasonable, and then states a lower bound on the size of the set of weights that can be made close to $f^*$ by moving in the tangent plane. So in effect they seem to be _assuming_ what NTK analyses _concludes_ from infinite-width scaling. In that light, the fact that this paper claims convergence guarantees for finite-width networks does not seem that impressive.

I am happy to raise my score if either of the above points are adequately addressed, as I actually found the paper quite interesting. I was originally inclined to give a stronger recommendation but my understanding on (2) seems to be a serious limitation.

Small remark / suggestion: some sentences are somewhat difficult to parse due to length, especially in the intro. For example: "Distinct from the fixed-space global optimality of non-convex optimization, this new form of convergence, and the techniques introduced to prove such convergence, pave the way for a usable deep learning convergence theory in the near future, without overparameterization assumptions relating the number of parameters and training samples"

**Questions:**

1. What do the authors mean when they say that networks whose width is not increasing with depth "could be ill-behaved"?

2. I didn't follow the sentence: ""Lastly, if there are many such subnetworks, then a small modification of parameters cannot substantially modify them all if the large network is large enough."

As I am not familiar with the field I cannot give a confident review, although I hope the authors will be able to atleast dispell my concerns about weakness 2 above.

---

> ### Author Response · Authors · 2023-11-21
> **Rebuttal ZupD**
>
> Thank you for your time in reviewing this submission and your valuable comments. We have added a new version as supplementary material (with changes in red) to address your concerns.
>
> **Regarding experiments:**
>
> We agree that concrete experiments would be a valuable addition to this work, and initially did not include them due to space limitations. However, we have nonetheless added two small-scale experiments in the appendix. First, a theoretical validation, in the very simple setting of learning a one-dimensional function, of the fact that quantiles of the loss converge to arbitrary precision, and this convergence depends on the architecture. Second, a more realistic experiment (although of limited scale) of classification on MNIST, to investigate whether sparse networks behave in a significantly different way from their dense counterparts.
> Note that these experiments remain small in scale, and thus admittedly not entirely convincing regarding the impact of sparsity on performance. However, these experiments are an indication that such sparse networks tend to have similar performance to their dense counterparts, and more extensive experiments on larger networks (for vision or NLP tasks) would be an interesting direction for future work.
>
> **Regarding Assumption A1:**
>
> First, we would like to clarify the fact that Assumption A1 is not an assumption made throughout the paper. In particular, the proof of Theorem 4.3 relies on proving that random sparse lifts do verify this assumption, and are thus learnable via gradient flow according to Theorem 2.1.
> Informally speaking, Theorem 2.1 has the form ["A1" implies "convergence"], while Theorem 4.3 proves ["Random sparse lift" implies "A1"], thus implying convergence.
> We made this important point clearer in the text, and apologize for the potential confusion.
> We have renamed "Assumption A1" to "Condition C1" in the updated version.
>
> Second, regarding the link to NTK assumptions. Our analysis is indeed quite strongly linked to the proofs leveraging the NTK in the vastly overparameterized
> regime (more parameters than there are samples), however assumption A1 is much weaker than positive-definiteness of the NTK. While positive definiteness *requires* vast overparameterization to ensure the NTK is not rank-deficient, assumption A1 can hold even with infinitely many samples, provided the architecture has universal approximation, which is indeed what we show later.
> While our results are still ''overparameterized'' in the sense that they require
> much more neurons to ensure learning than are needed to ensure universality,
> we argue that this is a more satisfactory analysis because it lifts the assumption of vast overparameterization. As such, Assumption A1 is a reasonable candidate to replace the assumption of positive-definiteness and extend the analyses using NTKs to networks with finite width and infinite samples.
>
> We show that for networks whose lift dimension satisfies certain condition, we can ensure convergence to arbitrarily good precision with arbitrarily high probability when scaled up.
> In particular, the ``probability of failure'' (the loss stops decreasing at a non-vanishing threshold and thus does not go arbitrarily low) vanishes when scaling up: they are in this sense well-behaved, because this is a desirable property.
> We cannot prove such claims for lift dimension which do not satisfy this condition,
> in particular, it would be consistent with his theory that constant-width MLP
> have an incompressible 1\% failure probability, which does not vanish when the
> width tends to infinity: they would be ill-behaved.
> It appears that this is not the case in experiments, but this theory is insufficient to explain this property.
>
> Regarding your last question, perhaps the confusion is caused by our use of Euclidean norms (not infinite norms). As such, if two disjoint subnetworks use weights $\theta_0$ and $\theta_1$ to compute features $F_i(\theta_0, x)$ and $F_j(\theta_1,x)$
> where $i$ and $j$ are intermediate vertices of the underlying graph,
> then the complete networks may have weights $\Theta = [ \theta_0, \theta_1, \theta_2, \ldots ]$ and compute $F(\Theta, x) = [ \ldots, F_i(\theta_0, x), F_j(\theta_1, x), \ldots ]$.
> Then, if $\Theta'$ is another weight vector such that $\lVert \Theta - \Theta' \rVert_2 \leq 2 \varepsilon$, this implies that  either $\lVert \theta_0' - \theta_0 \rVert_2 \leq \varepsilon$ or $\lVert \theta_1' - \theta_1 \rVert_2 \leq \varepsilon$,
> because $\lVert \Theta - \Theta' \rVert_2 \geq \lVert \theta_0 - \theta'_0 \rVert + \lVert \theta_1 - \theta_1' \rVert$.
> Therefore, either feature $i$ did not move much (because $\theta_0$ didn't)
> or feature $j$ didn't (because $\theta_1$ didn't).
> We use this for the case where feature $i$ and feature $j$
> are close to each other, and with more subnetworks to replace this constant $2$ with an arbitrary $k \in \mathbb{N}$.
> Does this example with concatenations of intermediate weights help address your concern ?

---

> > ### Comment · Reviewer_ZupD · 2023-11-21
> >
> > Thank you to the authors for their thorough replies. Following the clarification on assumption A1 and additional experiments, I believe this paper would be a valuable contribution to ICLR and am recommending acceptance. Nice work!

---

### Official Review · Reviewer_nC6x · 2023-10-29

**Soundness:** 2 fair
**Presentation:** 2 fair
**Contribution:** 3 good
**Rating:** 5
**Confidence:** 2

**Summary:**

This paper proposes random sparse lifts for neural networks and shows their convergence under gradient flow

**Strengths:**

Since the reviewer is not very well versed in this area of research and could not fully follow the technical part of this paper within a short amount of time, the decision should default to the other reviewers.

**Weaknesses:**

See "Questions"

**Questions:**

1. The discussion in Section 2 seems to suggest that the convergence analysis is an extension of the existing NTK analysis, how does the results in this paper different?
2. How should one understand the $\mathcal{C}$ in LiftPMod? The example given in the paper is not very helpful. It would be great if an example of random sparse lift is given for a specific NN architecture, say multi-layer feedforward networks.

---

> ### Author Response · Authors · 2023-11-21
> **Rebuttal nC6x**
>
> Thank you for your time in reviewing this submission.
> We have uploaded as supplementary material a new version with changes in red.
>
> This theory attempts to describe a regime in which the NTK is not positive definite,
> as such it can be understood as a generalization of NTK results to a setting in which they did not previously apply. Note, however that NTK results are typically restricted to dense architectures, while our convergence result is restricted to sparse architectures.
> More precisely, the NTK is constructed as $dF_\theta \cdot dF_\theta^T \in \mathbb{R}^{n \times n}$, where $n$ is the number of samples, and $dF_\theta \in \mathbb{R}^{m \times n}$, where $m$ is the number of parameters.
> In particular, by a rank argument, the NTK cannot have all eigenvalues positive if $m < n$, thus NTK convergence results only apply to the vastly overparameterized setting, of having more parameters than samples. In contrast, our theory extends
> to infinitely many samples, and while our results hold only for large networks,
> they do not require this vast overparameterization assumption.
>
> We have added in the appendix examples with classification on MNIST, where $\mathcal{C} = [28] \times [28] \subseteq \mathbb{N}^2$ are the coordinates of pixels.
> The random sparse lifts then use $n_0 = 784$ as the lift dimension for the input.
> However, one must specify how these are connected to the input grid (e.g. is the grid flattened by row or by column). The fact that $\\# \mathcal{C} = n_0$ is not sufficient to fully specify the network architecture.
> Does this example help clarify your concerns?

---

### Official Review · Reviewer_cG9T · 2023-11-01

**Soundness:** 3 good
**Presentation:** 1 poor
**Contribution:** 2 fair
**Rating:** 5
**Confidence:** 2

**Summary:**

The paper introduces a new class of neural networks, in particular multi-layer perceptrons (MLPs), with provable convergence guarantees when the number of parameters is large. The initial architectures are "lifted" to large sparse models. Tools from graph theory are used to represent these non-standard architectures in a graphical form and discuss the mapping in between them.

**Strengths:**

The graph-based method to represent neural networks seems powerful. I also enjoyed the idea of giving convergence guarantees for the pair of architecture and parameters where the architecture is built in a way to be within the proximity of the target function where the tangent approximation holds.

**Weaknesses:**

Although the approach and results look interesting and might be promising, unfortunately, I found the paper incomprehensible. See some specific points below

* The text is way too informal. The first sentence starts with "trying to learn...".
* Already in the first paragraph, $\theta$ is both an arbitrary variable and the true variable which I found confusing. I'd suggest changing it to "for some parameter $\theta^*$".
* It is not clear why a long list of citations from approximation theory (page 1) is given since the paper studies convergence.
* In many places, claims are unjustified. For example, 'the class of neural networks that can be described .. more expressive than Tensor Programs". Why is that so? It would be OK not to explain this if it was a trivial statement.
* Informal text: "full-support type assumption" on page 2.
* It is not clear to me how Theorem 2.1 helps the paper. As the authors say, it is a slight extension of Robin et al. 2022 with a very similar proof technique. It would help to add an intuitive explanation of how this result will be used in the paper.
* I have not seen how the class of architectures introduced here includes/related to convolutional neural networks.
* The definition of the random sparse lifts (which is in the title of the paper) comes on the last page (Def 4.2). This is very unusual and it makes it nearly impossible to follow up with the text until the last page. I'd kindly suggest arranging the text in the following order: (i) introduce the definitions (only those elements needed for the main theorem) (ii) main theorem (Theorem 4.3) (iii) discuss generalizations to other architectures such as transformers.

**Questions:**

In conclusion, it is stated that the theory of this paper gives a route to strong convergence results and **testable empirical predictions**. Is there any justification for how this method can be used for testable empirical predictions in the paper?

---

> ### Author Response · Authors · 2023-11-21
> **Rebuttal cG9T**
>
> Thank you for your time in reviewing this submission and your valuable comments. We have uploaded an updated version as supplementary material, with changes in red. In particular,
> we apologize for the informal style in which a few passages were written, and have reformulated several parts of the introduction to use a more formal writing style. Please let us know if you think other passages should be reworked as well.
>
> The reason we stressed prior work on approximation results is that our convergence guarantee is given with respect to the infimal loss of all lifts.
> However, this infimal loss is guaranteed to be null only for architectures satisfying a universal
> approximation property.
> Thus, our result turns a  universal approximation guarantee into a universal learning guarantee, and would typically only be applied in settings where universal approximation has already been shown, which led to justify that many settings
> already enjoy such guarantees, with links to the corresponding literature.
>
> Regarding the link to tensor programs, there are two main differences.
> First, tensor programs focus on only one parameter for scaling, whereas
> we use lift dimensions that allow for more scaling dimension, for instance,
> in the example of a multi-layer perceptron with non-constant widths.
> Secondly, tensor programs use weight matrices (whose dimension scales quadratically with respect to the scaling parameter) in a linear fashion, such as to represent the operation $x \mapsto W \cdot x$ with weight matrix $W \in \mathbb{R}^{n \times n}$.
> A random sparse lift similar to this would use a weight matrix $W \in \mathbb{R}^{n \times n}$ and a random connection matrix $B \in \\{0,1\\}^{n \times n}$ to compute
> the operation $x \mapsto (W \odot B) \cdot x$.
> The definition of random sparse lifts however also authorizes the operation
> $ x\mapsto \left[\sum_j B_{i,j} e^{W_{i,j}} x_j / \sum_j B_{i,j} e^{W_{i,j}} \right]_{i \in [n]}$, realized with the maps $M: w, x \mapsto (e^w x, e^w)$ and $\sigma : u,v \mapsto u / v$.
> While the operations described by tensor programs are by far the most common,
> the operations described as random sparse lifts are, in this sense, more general.
>
> We have added in the appendix an example of a convolutional network realized as a lift.
> The lift dimension, in this case, corresponds to an increase in the number of channels.

---

### Official Review · Reviewer_9UCJ · 2023-11-01

**Soundness:** 3 good
**Presentation:** 4 excellent
**Contribution:** 3 good
**Rating:** 8
**Confidence:** 3

**Summary:**

The paper is on the topic of proving global loss convergence for deep neural networks, which is challenging due to the presence of non-convexities. The paper describes a lifting procedure that generalizes the notion of overparameterization in the usual dense, fully-connected settings (for example in deep fully-connected MLPs). The resulting lifted networks can be sparse (which is a big difference to previous analyses of this type), which then allows for tools from algebraic topology to be used to ensure global convergence.

**Strengths:**

The paper is well-written with clear definitions, motivations, sketches and illustration. I find the view of reducing feed forward neural networks to a simpler base module via directed graph homomorphism very refreshing and may have potential applications in other areas of deep learning theory outside of convergence analysis.

Theoretical results presented are general and quantitative. The paper establishes global convergence of gradient flow (in a particular mode of convergence that exists in literature) for a large class of universal approximators and establishes that the learning number (smallest width sizes such that the network can learn) is not too large. This is applicable to a wide range of feedforward architectures, including those with skip connections or attention/self-attention.

The random sparse lift construction is interesting in that it is explicit and only requires a low-loss starting architecture (that can be obtained from experiments). Although I understand this is a theory paper, it would still be interesting for future work to try and construct this sparse life empirically to see if there are any gains in performances.

The assumptions are standard to the best of my knowledge.

**Weaknesses:**

Sparse networks are not very popular in practice and sparsity seems to be a very crucial part of the proof (to obtain fibration of the good smaller computational graphs in any larger computation graph). This is the main reason I'm hesitant to give higher overall score and contribution score.

The proof technique seems very targeted to sparse computational graphs where each node does not have a large influence on other nodes and may require a very different way of thinking about deep learning, compared to fully-connected networks.

**Questions:**

I have not studied the proof in the appendix in detail so my apology if some of these are addressed in the appendix.
- Do you have to maintain full support of the weight distribution throughout training (seeing that you have an assumption on initialization, similar to Nguyen and Pham, 2020)? If so, does sparsity of the network make maintaining full support harder than in the fully-connected case?
- Do you think this result can be extended to classification loss (e.g. logistic, exponential loss)? If so, how do you compare your result to existing convergence results for these loss (for instance, Ji and Telgarsky’s 2020 ‘Directional convergence and alignment in deep learning’)
- It seems from Definition 4.2 that the sparsified graph would have bounded average degree. Is this level of sparsity tight?
- Small typo: Page 2, last paragraph, “S_0 = {(s_0, s_1) \in \mathcal{N}^2 …” should be \in \mathbb{N}^2
- Typo: In section 3, sometimes ‘Euclidean’ is written with a lower case ‘e’.
- It may make things clearer to add interpretation of each item to Definition 3.3. In particular, I’m not sure what T does at first glance (and it’s only explained later on).

---

> ### Author Response · Authors · 2023-11-21
> **Rebuttal 9UCJ**
>
> Thank you for your time in reviewing this submission and your valuable comments. We have uploaded as supplementary material
> an updated version, with all typos corrected and changes marked in red.
>
> Regarding your questions, the full-support of the weight distribution is an assumption of the initialization only, it is, for instance immediately
> satisfied in the case of a multi-layer perceptron with normally-distributed weights,
> because Gaussian distributions have full support.
> We prove that this is sufficient to guarantee that the tangent approximation holds
> for a long time after the initialization, due in part to the sparsification.
>
> We think this result cannot be extended as-is to classification loss,
> but we conjecture that minimal additional assumptions, such as boundedness of
> the non-linearities associated to terminal vertices, will suffice
> to guarantee convergence to low classification loss in a similar regime.
> While we expect directional convergence, as studied by Ji and Telgarsky,
> for the last layer weights, this might yield interesting extensions
> to intermediate features which are not positively homogeneous (e.g. sigmoids instead of ReLUs, or more complicated activations).
> Note that the convergence we expect may be only a convergence in loss,
> and the convergence of all parameters, especially for intermediate layers,
> may require stronger assumptions.
>
> We have tightened our analysis to this specific kind of sparsity,
> and we suspect that other sparse regimes may have similar guarantees with varying
> details. As an example of varying detail, the set $S_0$ uses a growth
> of the width like $n \log n$ between a node and its parent,
> but other sparsifications may require
> a growth like $n^2$.
> Tightening the set of admissible lift dimensions may require tools that
> are tailored to the particular sparsification distribution.

---

> > ### Comment · Reviewer_9UCJ · 2023-11-21
> >
> > I'd like to thank the authors for their comment and confirm that I am keeping my score.

---

### Meta-Review · Area_Chair_47Z1 · 2023-12-05

**Metareview:**

This paper presents a framework for constructing a class of neural networks where gradient flow will reach low loss (with growing number of parameters). The constructions lead to large sparse networks which allows new techniques to be used for convergence. The reviews are overall positive although there is a separation. Some reviewers like the new approach and the strong guarantees, while other reviewers have minor concerns about what these results would imply in realistic scenarios. The paper can be made stronger by including experiments on the proposed model.

**Justification For Why Not Higher Score:**

I personally agree more with the slightly negative reviews. The paper uses a lot of math but at the end of the day it doesn't seem to go very far beyond having some random features (although they are much more structured in this paper, which makes it interesting).

**Justification For Why Not Lower Score:**

The reviewers are overall positive, and there are some interesting ideas in the paper.

---

### Decision · Program_Chairs · 2024-01-16

Accept (poster)